# MILA (Multilingual Indic Language Archive): A Dataset for Equitable Multilingual LLMs

## Abstract

Large Language Models (LLMs) are structurally biased toward high-resource languages like English due to corpus skew, a problem particularly severe for Indic languages. To address this deficit, we introduce **MILA**, the largest expert-curated Indic corpus to date, comprising **7.5 trillion tokens** across **16 scheduled Indic languages** and English. MILA is constructed via a multi-stage data engineering pipeline that integrates large-scale **web acquisition**, script-sensitive **OCR** for under-digitized Indic writing systems, LLM-assisted post-correction for **Translation** fidelity, and **targeted data distillation** through the **Indic-Persona Hub**. The pipeline further incorporates **synthetic augmentation and rewriting**, followed by stringent **quality, toxicity, language, and deduplication filtering**, and culminates in **human-in-the-loop linguistic** and cultural validation with comprehensive **PII redaction** and ensuring **downstream task and benchmark-based decontamination**. This pipeline yields a distributionally stable, contamination-controlled, high-fidelity pretraining substrate. Alongside, we release **Indic-MMLU**, a translated and verified adaptation of MMLU into 16 Indian languages, offering the first large-scale Indic multilingual benchmark for assessing LLMs and their extent of cross-lingual knowledge transfer. We further propose a **Parity-based fairness Metric** capturing cross-lingual performance asymmetries relative to English. Comprehensive experiments including controlled ablations of translation quality, OCR incorporation, synthetic SFT generation, and continual pretraining demonstrates that models trained on MILA achieve substantial gains on Indic-MMLU and materially narrow cross-lingual disparities. Collectively, MILA, Indic-MMLU, and the associated validation protocols establish a scalable foundation for equitable multilingual modeling in the Indic context. All resources are released anonymously for reproducibility.[1]

## 1 Introduction

The trajectory from early monolingual language models to modern multilingual architectures reflects the rapid consolidation of Transformer-based NLP. Foundational models such as BERT (Devlin et al., 2018) and GPT (Radford & Narasimhan, 2018; Radford et al., 2019; Brown et al., 2020) established the efficacy of self-attention yet operated within limited linguistic regimes. Subsequent multilingual systems including mT5 (Xue et al., 2020), XLM-R (Conneau et al., 2019), Bloom (Muennighoff et al., 2023), LLaMA (Touvron et al., 2023a;b; Grattafiori et al., 2024), Gemma (Team et al., 2024a;b; 2025), Mistral (Jiang et al., 2023), Qwen (Bai et al., 2023; Yang et al., 2024; Qwen et al., 2025; Yang, 2025), and Nemotron (Nvidia et al., 2024) extended this breadth, enabled by massive corpora such as Common Crawl (Common Crawl Foundation), Wikipedia (Wikimedia Foundation), CCMatrix (Schwenk et al., 2019), mC4 (Xue et al., 2020), OSCAR (Ortiz Suárez et al., 2019), and Dolma (Soldaini et al., 2024). However, global data distributions remain starkly imbalanced: Indic languages, despite their demographic scale, are acutely underrepresented, rendering high-quality tokens disproportionately impactful. This structural asymmetry constrains multilingual generalization. We introduce MILA, a 7.5T-token corpus across 16 Indic languages with OCR, translation, and synthetic augmentation, alongside Indic-MMLU and a cross-lingual fairness metric.

---

[1] https://github.com/anonymous-submitter0104/iclr-submission

## 2 RELATED WORK

Large-scale corpora like RedPajama (Weber et al., 2024), SlimPajama (Shen et al., 2024), DCLM (Li et al., 2025), Pile (Gao et al., 2021), Zyda (Tokpanov et al., 2024b;a), TxT360 (Tang et al., 2024) power LLMs yet remain mostly English-biased. Multilingual datasets such as mC4 (Xue et al., 2020), OSCAR (Ortiz Suárez et al., 2019), CC100, ROOTS (Laurençon et al., 2023), ParaCrawl (Bañón et al., 2020), FineWeb2 (Penedo et al., 2025), CulturaX (Nguyen et al., 2023), MultiUN (Eisele & Chen, 2010), Dolma (Soldaini et al., 2024) improve coverage but sparsely represent Indic languages. Indic-focused corpora (Samanantar (Ramesh et al., 2022), Sangraha Synthetic (Khan et al., 2024), IndicCorp (Doddapaneni et al., 2023)) trade scale for fidelity. Existing curation pipelines (Lee et al., 2022; Khan et al., 2025; Zhang & Salle, 2023; Sharma et al., 2024) degrade on noisy, code-mixed Indic data (Ousidhoum et al., 2025); OCR remains error-prone (Mathew et al., 2024), and synthetic augmentation often misaligns culturally (Ousidhoum et al., 2025; Yu et al., 2022). Evaluation benchmarks FLORES (Goyal et al., 2022), IndicGenBench (Singh et al., 2024), MILU (Verma et al., 2025) reveal persistent English–Indic performance gaps, motivating parity-aware metrics for equitable multilingual modeling.

**Contributions:** We introduce MILA, the largest and most diverse curated Indic multilingual dataset, supported by novel data curation and production pipelines tailored for India's linguistic landscape. Our data curation recipes include in-house quality filters for Indic languages, encompassing toxicity and low-quality content detectors. Data production recipes/pipelines comprise (i) a scalable OCR system for digitizing Indic books, (ii) a high-fidelity translation pipeline for 16 languages, (iii) the Indic Persona Hub for persona-conditioned data generation, and (iv) large-scale synthetic rewriting and augmentation strategies. We further present Indic-MMLU, the first comprehensive multilingual evaluation suite across major Indian languages, designed to benchmark reasoning and knowledge abilities robustly. Finally, we open-source all components, including the Indic-MMLU benchmark, the full MILA dataset (OCR + ISOB, translations, synthetic rewrites, the virtual Indian personas as well as persona-generated data, and high-quality Indic web crawl), and a large collection of image-text pairs to facilitate future VLM and Indic OCR model development.

## 3 PARADIGMS IN DATA PREPARATION

### 3.1 DATA ACQUISITION AND GOVERNANCE

Our corpus is assembled via a multi-pronged acquisition pipeline integrating large-scale Indic web crawling, institutional and archival digitization, and license-compliant open datasets (see Appendix). All sources are normalized under a unified provenance framework with standardized metadata (ISBN/DOI/archive IDs), URL/MD5 deduplication, and quantum identifiers for traceability. High-concurrency, source-specific crawlers and optimized ingestion pipelines enable efficient, scalable processing. The resulting corpus is a large, diverse, culturally grounded, and reproducible resource suitable for large-scale Indic pretraining; full licensing, source audits, and acquisition statistics are provided in Appendix D, Supplementary repository. [2], Overall Distribution & Open Release A

### 3.2 DATA CURATION

High-quality data is essential for building robust multilingual foundation models; noisy, low-quality, or misclassified text degrades linguistic fluency, factual grounding, and safety (Paullada et al., 2021; Liu et al., 2024; Yu et al., 2024; Rae et al., 2022). Indic languages introduce unique curation challenges—diverse scripts, rich morphology, OCR-induced artefacts, and extensive code-mixing. To address these, we develop a multi-stage, Indic-specific curation pipeline inspired by general-purpose data curation frameworks and augmented with custom language-aware modules. The resulting corpus is clean, diverse, safe, culturally aligned, and legally compliant. **In-House Quality Filters.** We train language-specific fastText classifiers to categorize documents into High, Medium, or Low quality. High-quality text is defined by two criteria: (i) Linguistic well-formedness, including correct script and Unicode rendering, minimal OCR noise, absence of boilerplate or HTML artefacts, reduced off-script code-mixing, and no toxicity or spam; (ii) Semantic coherence, requiring multi-sentence passages with clear discourse structure, consistent propositions, and absence of stitched,

---

[2]https://github.com/anonymous-submitter0104/iclr-submission/tree/main/data-acquisition

malformed, or machine-broken text. For each language, we sample approximately 450K passages from OCR, Crawl, Translation, and Synthetic data. We additionally construct adversarial low-quality examples (script mixing, character noise, reordering, punctuation removal, synthetic corruptions). Labeling uses (a) LLM-based instruction-following and (b) heuristics that detect script coverage and boilerplate. **Multilingual Language Identification.** We apply a fastText-based language filter enhanced with regex-driven rules to ensure script–language consistency across Indic languages. This mitigates cross-script contamination, Romanized drift, and mixed-language artefacts common in web sources. **Heuristic Filters.** We incorporate structural and content-level modifiers (Table 1) that normalize text by removing boilerplate strings, HTML tags, malformed Unicode, inconsistent quotations, and excess whitespace. Additional heuristics eliminate degenerate content through word-count thresholds, repeated $n$-grams ($n = 2, 3$), excessive URLs, symbol-heavy passages, and number-dominated text. **Deduplication.** A two-level deduplication pipeline removes redundancies

Table 1: Modifiers and Heuristic Filters.

| Modifiers | Heuristic Filters |
|---|---|
| Boilerplate String Modifier | Word Count Filter |
| HTML Tag Modifier | Repeating Top -n-grams Filter $n = 2, n = 3$ |
| Unicode Reformatter | URLs Filter |
| Quotation Unifier | Symbols to Words Filter |
| Excess White Space Remover | Numbers Filter |

across sources. Exact duplicates are removed via URL/MD5 signatures; GPU-accelerated fuzzy deduplication eliminates near-duplicates such as templated pages, OCR variants, and lightly modified copies (Lee et al., 2022; Khan et al., 2025). This reduces memorization and improves pretraining efficiency. **Toxicity Filtering.** Following recent multilingual safety literature (Mendu et al., 2025), we adopt a two-stage toxicity filter: (i) rule-based scanning to eliminate explicit harmful content, and (ii) a multilingual RoBERTa classifier that recovers false positives and refines borderline cases. (Ablations in App B.1.2) **PII Redaction.** We integrate a multilingual PII removal module that identifies and redacts personally identifiable information in Indian languages, ensuring compliance with privacy and data-protection requirements. (Appendix B.2) **Decontamination.** We perform benchmark and task decontamination across OCR, Crawl, Translation, and Synthetic corpora to prevent leakage into pretraining. We combine n-gram overlap filtering (8–13 grams) with Infinigram-based cross-domain detection. Contamination Stats in Appendix Table 8 Details:B.1.3. Overall Curation Workflow 1. Supplementary.[3]. Overall Curation Ablation refer B.1

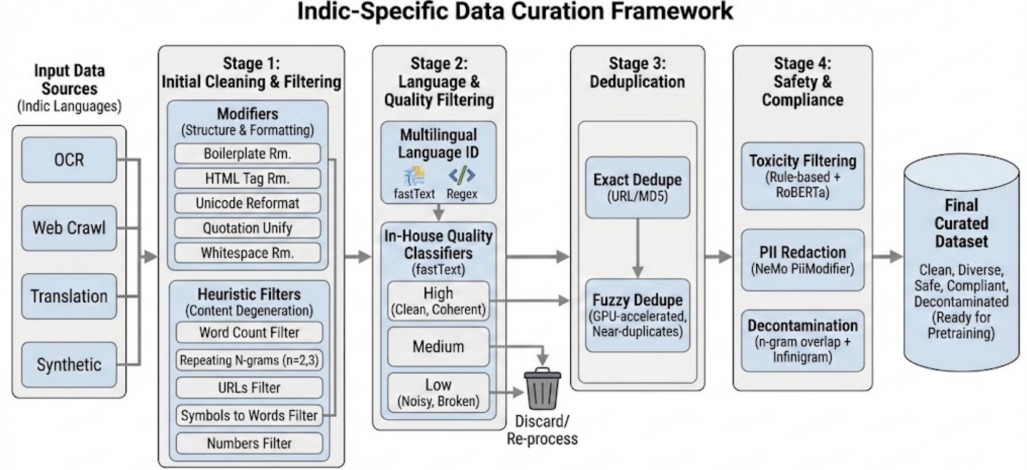

Figure 1: Overview of the Multi-stage, Indic-specific data curation pipeline, from raw input sources to the final high-quality pretraining dataset.

---

[3]https://github.com/anonymous-submitter0104/iclr-submission/tree/main/data-curation

## 3.3 DATA PRODUCTION

### 3.3.1 OCR PIPELINE

High-quality OCR for Indic languages is a fundamental prerequisite for constructing native vocabularies and producing reliable pretraining corpora. The diverse script families, complex ligatures, heterogeneous typography, and the prevalence of noisy or degraded scans make Indic OCR significantly more challenging than Latin-based pipelines. We therefore design a two-stage, language-specific OCR pipeline that combines curated human evaluation, scalable LLM/VLM-based assessment, and post-correction via high-capacity LLMs. Full model lists, benchmark details, and ablations appear in the Appendix. B.3.4 B.3.2, B.3 Supplementary Repository. [4] Additional Benchmarks and Details I

**Pipeline Overview.** For each language, the pipeline consists of (i) OCR/Parsing and (ii) Post-Correction, preceded by a unified preprocessing module. We prioritize layout preservation, including block structure and reading order, as this improves contextual grounding during pretraining. For every language, we evaluate a pool of state-of-the-art OCR and VLM models on public Indic OCR datasets and our in-house **Indic Small OCR Benchmark (ISOB)**, yielding a shortlist of top-performing candidates.

**Stage 1: OCR / Parsing.** We first construct a representative page sample covering: crawled books, partner-sourced documents, and a spectrum of "easy to hard" pages. A strong VLM (e.g., Qwen-VL-32B) classifies pages by OCR difficulty based on visual cues. Each page is processed using the top-$k$ candidate OCR/VLM models. Native linguists evaluate outputs for: native-word preservation, spelling accuracy, absence of spurious artifacts, and completeness. To scale beyond human throughput, we adopt a VLM-LLM-as-Judge framework. A reasoning-capable VLM produces chain-of-thought evaluations on the same linguistic criteria on a larger, more diverse pool of samples than that of human linguists; a second LLM independently verifies the reasoning trace and final scores given by the VLM for consistency. Linguist and LLM scores are aggregated to select a consensus Stage-1 model per language. Expanded benchmarking results are reported in the Experiments Appendix I and B.3.2

**Stage 2: Post-Correction.** Even state-of-the-art OCR/VLM systems exhibit minor but systematic errors such as spelling inconsistencies, missing graphemes, layout-induced mis-segmentation, and low-frequency artifacts. These subtle errors accumulate at scale and are costly to correct manually. Therefore, we select a post-correction LLM via a language-specific benchmark suite focusing on the following linguistic criteria: contextual fidelity, native fluency, factual alignment, tone preservation, hallucination resistance, and topic consistency. Refer App for Ablation on Post-Correction B.3.4

The post-correction engine operates at the page level, consuming: (i) raw OCR output, (ii) outputs from other top OCR candidates, (iii) summaries of the preceding and following pages, and (iv) detailed reasoning traces generated during linguistic evaluation by reasoning-based VLM. Low-quality or "hard to OCR" pages invoke the full reasoning-based correction workflow; high-quality pages undergo a lightweight language-specific LLM based post-correction pass. The objective is meaning preservation over exact lexical fidelity, preventing semantic drift while allowing minor lexical normalization that benefits pretraining.

**Production at Scale.**

1. **Pre-processing.** Each batch is processed by a VLM to infer page orientation, blur/noise levels, and OCR difficulty. Misoriented pages are corrected, and noisy scans are denoised using standard image enhancement.

2. **Two-Stage Execution.** Stage 1 produces raw OCR pages and classifies them as high- or low-quality using the LLM classifier. High-quality pages directly enter Stage 2. Low-quality and "hard" pages trigger the multi-context LLM correction pipeline described above.

3. **Human-in-the-loop Verification.** A stratified sample from all quality tiers is reviewed by native linguists for **cultural and contextual sensitivity, tone, factual alignment, topic consistency, and hallucination**. A larger sample is evaluated by LLMs using the same criteria. Consensus between human and LLM evaluations determines batch acceptance.

---

[4]https://github.com/anonymous-submitter0104/iclr-submission/tree/main/ocr-pipeline

**Empirical Observations.** Stage-2 post-correction consistently improves linguistic quality, semantic coherence, and downstream pretraining convergence. While post-corrected text may deviate lexically from the raw OCR, semantic content is preserved and often expressed with improved consistency. In early pretraining phases, both high- and low-confidence outputs are beneficial; during annealing, we recommend upsampling high-quality OCR pages yield measurable gains due to increased exposure to native, cleaner text. Detailed ablations are provided in Appendix B.3.

**ISOB Benchmark.** We release the Indic Small OCR Benchmark (ISOB-Small), covering 16 languages with synthetic and naturally occurring OCR artifacts, enabling systematic evaluation of OCR/VLM systems. Dataset construction details, annotation guidelines, and benchmark recipes are documented in the Appendix & Supplementary repositories [5], ISOB end2end pipeline 5 B.3.1

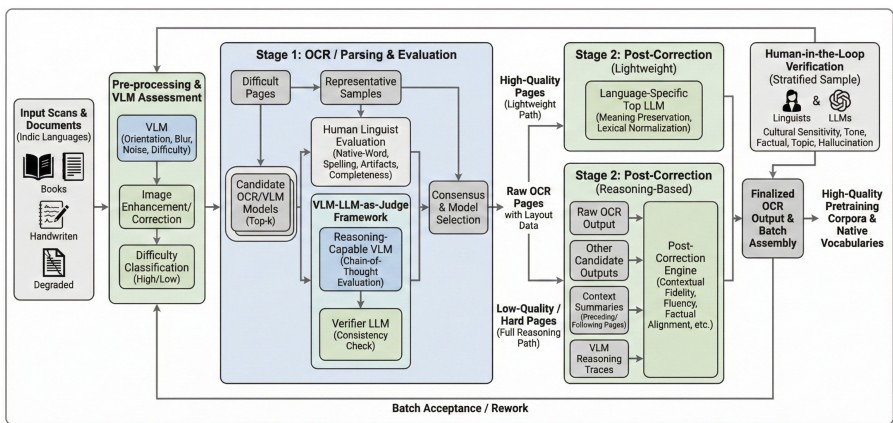

Figure 2: Illustrative diagram of the Two-stage Indic OCR pipeline, combining scalable VLM/LLM-based assessment with targeted human evaluation and a split post-correction workflow for high- and low-quality pages.

### 3.3.2 TRANSLATION PIPELINE

Translating long-form English text into 16 Indic languages presents several challenges: preserving document-level coherence, ensuring culturally appropriate vocabulary, handling mathematically or structurally complex content, and maintaining consistency across long passages. To address these issues, we design a **hybrid MT→LLM translation pipeline** that combines the precision of specialist Machine Translation (MT) models with the fluency and reasoning capabilities of modern LLMs. This section integrates our empirical findings, model analyses, and production workflow. Extended model lists, ablations, and metrics appear in Appendix B.4 H Refer Figure, End 2 End Pipeline: 7, Supplementary Repository [6]

**Motivation and Empirical Observations.** As high-quality open-source MT systems and multilingual LLMs continue to evolve, we maintain an expanding pool of Indic-language translation benchmarks and routinely evaluate newly released models. Two consistent patterns emerged. First, *specialist MT models* exhibit exceptionally low error rates and strong lexical fidelity, especially for terminology-heavy or domain-specific text. However, their outputs often sound overly rigid or "robotic," with limited natural variation in phrasing. Second, *generalist LLMs* produce more coherent and natural-sounding translations but occasionally sacrifice precision, for example, weaker vocabulary grounding, inconsistent handling of rare words, or subtle semantic drift.

These observations motivate a hybrid approach in which MT outputs provide lexical grounding while LLMs supply fluency, cultural nuance, and long-context consistency. Through controlled overlap-based analysis, we found that LLM rewriting of MT outputs, with small contextual windows from preceding chunks, significantly improves fluency *without* degrading meaning preservation. Our pipeline comprises three tightly integrated components: Translation End 2 End Pipeline Diagram: 7

---

[5]https://github.com/anonymous-submitter0104/iclr-submission/tree/main/opensource-release/isob-small-hard

[6]https://github.com/anonymous-submitter0104/iclr-submission/tree/main/translation-pipeline

1. **Specialist MT Generation.** We first translate each document using high-coverage specialist MT systems. These models preserve terminology and reduce hallucination risk, making them ideal as the grounding backbone of the pipeline. (Baseline Benchmarkings Appendix H)

2. **LLM-Based Post-Correction.** Generalist LLMs refine MT output to improve readability, vocabulary richness, cultural appropriateness, and stylistic naturalness. The MT output serves as a semantic anchor, preventing LLM-induced deviations or hallucinations. (Baselines discussion and detailed experiments: B.4.1 B.4.2 H)

3. **Long-Context and Overlap Handling.** Documents are chunked into segments $\{C_1, C_2, \ldots, C_n\}$. For each chunk $C_i$, the LLM receives: (i) the specialist MT translation of $C_i$, (ii) a compact summary of $C_{i-1}$ (and optionally $C_{i-2}$), and (iii) a brief English-context summary. Among four configurations tested, MT-only, LLM-only, LLM-refine-MT, and LLM-refine-MT with contextual overlap, the final configuration achieves the best balance of fluency and semantic fidelity for most languages.

**Quality Classification and Selective Refinement.** Refining all outputs with an LLM is computationally expensive and unnecessary. Instead, we employ a *two-level LLM classifier* to selectively route documents:

- **Hard-to-Translate Detector (pre-translation).** Before translation, we classify incoming pages into easy vs. hard categories. Hard cases include text containing math, code, tables, poetry, dense technical jargon, or culturally sensitive content.
- **Translation Quality Classifier (post-translation).** After Stage 1 MT output, a second LLM classifier scores each page on fluency, adequacy, structure, and terminology preservation. Pages are categorized as *high-quality* (requiring only lightweight LLM refinement), *low-quality*, and *hard cases* (requiring deeper reasoning-based correction).

**Reasoning-Based Correction Workflow.** Low-quality and hard pages are routed to a multi-context reasoning-enabled LLM. This model receives the English source chunk, the MT output, the LLM-refined candidate (if available), and surrounding context summaries. A verifier LLM adjudicates the final output, preventing semantic drift and ensuring meaning preservation over exact lexical fidelity. High-quality pages skip this heavy pipeline and undergo a lightweight post-correction step that normalizes style and improves naturalness.

**Human Evaluation Protocol.** To select the per-language production configuration, we conduct a multi-stage human evaluation. Three native linguists are assigned to each of the 16 languages, covering domains such as conversational writing, literature, technical text, mathematics, code, and administrative documents. All instances are rated on a 1–5 scale across seven criteria: (1) fluency, (2) adequacy and meaning preservation, (3) vocabulary richness, (4) cultural appropriateness, (5) grammar and syntax, (6) cross-chunk consistency, and (7) overall readability. We compute Krippendorff's $\alpha$ and achieve substantial agreement ($\alpha > 0.68$). Instances with disagreement undergo adjudication. For languages with lower agreement or morphologically complex structures, an additional reasoning-based LLM refinement step substantially improves semantic accuracy and structural consistency.

**Production Deployment at Scale.** After selecting optimal configurations per language, we deploy the following production pipeline:

1. **Pre-Processing.** An LLM classifier identifies free text vs. math, code, tables, or linguistically challenging segments.
2. **Two-Stage Translation.**
   - *Stage 1:* Specialist MT systems generate raw translations for each chunk and classify quality.
   - *Stage 2:* High-quality pages receive lightweight LLM post-correction; low-quality or hard pages are routed through full reasoning-based correction with a verifier LLM.
3. **Human-in-the-Loop Verification.** Stratified samples from each quality tier are reviewed by professional linguists for cultural sensitivity, tone correctness, structural fidelity, and hallucination detection. A larger sample is evaluated by LLMs using identical criteria. Batch acceptance requires consensus between human and LLM evaluations.

**Summary.** Our hybrid MT→LLM pipeline, enhanced with overlap checks and reasoning-based refinement, consistently surpasses MT-only and LLM-only baselines. MT models provide grounding

and accuracy, while LLMs improve fluency, cultural fit, and long-context coherence. The resulting translations are high-quality and semantically faithful for multilingual pretraining and evaluation.

### 3.3.3 DATA DISTILLATION VIA INDIC PERSONAHUB: CONSTRUCTING CULTURALLY-GROUNDED SYNTHETIC POPULATION

The **Indic PersonaHub** is a large-scale synthetic population designed to capture the linguistic, cultural, and cognitive diversity of the Indian demographic. Unlike global persona datasets, our framework is strictly grounded in Indian contexts, constructing 200 million unique virtual "citizens" derived from proprietary Indian-language web crawls, regional literature, and localized digital footprints. Persona generation follows a dual-strategy protocol to maximize both sociodemographic coverage and relational depth. **Text-to-Persona.** This bottom-up strategy captures the long tail of Indian society. Diverse texts from web and literature sources, ranging from regional blogs and niche technical forums to village records, are processed by a reasoning-heavy LLM prompted to infer detailed sociodemographic profiles. Grounding persona generation in observed distributions ensures alignment with real-world linguistic, occupational, and cultural patterns, avoiding mode collapse. **Persona-to-Persona.** To address underrepresented groups, such as the elderly, informal laborers, or rural homemakers, a top-down relational expansion leverages high-confidence seed personas to generate additional profiles through social graph modeling. The model infers plausible social connections, including family, occupational peers, and community roles, producing a cohesive and interconnected virtual society. **Filtration and Refinement.** Raw persona profiles are mapped against a comprehensive Indian Demographic Taxonomy (state, language, profession, urban-rural) to monitor coverage and trigger targeted generation for underrepresented subgroups. Semantic deduplication ensures the 200 million personas represent distinct viewpoints rather than duplicates. **Task Assignment and Synthetic Generation.** Each persona receives a personalized task tailored to its niche expertise and background, and presents its reflections in the context of India. The LLMs then generate responses that remain consistent with the persona's characteristics, after which all outputs are passed through our Hybrid Translation Pipeline to produce aligned translations across all 16 scheduled languages. This process enriches the pretraining corpus with high-entropy, culturally grounded tokens. To ensure quality, we developed two LLM-based judges: a Cultural Compliance Judge, which verifies whether the virtual persona's behavior adheres to Indian cultural norms, and a Task Relevance Judge, which evaluates whether the assigned task is appropriate and meaningful for that persona. **Release.** We release a representative subset of PersonaHub, including some personas and the generated data in 16 Indian languages[7]. Refer Appendix. B.5 Workflow Diagram: 8

### 3.3.4 SYNTHETIC AUGMENTATION AND REWRITING PIPELINE

To construct a large and culturally grounded Indic instruction corpus suitable for pretraining and instruction tuning, we develop a unified augmentation pipeline that integrates structured knowledge extraction, persona-driven QA synthesis, controlled rewriting, and grounded template generation. The pipeline transforms raw Indic text into high quality supervision signals while preserving factual fidelity and cultural authenticity. **Structured Knowledge Extraction: Context-Aware Chunking.** Raw Indic documents are segmented into coherent spans of 1000–4000 tokens using a hierarchical chunking algorithm that respects paragraph boundaries, section markers, mathematical blocks, and functional definitions. Each span forms a self-contained semantic unit that supports reliable QA generation without cross-span dependencies. **Relevance and Domain Classification:** Each chunk is filtered to remove noisy or non-Indic-relevant content. A multi-label classifier assigns domain tags such as Healthcare, Finance, History, Culture, Governance, Law, Education, BFSI, News, Sports, or Tourism. Many Indic sources naturally span multiple domains, and this classification retains such multidomain structure while ensuring coverage of culturally specific content often missing from existing datasets. **Structured QA Extraction:** Validated chunks are converted into question-answer pairs covering comprehension, cultural commonsense, causal reasoning, and open-ended analytical prompts. Each question receives two answers: a concise fact-based response and a longer explanatory response that provides additional background. This dual format supports both instruction tuning and reasoning-intensive tasks.
**Persona-Driven QA Diversification:** To increase linguistic variety, social grounding, and conversational realism, we integrate personas from our curated persona bank into the QA construction

---

[7]https://github.com/anonymous-submitter0104/iclr-submission/tree/main/opensource-release/indic-personahub

stage. For a subset of chunks, the question is posed through a selected persona profile, producing variations in tone, intent, curiosity, formality, and reasoning style. The underlying facts always remain grounded in the chunk, but the question framing changes according to the persona. This captures sociolinguistic diversity, enhances pragmatic variation, and enables training models that better understand personality shifts in dialogue. **Synthetic Rewriting and Grounded Generation: Instruction-Style Rewriting:** We apply controlled rewrites of Indic text using prompt-guided transformations that preserve meaning while improving clarity, fluency, and stylistic uniformity across languages. Operations include summarization, elaboration, paraphrasing, style transformation, and ambiguity removal. These rewrites increase lexical diversity and cross-lingual consistency without altering the factual content.**Template-Based Grounded Generation:** Complementing rewriting, we use template-driven grounded generation. A curated template pool covering QA, classification, reasoning, explanation, and paraphrasing is sampled for each chunk. The model is conditioned on the template and the chunk content, enabling synthetic outputs that reflect realistic task formats while remaining strictly grounded in source information. **Resulting Corpus:** The unified pipeline produces a large, culturally aligned, and stylistically rich Indic supervision corpus. Structured QA, persona-driven variation, controlled rewrites, and template-based grounded generation collectively enhance cross-lingual coherence, boost reasoning diversity, and supply high-quality instruction data required for training robust multilingual foundation models. Appendix for Details & Ablations B.6

## 4 INDIC-MMLU: A MULTILINGUAL EVALUATION BENCHMARK

We introduce **Indic-MMLU**, a multilingual adaptation of MMLU (Hendrycks et al., 2020) covering 16 major Indian languages and English. The benchmark enables (i) rigorous evaluation of multilingual LLMs on Indic languages, (ii) measurement of cross-lingual knowledge transfer from English to low-resource scripts, and (iii) evaluate downstream use in rewriting, distillation, and instruction-data generation for Indic LLM development. Current LLM evaluation remains overwhelmingly English-centric, obscuring whether models genuinely reason in Indic languages or rely on hidden translation heuristics. This gap is critical as modern post-training increasingly depends on high-quality Indic data. **Indic-MMLU** provides a standardized, semantically faithful benchmark spanning multiple scripts and typologies, enabling principled assessment of native generalization versus resource-driven collapse arising from limited exposure or tokenization mismatch. Indic-MMLU comprises carefully aligned translations of the English MMLU test set into 16 Indic language. Using our **Translation Pipeline** (Section 3.3.2), MT systems provide first-pass translations which are refined by strong multilingual LLMs for correctness, fluency, and idiomaticity. All domain terms, equations, and answer mappings are preserved. Low-fidelity or ambiguous instances undergo iterative correction to maintain strict semantic equivalence while ensuring linguistic naturalness.

We perform a compact but rigorous multi-stage validation pipeline (with full details in **Appendix C & F**). We employ: a) **LLM Judges:** persona-based *Linguistic*, *Subject-Matter*, and *Cross-Lingual Coherence* experts, each scoring fluency, correctness, and semantic alignment, and b) **Semantic Checks:** embedding-based cosine similarity to enforce intent preservation.

Comparison of different frontier LLMs on Indic MMLU (Appendix C.3), and detailed workflow 9. Indic-MMLU provides a high-quality, semantically consistent multilingual benchmark suitable for evaluating reasoning, linguistic robustness, and cross-lingual transfer in modern LLMs. By decoupling translation from evaluation and enforcing stringent LLM-judge and semantic validation, Indic-MMLU offers a reproducible and equitable foundation for Indian-language evaluation and a scalable substrate for future post-training and data-curation pipelines. Additional details: G C.4

## 5 EXPERIMENTS & ABLATIONS

We evaluate MILA through experiments measuring performance, fairness, and robustness across Indic languages.

### 5.1 MODEL PERFORMANCE AND ANALYSIS: CONTINUAL PRETRAINING ON QWEN3-600M

To isolate the effect of MILA, we take the Qwen3-600M (Q600) pretrained checkpoint (Yang, 2025) and evaluate its baseline performance on Indic-MMLU and other Indian-specific (both language and

culture) benchmarks. We then continually pretrain (CPT) Q600 on $200B$ subset of MILA and re-evaluate, enabling direct attribution of gains to our dataset.

**Experimental Setup.** We maintain identical architecture, optimizer, and training recipe as the original Qwen3, with the sole addition of pretraining on MILA. Evaluation is performed on sixteen Indic

languages using the available Indic benchmarks. From Table 2, it can be observed that CPT on Q600 using MILA has provided consistent performance improvements across different benchmarks like MMLU, MILA, SANSKRITI (examining historical knowledge and philosophy), Belebele, and ARC Challenge. Indic MMLU covers all Indic languages from MILA; however, others may or may not be available in all target 16 languages.

Table 2: Benchmark Performances: Qwen3-600M Base vs. Continual Pretraining on MILA. Performances across different languages are averaged for each benchmark.

| Model | Indic MMLU | MMLU | MILU | Sanskriti | Belebele | ARC Challenge |
|---|---|---|---|---|---|---|
| Q600 Base | 0.3212 | 0.3678 | 0.2751 | 0.4288 | 0.3233 | 0.3123 |
| Q600 CPT | **0.3486** | **0.4195** | **0.2812** | **0.4879** | **0.3563** | **0.3524** |

Even though the CPT was done using a small subset of $200B$, equally distributed across all 17 languages, the performance improvement showcase it usefulness. Also, using a bilingual (English and Hindi) LLM, Param-2.9B, we show the usefullness of our dataset in Appendix B.1.1.

## 5.2 PARITY-BASED ANALYSIS

To measure fairness across languages, we define *parity* as:

$$\text{Parity}_L = \frac{\text{MMLU score in language } L}{\text{MMLU score in English}}. \tag{1}$$

The following table 3 captures how equitably the model performs across Indic languages versus English. An average Indic parity improvement from $0.819 \rightarrow 0.874$, reducing the performance gap with English and improvements observed for all Indic languages highlights MILA's role in promoting balanced multilingual representation.

Table 3: Parity Results comparing Q600 (original) and Q600 (MILA-CPT) across Indic languages.

| Model | As | Bn | Gu | Hi | Kn | Ml | Mr | Ne | Or | Pa | Sa | Sd | Ta | Te | Avg |
|---|---|---|---|---|---|---|---|---|---|---|---|---|---|---|---|
| Q600 (original) | 0.806 | 0.821 | 0.802 | 0.867 | 0.791 | 0.797 | 0.816 | 0.807 | 0.778 | 0.802 | 0.807 | 0.762 | 0.806 | 0.813 | 0.819 |
| Q600 (MILA-CPT) | **0.857** | **0.879** | **0.855** | **0.919** | **0.841** | **0.852** | **0.871** | **0.860** | **0.830** | **0.857** | **0.860** | **0.817** | **0.860** | **0.865** | **0.874** |

## 5.3 DOMAIN-BASED ABLATION EXPERIMENTS

We perform domain-wise ablations on Qwen3-600M CPT, separately for OCR text, synthetic+translated text, and web-crawled text. Due to resource constraints, we performed these three experiments on $56B$ tokens each comprising only 4 languages ($20B$ English and $12B$ each for Hindi, Bengali, and Tamil). Table 4 reveals the relative contribution of each domain to different benchmarks. It can be observed that each source of data contributed equally importantly towards the performance gain.

Table 4: Domain Ablation: Impact on Benchmark Performance

| Benchmark | Q600-OCR | Q600-Synth+Trans | Q600-Crawl |
|---|---|---|---|
| MMLU | 0.4677 | 0.4668 | 0.4570 |
| MILU | 0.2972 | 0.2976 | 0.3018 |
| Sanskriti | 0.5279 | 0.5281 | 0.5706 |
| Belebele | 0.3137 | 0.3074 | 0.2952 |
| ARC Challenge | 0.3456 | 0.3447 | 0.3328 |
| HellaSwag | 0.3888 | 0.3889 | 0.3931 |

## 6 CONCLUSION

In this work, we present **MILA**, a carefully curated Indic dataset to address the scarcity of high-quality training data for low-resource languages. Using Param-2.9B for ablation experiments and Qwen3-600M for final experiments, we show that this dataset not only boosts absolute task performance measured by various benchmarks but also improves parity across languages, as measured by Indic-MMLU. The dataset was constructed with attention to linguistic accuracy, diversity, and coverage across 16 scheduled Indic languages, reflecting the challenges of low-resource research. Our results highlight the central role of curated data in enabling LLMs to perform fairly and robustly across diverse linguistic contexts, complementing advances in model scale and architecture.

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

# Contents

## A  Token Distribution of the MILA Corpus

This section (table: 5) provides a concise breakdown of the 7.5T-token MILA corpus, summarizing each data source—its scale, provenance, and learning contribution. The corpus comprises 4.5T Indian-language tokens and 3T English web-crawl tokens. Representative Open Release.[8].

Table 5: Overview of Token count and semantics of each source in the MILA Corpus. [T: trillions, B: billions]

| Data Source | Token Count | Provenance | Learning Role |
|---|---|---|---|
| Pure Synthetic (Persona-Driven) | 2.5T | In-house persona and distillation generators | Cultural grounding, reasoning, style diversity |
| Translated Knowledge | 1.2T | In-house curated DCLM, FineWeb-Edu translations | World-knowledge transfer to Indic languages |
| Filtered Web Crawl Indian Language Corpus | 410B | Crawled text obtained from filtering Common Crawl (similar to FineWeb-2 methodologies) | |
| Native Literature (OCR) | 200B | OCR from public repositories and partners | Vocabulary depth, formal/archaic structures |
| Wikipedia India | 200B | Indic Wikipedia dumps | High-precision factual and entity knowledge |
| English | 3T | In-house curated DCLM, FineWeb-Edu translations | Source for World-knowledge |

---

[8]https://github.com/anonymous-submitter0104/iclr-submission/tree/main/opensource-release

## B  EXPERIMENTS AND ABLATIONS

### B.1  DATA CURATION

Refer Supplementary Repository [9]

### B.1.1  ABLATION STUDY: TASK PERFORMANCE IMPROVEMENTS

To validate the effectiveness of our curation pipeline, we conduct two complementary ablation experiments that isolate the impact of data quality on model performance and safety. The first experiment directly compares models trained on conventional versus curated data under otherwise identical conditions. We continually pretrain two instances of Param-1 (Pundalik et al., 2025), a 2.9 billion parameter causal language model, on 2 trillion tokens of English and Hindi data: one using raw web-scraped text processed only with basic cleaning, and another using the fully curated pipeline described above. The model architecture, detailed in Table 6, employs grouped-query attention with 32 hidden layers, a hidden dimension of 2048, an intermediate dimension of 7168, and fast-swiglu activation functions. All hyperparameters, training duration, batch size, learning rate schedule, and computational infrastructure remain strictly identical between the two experiments, ensuring that any performance differences arise solely from data quality rather than confounding factors.

| Architecture attributes | Values |
|---|---|
| Model Architecture | causal-language-model |
| Hidden size | 2048 |
| Intermediate size | 7168 |
| Max Position Embeddings | 2048 |
| Num of Attention Heads | 16 |
| Rope theta | 10000 |
| Num of Hidden Layers | 32 |
| Num of Key Value Heads | 8 |
| Activation Function | fast-swiglu |
| Attention Type | Grouped-query attention |
| Precision | bf16-mixed |

Table 6: Architecture Details of PARAM-1

The results, presented in Table 7, demonstrate substantial and consistent improvements across all evaluated benchmarks. On ARC Challenge, the curated model achieves 53.6% accuracy compared to 46.5% for the conventional baseline, representing a 7.1 percentage point gain on this challenging reasoning benchmark. ARC Easy shows a more modest but still meaningful improvement from 73.6% to 74.2%. HellaSwag performance remains stable at approximately 73.5-73.8% for English, but the Hindi variant reveals dramatic gains: the curated model achieves 41.4% accuracy versus only 28.9% for the conventional baseline, a 12.5 percentage point improvement that highlights the particular value of curation for low-resource languages where noisy training data disproportionately degrades performance. MMLU results follow a similar pattern, with the curated model reaching 46.2% on English MMLU compared to 41.3% for the baseline, and 34.6% on Hindi MMLU versus 26.2%, an 8.4 percentage point improvement that again underscores curation's amplified benefits for Indic languages. These results provide compelling evidence that systematic data curation translates directly into stronger downstream task performance, with particularly pronounced effects in multilingual settings where script variation, code-mixing, and sparse high-quality resources make conventional cleaning insufficient.

Table 7: Benchmark Results: Conventional vs Curated.

| Model | ARC Challenge | ARC Easy | Hella Swag | Hella Swag Hi | MMLU | MMLU Hi |
|---|---|---|---|---|---|---|
| Conventional | 46.5 | 73.6 | 73.5 | 28.9 | 41.3 | 26.2 |
| Curated | **53.6** | **74.2** | **73.8** | **41.4** | **46.2** | **34.6** |

---

[9]https://github.com/anonymous-submitter0104/iclr-submission/tree/main/data-curation

### B.1.2 ABLATION STUDY: SAFETY AND TOXICITY REDUCTION

Beyond task-specific performance, our second ablation experiment investigates whether curation improves model safety, a critical consideration for deployment in sensitive linguistic and cultural contexts. We evaluate toxicity using the Toxigen benchmark via LLM360's Safety360 suite[10], which provides both explicit and subtle adversarial prompts spanning identity-based categories (race, religion, gender, nationality) and general offensive content. The evaluation protocol generates model completions from curated Toxigen templates, then classifies outputs using a RoBERTa-based detector fine-tuned for nuanced and context-dependent toxic language detection. This methodology captures not only overt hate speech but also subtle stereotype amplification, coded language, and identity-based microaggressions that simpler keyword-based detectors miss. We compare our curated PARAM-1 (Pundalik et al., 2025) model against three multilingual baselines: SARVAM-1[11], LLaMA3.2-9T-3B (Touvron et al., 2023a;b; Grattafiori et al., 2024), and Gemma2-2T-2B (Team et al., 2024a;b; 2025), all of which represent state-of-the-art multilingual language models trained on substantial corpora but without our specialized pipeline.

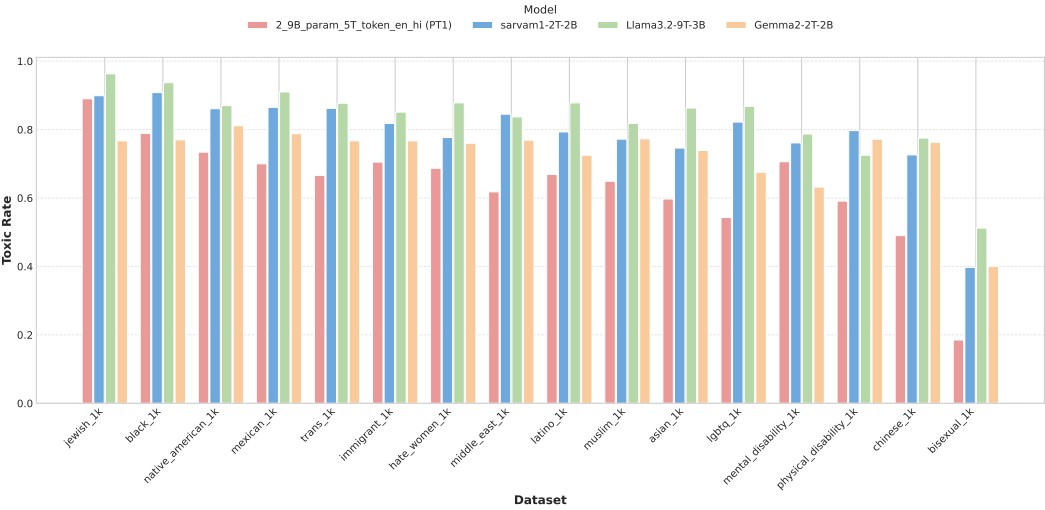

Figure 3: Toxicity Comparison

Figure 3 visualizes toxicity rates across 16 datasets encompassing both neutral baseline prompts and adversarial examples designed to elicit harmful outputs. The curated PARAM-1 model consistently maintains lower toxicity rates than all three baselines across nearly every evaluation condition. On adversarial identity-based prompts, the most challenging category where models are explicitly prompted to generate stereotypical or prejudiced content,PARAM-1 demonstrates particularly strong resistance, producing toxic outputs at rates 15-30% lower than comparable models. Critically, this improved safety profile does not come at the cost of over-censorship or reduced utility: the model continues to generate fluent, contextually appropriate responses to neutral prompts while declining to amplify harmful stereotypes or engage with bad-faith adversarial framing. These results establish that our two-stage toxic filtering process, combining rule-based initial flagging with multilingual RoBERTa-based reclassification, effectively removes training examples that would otherwise teach models to reproduce harmful patterns, without introducing excessive false positives that would degrade linguistic coverage or cultural representativeness. Taken together, these ablation experiments provide robust evidence that our curation pipeline delivers dual benefits: substantial improvements in task-specific accuracy and reasoning capabilities, alongside meaningfully safer generation behavior that avoids stereotype amplification and identity-based harm. The performance gains on Hindi benchmarks and the reduced toxicity rates in multilingual contexts are particularly noteworthy, demonstrating that careful attention to data quality yields compounding returns for low-resource languages where existing models struggle most. By combining language-specific quality classification, synthetic rewriting for medium-quality content, aggressive deduplication, nuanced toxic con-

---

[10]https://github.com/LLM360/Analysis360
[11]https://www.sarvam.ai/

Table 8: Contamination % of each benchmark across different sources of our dataset. We use InfiniGram to check the contamination % using n-gram overlaps. Apart from MMLU, *no other benchmarks are observed to be contaminated* to the **MILA** corpus.

| Benchmark | OCR | Translation | Synthetic | Crawl |
|---|---|---|---|---|
| Indic MMLU | 0.00% | 0.00% | 0.00% | 0.00% |
| MMLU | 0.00% | 0.00% | 0.00% | 0.75% |
| IndicGenBench | 0.00% | 0.00% | 0.00% | 0.00% |
| MILU | 0.00% | 0.00% | 0.00% | 0.05% |
| Sanskriti | 0.00% | 0.00% | 0.00% | 0.00% |
| HellaSwag | 0.00% | 0.00% | 0.00% | 0.00% |
| HellaSwag-Hi | 0.00% | 0.00% | 0.00% | 0.00% |
| ARC-Challenge | 0.00% | 0.00% | 0.00% | 0.01% |
| ARC-Challenge-Hi | 0.00% | 0.00% | 0.00% | 0.00% |
| SQuAD | 0.00% | 0.00% | 0.00% | 0.00% |
| SQuAD-Hi | 0.00% | 0.00% | 0.00% | 0.00% |
| Belebele | 0.00% | 0.00% | 0.00% | 0.00% |

tent filtering, and comprehensive PII redaction, we establish a strong foundation for responsible deployment of large language models in multilingual and culturally diverse settings. This curation infrastructure not only enhances MILA's capabilities but provides reusable patterns for future corpus construction efforts aimed at equitable language technology development across the world's linguistic diversity.

### B.1.3 DOWNSTREAM BENCHMARK DECONTAMINATION

We perform comprehensive benchmark and task decontamination across all four of our major data sources, **OCR**, **Crawl**, **Translation**, and **Synthetic**, to rigorously prevent evaluation leakage into pretraining. Our decontamination pipeline integrates multiple complementary strategies to maximize coverage and robustness.

First, we apply **n-gram–based overlap filtering** (8–13 grams), following the methodology provided in NVIDIA's NeMo Task Decontamination toolkit.[12] This step eliminates any document fragments that exhibit high lexical overlap with known benchmark content.

Next, we extend the filtering process using **InfiniGram**[13], which enables cross-domain, long-context retrieval for detecting semantically similar spans that may not share explicit surface-level overlap. This allows us to identify subtle or paraphrased contaminations that traditional n-gram approaches may fail to detect.

Finally, beyond English datasets, we also conduct **multilingual benchmark decontamination**, covering both (i) *native multilingual benchmarks* and (ii) *translated versions of widely used English benchmarks*. This ensures that no part of our multilingual pretraining corpus inadvertently memorizes evaluation sets in any of the 16 Indic languages.

Together, these steps constitute a high-precision, multi-layered decontamination pipeline designed to preserve the integrity and fairness of downstream evaluations. Refer table for Decontamination Stats 8

### B.2 PII IDENTIFICATION AND REDACTION

**PII Identification and Redaction.** We apply comprehensive multilingual PII detection and removal using NVIDIA NeMo's Data Curation Toolkit,[14] which supports both rule-based and model-driven

---

[12]https://docs.nvidia.com/nemo-framework/user-guide/25.04/datacuration/taskdecontamination.html
[13]https://infini-gram-mini.io/
[14]https://docs.nvidia.com/nemo-framework/user-guide/25.07/datacuration/personalidentifiableinformationidentificationandremoval.html

identification. The pipeline detects names, contact details, addresses, financial identifiers, and other sensitive spans across English and all 16 Indic languages. Detected PII is consistently redacted or masked to ensure privacy safety across OCR, crawl, translation, and synthetic corpora. This guarantees that no personally identifiable information leaks into the pretraining dataset.

## B.3 OCR Pipeline

### B.3.1 ISOB-Small: A Synthetic In-House Benchmark for Indic OCR

To rigorously evaluate OCR quality and guide pipeline improvements, we developed the Indic Synthetic OCR Benchmark (ISOB-Small), a controlled testbed spanning 16 Indian languages across 110 synthetically generated pages. Direct evaluation on real scanned documents proved infeasible due to copyright restrictions on the archival materials obtained through formal memoranda of understanding with institutional partners. Rather than compromising evaluation rigor, we designed ISOB-Small to systematically reproduce the challenges observed in real-world digitization through synthetic generation. Each page in ISOB-Small incorporates realistic degradations encountered during document processing: multi-column layouts that test layout analysis, dense tables and figures that challenge region segmentation, mathematical expressions with mixed scripts, watermarks and stamps that introduce visual noise, paper folds and shadows that create uneven illumination, font variations that stress glyph recognition, and controlled blur levels that simulate aging or poor scan quality. By programmatically generating these artifacts from clean ground truth text, ISOB-Small provides a copyright-compliant evaluation framework while exposing the specific failure modes that generic OCR systems exhibit on Indic scripts. Representative examples from the benchmark are shown in Figure 4, illustrating the diversity of layouts, scripts, and degradations included in the test set. Full ISOB Construction Pipeline Workflow: 5

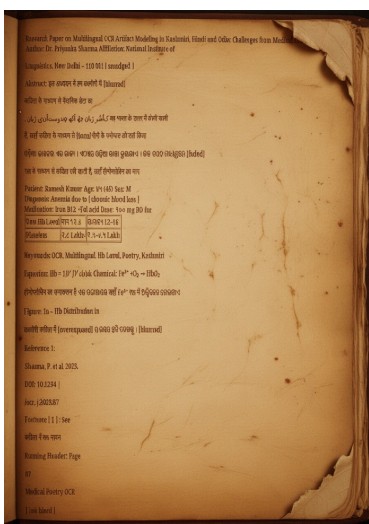 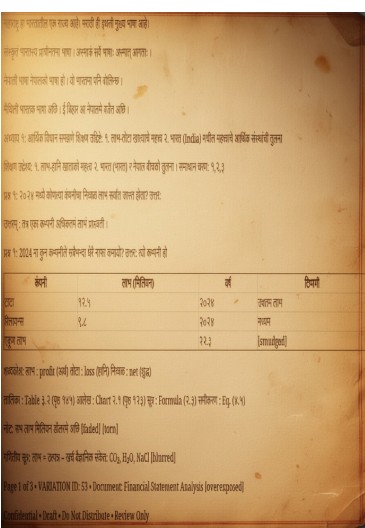

Figure 4: Samples from ISOB Benchmark

The benchmark creation pipeline begins with seed corpus selection from existing OCR'd pages in hOCR format, filters for high-difficulty documents using confidence scores and VLM-based complexity prediction, randomly selects 3-10 target languages, extracts a taxonomy of hard artifacts from real documents, augments the selected pages with these artifacts while translating text into target languages, renders the enriched hOCR into visual form, applies style transformations using image editing models prompted with Indian manuscript aesthetics, and finally introduces low-level image augmentations including orientation changes, contrast shifts, noise injection, and geometric distortions. This systematic generation process ensures comprehensive coverage of OCR challenges while remaining fully reproducible and extensible to additional languages or artifact types.

Beyond immediate corpus construction, ISOB-Small represents a foundational contribution to the broader research community working on low-resource language digitization. Recognizing that legal

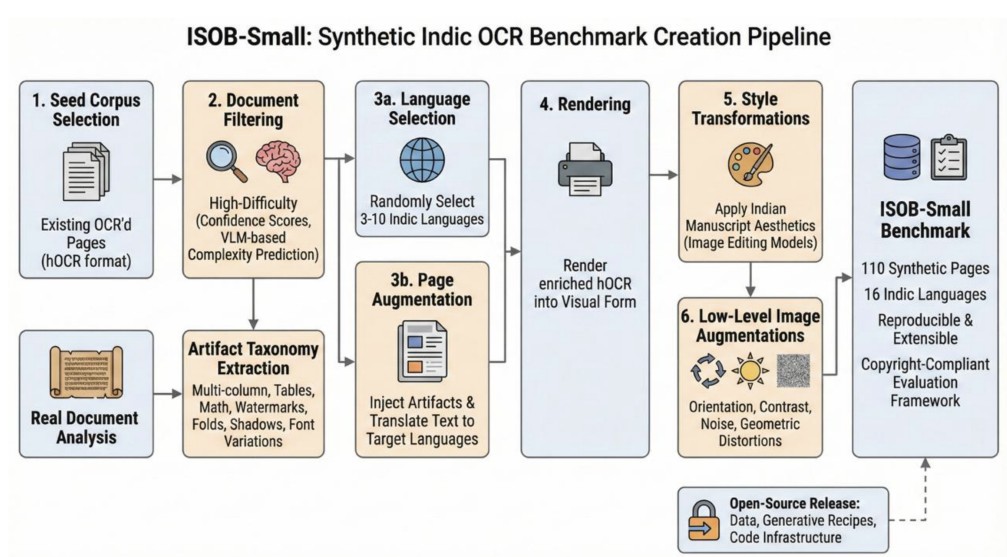

Figure 5: End-to-End Construction flow of ISOB Pipeline

and ethical constraints prevent many researchers from accessing real archival materials for evaluation, we are releasing not only the benchmark dataset itself but also the complete generative recipes and code infrastructure used to construct it. This enables researchers to extend ISOB-Small to additional languages, generate larger test sets with customized difficulty profiles, or create domain-specific variants targeting particular document types such as legal texts, scientific literature, or historical manuscripts. The synthetic generation approach circumvents copyright barriers while providing controlled evaluation that isolates specific OCR challenges, a methodology applicable far beyond the Indic language context to any script or language lacking adequate digitization benchmarks. By open-sourcing both data and methodology, we aim to accelerate progress on OCR for underserved writing systems worldwide, fostering reproducible experimentation and enabling fair comparison across OCR systems and postprocessing techniques.

### B.3.2 COMPARATIVE EVALUATION AND POSTPROCESSING IMPACT

Evaluation on ISOB-Small revealed stark performance disparities across OCR systems and exposed critical weaknesses in vision-language models applied to Indic text recognition. Table 9 presents comprehensive results on both existing Indian OCR benchmarks (Bhashini) and the synthetic ISOB-Small testbed, tracking character error rate (CER), word error rate (WER), position-independent word error rate (PI-WER), and character-level 3-gram F1 scores. Specialized OCR models such as DotsOCR and Surya achieve substantially lower error rates on Bhashini (CER of 0.168 and 0.2 respectively) compared to general-purpose vision-language models like Qwen2.5-VL-72B (Bai et al., 2023; Yang et al., 2024; Qwen et al., 2025; Yang, 2025) (CER of 0.676) and Llama-4-Scout (Touvron et al., 2023a;b; Grattafiori et al., 2024) (CER of 0.259). While VLMs offer broader task coverage and can handle diverse document types without specialized training, they suffer from hallucination on Indic scripts—generating plausible-looking but semantically incorrect text that is difficult to detect through automated metrics alone. In contrast, traditional OCR models produce more predictable error patterns that can be systematically addressed through postprocessing rules and language model correction. The performance gap widens dramatically on ISOB-Small, where VLMs struggle with the synthetic complexity: models like pixtral-12B and GLM-4.1V-9B-Thinking exhibit CER exceeding 4.0, while SmolDocling and InternVL variants fail catastrophically with error rates above 38. These results validate ISOB-Small as a genuine stress test that exposes brittleness in systems that perform adequately on cleaner benchmarks.

The critical importance of postprocessing becomes evident when examining performance improvements after applying our language-specialized correction pipeline. Table 10 demonstrates that targeted enhancements—including dictionary-based correction, language model rescoring, ligature re-

Table 9: Model Performance Benchmarks for Different Pipelines

| List of Models | Bhashini | | | | Mozhi | | | |
|---|---|---|---|---|---|---|---|---|
| | CER | WER | PI-WER | Char3 F1 | CER | WER | PI-WER | Char3 F1 |
| DotsOCR | 0.168 | 0.253 | 0.23 | 0.801 | 0.12 | 0.19 | 0.9 | 0.88 |
| Surya | 0.2 | 0.28 | 0.138 | 0.867 | 0.14 | 0.21 | 0.91 | 0.89 |
| Llama-4-Scout-17B-16E-Instruct | 0.259 | 0.445 | 0.398 | 0.672 | 4.35 | 1.38 | 0.619 | 0.31 |
| NuMarkdown-8B-Thinking | 0.361 | 0.537 | 0.508 | 0.556 | 53.31 | 9.21 | 0.677 | 0.168 |
| Llama-4-Maverick-17B-128E-Instruct | 0.4 | 0.58 | 0.418 | 0.645 | 12 | 4 | 0.72 | 0.22 |
| Qwen2.5-VL-72B-Instruct | 0.676 | 0.847 | 0.45 | 0.613 | 18.22 | 4.16 | 0.677 | 0.266 |
| SmolDocling-256M-preview | 1.235 | 1.4 | 0.988 | 0.016 | 137.66 | 55.57 | 0.946 | 0.0001 |
| RolmOCR | 1.938 | 2.019 | 0.498 | 0.552 | 986.91 | 263.08 | 0.692 | 0.111 |
| olmOCR-7B-0825 | 2.068 | 1.842 | 0.516 | 0.531 | 28.53 | 6.48 | 0.704 | 0.126 |
| Nanonets-OCR-s | 3.573 | 2.318 | 0.568 | 0.471 | 305.27 | 42.57 | 0.685 | 0.161 |
| GLM-4.1V-9B-Thinking | 4.384 | 3.88 | 0.893 | 0.08 | 755.1 | 321.92 | 0.985 | 0.0001 |
| MinerU2.5-2509-1.2B | 5.176 | 3.214 | 0.906 | 0.095 | 180 | 55 | 0.91 | 0.1 |
| pixtral-12B | 5.847 | 4.86 | 0.893 | 0.163 | 1.47 | 0.999 | 0.941 | 0.0039 |
| InternVL3_5-GPT-OSS-20B-A4B-Preview-HF | 38.87 | 4.537 | 0.994 | 0.0029 | 195.85 | 240.2 | 0.919 | 0 |

pair, and Unicode normalization—substantially reduce error rates even for already-strong baselines. DotsOCR improves from 0.168 to 0.085 CER on Bhashini after postcorrection, while Surya advances from 0.2 to 0.095 CER. These gains translate directly to corpus quality: reducing CER by half means doubling the amount of usable training data extracted from each scanned page. On ISOB-Small, postcorrected models achieve scores above 0.86, confirming that the combination of specialized OCR with targeted postprocessing provides a robust solution for Indic digitization. These targeted enhancements proved critical not only for benchmark performance but for producing high-quality, machine-readable text suitable for downstream language model training.

Table 10: Model Performance on Benchmarks

Preprocessing / Post-Correction Performance Bhashini

| List of Models | CER | WER | PI-WER | Char3 F1 |
|---|---|---|---|---|
| Dots.OCR - postcorrected | 0.085 | 0.145 | 0.12 | 0.91 |
| Surya - postcorrected | 0.095 | 0.16 | 0.11 | 0.925 |

ISOB-Small Results

| List of Models | ISOB-Small |
|---|---|
| Dots.OCR | 0.8616 |
| Surya | 0.8982 |

### B.3.3 LLM-ASSISTED QUALITY EVALUATION

To validate that improvements in traditional metrics (CER, WER) actually translate to better semantic quality, we designed a comprehensive evaluation framework leveraging large language models as quality judges. Conventional OCR metrics measure surface-level string similarity but fail to capture whether postprocessing interventions—such as sentence reordering for improved coherence, ligature repair that changes character sequences, or contextual corrections that substitute semantically equivalent terms—preserve or enhance meaning. This limitation is particularly acute for quality-enhanced text where deliberate modifications may increase string distance from raw OCR while improving linguistic fidelity. Our LLM-assisted evaluation addresses this gap through multi-stage assessment using state-of-the-art models including GPT-OSS-120B (OpenAI et al., 2025), Deepseek (DeepSeek-AI et al., 2025), and Qwen (Bai et al., 2023; Yang et al., 2024; Qwen et al., 2025; Yang, 2025). Each model is prompted to compare original ground truth with both raw OCR and postprocessed outputs, providing consistency scores that reflect semantic preservation independent of surface form.

### B.3.4 ABLATION STUDY: CONVENTIONAL VS PROCESSED OCR DATA

The practical impact of **OCR quality** on downstream model training is demonstrated through controlled experiments using a 310M parameter dense language model (refer to Table 11 for model configs). The training scripts and reproducibility details are provided on github[15]. We compared training dynamics on two versions of the same corpus: raw OCR output with typical error rates around 15-20 percent, and quality-enhanced text after applying our full postprocessing pipeline. Pretraining on raw OCR data produced highly unstable loss curves with frequent spikes, irregular

---

[15]https://github.com/anonymous-submitter0104/iclr-submission/tree/main/ocr-pipeline

perplexity behavior, and slow convergence, clear indicators of training corpus noise overwhelming the learning signal. Models trained on this noisy data struggled to learn coherent linguistic patterns, exhibiting high validation perplexity and poor performance on downstream tasks. In stark contrast, training on postcorrected text yielded smooth, monotonic loss reduction and stable perplexity curves characteristic of high-quality pretraining data, as visualized in Figure 6.

| Architecture attributes | Values |
|---|---|
| Model Architecture | causal language model |
| Hidden size | 768 |
| Intermediate size (FFN) | 3108 |
| Max Position Embeddings | 2048 |
| Num of Attention Heads | 12 |
| Num of Hidden Layers | 12 |
| Num of Query Groups | 12 |
| Normalization | RMSNorm |
| Activation Function | swiglu |
| Attention Type | Multi-head Attention |
| Position Embedding Type | RoPE (rotary) |
| Dropout (hidden/attn/ffn) | 0.0 / 0.0 / 0.0 |
| Precision | bf16 (AMP O2) |

Table 11: Architecture Details of 310M Parameter Model

The quality-enhanced model achieved markedly lower perplexity and superior Indic benchmark performance, illustrating that careful preprocessing directly elevates model capability. In low-resource settings, noisy OCR exacerbates data scarcity, whereas robust OCR and postprocessing transform limited corpora into high-impact training material. Beyond MILA, our infrastructure and validation framework provide reusable tools for equitable multilingual model development.

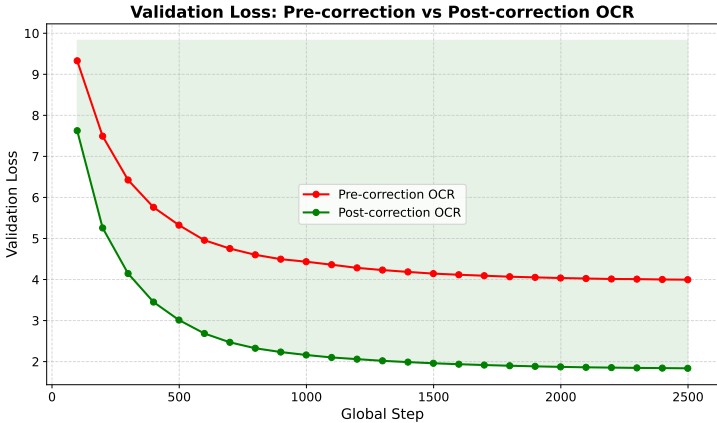

Figure 6: Validation loss comparison between models trained on conventional OCR output versus postcorrected text. The postcorrected corpus produces substantially more stable training dynamics and lower final perplexity, demonstrating the direct impact of OCR quality on model performance.

### B.4 TRANSLATION PIPELINE

Figure 7 shows End 2 End Translation Pipeline as described in the Main Paper Section. A fundamental challenge in constructing large-scale Indic datasets lies not merely in the absence of digital text, but in the structural scarcity of high-quality monolingual and parallel corpora that could enable cross-lingual transfer from resource-rich languages. Translation-based data generation directly addresses this bottleneck by producing parallel corpora that facilitate knowledge transfer from languages with abundant digital resources, primarily English, into the 16 Indic languages that form the backbone of MILA. This capability proves critical for multilingual model development, as re-

cent work has demonstrated that downstream performance on mathematical reasoning, STEM domains, and code generation benefits substantially from exposure to parallel corpora during pretraining (Chen et al., 2023; Lin et al., 2024). The mechanism underlying these gains appears to be the model's ability to align semantic representations across languages, learning that mathematical operations, logical reasoning patterns, and algorithmic structures transcend linguistic boundaries. By providing models with paired examples that express identical concepts in multiple languages, parallel corpora enable the extraction of language-invariant knowledge structures that generalize more robustly than monolingual training alone.

Figure 7: End-to-end view of our hybrid MT→LLM translation pipeline integrating specialist MT grounding, contextual LLM refinement, and reasoning-enabled correction.

### B.4.1 THE SPECIALIST-GENERALIST TENSION IN LOW-RESOURCE TRANSLATION

The production of high-quality translations for low-resource Indic languages presents challenges that extend far beyond simply applying off-the-shelf systems. As demonstrated in Table 12, no single model achieves universally optimal performance across the 15 Indic languages we target. Dedicated machine translation systems such as IndicTrans2 (IT2) perform strongly across a broad range of languages, particularly when combined with preprocessing and postprocessing pipelines, where they achieve consistent improvements over the raw IT2 outputs (e.g., +2–3 chrF++ on average). Nevertheless, IT2 still struggles with truly low-resource languages such as Maithili and Sanskrit, where limited parallel training data constrains model effectiveness (27.70–30.24 chrF++ on Sanskrit). By contrast, general-purpose multilingual models such as NLLB-200-3.3B (Team et al., 2022), NLLB-moe-54B (Team et al., 2022), DeepSeek V3.1 Think (DeepSeek-AI et al., 2025), and Llama-4-Maverick-17B (Touvron et al., 2023a;b; Grattafiori et al., 2024) provide broader linguistic coverage without explicit fine-tuning. These generalist systems achieve competitive scores on well-resourced languages (e.g., NLLB-moe-54B reaches 57.03 chrF++ on Kannada; Llama-4-Maverick 57.34 on Telugu), but their performance remains uneven on low-resource settings, with noticeable drops on Sanskrit and Maithili. Moreover, their outputs often exhibit subtle semantic drift, morphological inconsistencies, or code-mixing with English or Hindi, which undermines corpus quality. Alternative approaches such as Hunyuan-MT demonstrate catastrophic failure modes, producing near-zero scores on Nepali, Oriya, and Punjabi, underscoring the fragility of systems lacking robust multilingual grounding.

This fundamental tension between specialist precision and generalist coverage motivates our carefully architectured translation pipeline, as shown in Figure 7, which eschews reliance on any single system in favor of an ensemble approach that combines specialist and generalist models through multi-stage processing. Our pipeline begins with synthetic augmentation, translating content from English and other resource-rich languages into the 16 target Indic languages using an ensemble of specialist machine translation models and generalist large language models. Rather than treating translation as a one-pass operation, we implement a robust LLM-based post-correction phase that repairs syntactic and semantic inconsistencies introduced during initial translation, enhances con-

text preservation across sentence boundaries, and addresses the subtle morphological and syntactic variations that distinguish natural Indic language use from mechanically translated text. This post-correction stage leverages state-of-the-art multilingual language models, specifically those demonstrating strong performance on benchmarks as shown in Table 12 (b), to ensure that postprocessing genuinely improves linguistic quality rather than introducing new errors through models with insufficient Indic language proficiency.

| List of Models | Translation Benchmarks | | | | | | | | | | | | | |
|---|---|---|---|---|---|---|---|---|---|---|---|---|---|---|
| | As | Be | Gu | Ka | Hi | Mai | Ml | Mr | Ne | Or | Pa | Sa | Sd | Ta | Te |
| NLLB-200-3.3B | nan | 48.58 | 52.10 | 56.87 | 52.67 | 44.08 | 49.35 | 47.03 | 45.93 | 46.05 | 49.48 | 25.28 | 52.49 | 54.37 | 48.24 |
| NLLB-moe-54B | nan | 49.86 | 53.30 | 57.03 | 53.08 | 46.63 | 51.47 | 47.85 | 45.10 | 45.34 | 48.75 | 25.56 | 53.46 | 55.72 | 48.71 |
| hunyuan-mt | nan | 42.22 | 41.38 | 46.62 | 45.01 | 8.40 | 1.91 | 43.02 | 0.83 | 0.95 | 1.04 | 18.80 | 42.94 | 40.03 | 39.75 |
| deepseek v3.1 Think | 39.10 | 44.30 | 47.95 | 53.12 | 47.56 | 39.59 | 46.86 | 44.80 | 47.23 | 44.34 | 46.80 | 25.37 | 48.90 | 48.68 | 46.05 |
| Llama-4-Maverick-17B | 40.35 | 47.09 | 48.71 | 53.21 | 47.57 | 42.03 | 44.88 | 46.64 | 44.65 | 39.34 | 45.88 | 28.13 | 48.05 | 47.42 | 57.34 |

(a) Translation Model Benchmarks Without Processing

| List of Models | Translation Benchmarks | | | | | | | | | | | | | |
|---|---|---|---|---|---|---|---|---|---|---|---|---|---|---|---|
| | As | Be | Gu | Ka | Hi | Mai | Ml | Mr | Ne | Or | Pa | Sa | Sd | Ta | Te |
| IT2 | 45.10 | 48.67 | 53.38 | 55.62 | 52.26 | 47.04 | 52.23 | 49.33 | 53.42 | 50.47 | 49.82 | 27.70 | 53.03 | 54.77 | 49.19 |
| IT2 Processed | 48.40 | 51.71 | 55.53 | 58.71 | 54.97 | 49.49 | 55.99 | 51.00 | 56.01 | 52.28 | 51.08 | 30.24 | 56.86 | 58.65 | 50.88 |

(b) IndicTrans2 as a Candidate Specialist MT Model and its Corresponding Postprocessing

Table 12: Translation Model Performance on FLORES Benchmarks using chrF++

### B.4.2 LLM-BASED POST-CORRECTION AND HUMAN VALIDATION

The impact of this post-correction architecture becomes evident in Table 12 (b), where our processed IndicTrans2 (IT2) pipeline achieves substantial improvements over baseline translation quality. Post-correction elevates performance from 45.10 to 48.40 chrF++ for Assamese, from 48.67 to 51.71 for Bengali, and from 52.23 to 55.99 for Malayalam, demonstrating consistent gains across the entire language spectrum. More critically, the pipeline shows its greatest impact precisely on the low-resource languages where initial translation quality is weakest: Sanskrit improves from 27.70 to 30.24 chrF++, while Sindhi advances from 53.03 to 56.86. These gains translate directly to corpus utility, improving translation quality by 5-7 percent effectively increases the amount of usable training data by similar margins, as higher-fidelity translations reduce the noise that would otherwise corrupt the learning signal during language model pretraining. The consistent improvements across all 15 languages validate our hypothesis that ensemble translation followed by targeted LLM-based correction provides more robust quality than relying on any single translation system, however sophisticated.

Human evaluation plays a critical role at multiple stages of this pipeline, providing ground truth validation that guides both model selection and quality assessment. We integrated three expert language evaluators for each target language, who reviewed initial translation outputs to identify systematic error patterns, evaluate cultural appropriateness of linguistic choices, and ensure that translations reflect natural language use rather than the stilted, overly literal renderings characteristic of naive machine translation. This human-in-the-loop approach proved particularly valuable for detecting subtle issues invisible to automated metrics: gender agreement errors in morphologically rich languages like Hindi and Bengali, inappropriate register choices that clash with the formality level of source content, and culturally insensitive translations that preserve denotative meaning while losing connotative appropriateness. The evaluators' feedback directly informed our model selection process, leading us to preferentially weight outputs from models that demonstrated stronger alignment with natural Indic language patterns as judged by native speakers. This validation framework ensures that our pipeline produces not merely linguistically correct translations, but culturally aligned representations that will support language models in learning authentic Indic language use rather than absorbing artifacts of mechanical translation. We have showcased a study on readability of translation experiments in Section F.

For the post-correction phase, we first identify the most suitable large language models (LLMs) on a per-language basis by consulting the Indic MMLU benchmark results (Table 16). This ensures that the models chosen for grammatical refinement are not only capable in general reasoning but also demonstrate strong competence in the specific Indic language of interest. For instance, DeepSeek V3.1 (DeepSeek-AI et al., 2025) was selected for Hindi owing to its consistently strong Indic MMLU scores, while Gemma-3 27B (Team et al., 2024a;b; 2025) was preferred for Tamil due to its comparatively better alignment on Dravidian languages. Once selected, these models are

guided through carefully designed prompts that frame post-correction as an expert linguistic transformation task, positioning the model as a specialist in the target language with deep understanding of grammar, syntax, and natural conventions. The prompt explicitly directs the model to convert poorly structured, grammatically incorrect, or awkward translations into natural, well-formed text while adhering to several critical constraints: strict semantic preservation with no omissions, exclusive use of the target language with no English or Hindi code-mixing, grammatical accuracy with natural flow, and complete avoidance of meta-commentary that would contaminate the corpus. This design reflects lessons learned from extensive experimentation: early prompt versions that omitted explicit prohibitions against code-mixing often produced outputs interspersed with English terms, while lack of explicit length guidance led to truncated or verbose outputs that distorted information density. The refined prompt achieves a careful balance, enabling models to repair genuine errors while preserving the integrity and content of the original translation.

---

**Prompt Template for Post-Correction**

**Role and Context:** You are an expert linguist specializing in the {language} language with deep understanding of grammar, syntax, and natural language use.
**Task:** Transform {language} text that is poorly structured, grammatically incorrect, awkwardly translated, or unnatural into well-formed, grammatically correct, and natural-sounding {language} text.
**Input Text:** '{input_text}'
**Output Requirements:**

- Return complete rephrased text with no omissions wherever needed

- Never return empty responses

- Maintain original language with no English/Hindi mixing

- Focus on grammatical correctness and natural flow

- Do not provide explanations, notes, or meta-commentary

- Keep the length close to the original text

---

After post corrections, the improvements span multiple dimensions of linguistic quality: grammatical structures are regularized to match target language conventions, sentence flow is enhanced through appropriate use of discourse markers and transitional phrases, and awkward word choices are replaced with more natural alternatives that better capture the intended meaning. Critically, these transformations preserve semantic fidelity, the corrected text expresses the same propositional content as the original translation while rendering it in more fluent, idiomatic form. This balance between correction and preservation proves essential for corpus quality: overly conservative post-correction leaves errors intact, while overly aggressive rewriting risks introducing semantic drift that corrupts the training signal. Our prompt-guided approach navigates this tension by giving models clear objectives (improve naturalness, correct grammar) alongside explicit constraints (preserve meaning, maintain length), enabling reliable quality enhancement without the semantic instability that plagued earlier correction attempts. Additional Detailed Baselining and Benchmarking is given here: H

### B.5    INDIC PERSONAHUB: ENGINEERING CULTURAL IDENTITY AT SCALE

To operationalize culturally grounded distillation, we developed Indic PersonaHub, a large-scale repository of over 300 million Indian personas spanning 1400+ domains. The motivation behind PersonaHub lies in a fundamental challenge: most foundation models, trained on predominantly Western-centric corpora, default to perspectives, reasoning patterns, and cultural assumptions that misalign with Indian discourse. Without intervention, even sophisticated downstream adaptations risk reproducing these biases, thereby diluting authenticity in tasks that require contextually grounded knowledge. PersonaHub addresses this gap by serving as a structured mechanism to embed Indian perspectives directly into synthetic training data.

Unlike demographic templates or shallow role labels, a persona in our framework is conceived as a multidimensional specification that guides language models toward authentic and culturally resonant responses. Inspired by billion-scale persona generation efforts (Ge et al., 2025), our de-

sign extends beyond age, gender, or occupation to encode rich contextual detail: linguistic preferences, cultural traditions, domain expertise, regional affiliations, and value systems. For example, an Ayurvedic practitioner from Kerala approaches medical reasoning through centuries, old holistic frameworks, while an AIIMS, trained allopathic physician employs biomedical evidence and global clinical guidelines. Similarly, a Tamil literary scholar interprets texts through Sangam literature rather than Western critical theories. Such grounding ensures that personas operate not as abstract demographic placeholders but as situated voices representing diverse Indian knowledge systems.

---

**Sample Indian Virtual Persona + Task (associated triggering Question)**

You are a patriotic Indian poultry farmer deeply committed to the nation's agricultural self-reliance and food security. They are vigilant about potential health risks and economic impacts of poultry diseases, particularly in the context of India's thriving backyard poultry sector. As a proud member of the Indian Poultry Farmers Association, they actively promote indigenous poultry breeds and sustainable farming practices. They are well-versed in disease management protocols and quarantine measures, recognizing their importance in safeguarding India's poultry industry. Their concerns extend to protecting commercial poultry farms, which are vital to India's rural economy and protein supply. They view their work as a service to the nation, aligning with the government's initiatives to boost agricultural productivity and rural livelihoods. Your role is to engage with users based on your expertise. Stay within your domain and maintain the persona's tone and expertise.

# CONTEXT # The need for this dataset stems from the desire to uphold a standard of excellence in English language content within the field of Poultry Farming and Livestock Management in India. By compiling a diverse range of well-structured and authentic texts, this collection will help maintain a rich linguistic resource that supports clarity, readability, and contextual accuracy in various forms of communication.

# OBJECTIVE # I want you to generate text paragraphs strictly in English language with 900+ words for: Poultry Farming and Livestock Management in India that is easy to read, flows naturally, and sounds like it was written by a human. Generated text data should mimic real world data so that it can be also used to improve research and innovation. Use clear transitions between sentences and paragraphs while maintaining a consistent narrative or argument ensuring a logical progression of thought. Ensure the writing is engaging and not mechanically repetitive.

# QUESTION # How can India balance the growing demand for poultry and livestock products with the urgent need for sustainable and ethical farming practices, while addressing the socioeconomic challenges faced by small-scale farmers and ensuring food security for its vast population?

# STYLE # Follow the simple writing style common in communications. Be persuasive yet maintain a neutral tone. Avoid sounding too much like sales or marketing pitch.

# AUDIENCE # The primary audience is of Indian origin, so content should incorporate cultural familiarity, societal norms, and linguistic nuances relevant to Indian readers.

# RESPONSE # Generate a well-structured and engaging piece of content adhering to the above parameters. The writing should feel natural, contextually appropriate, and resonate with the target audience.

---

A critical dimension of construction involves what we term Indianization of personas. This refers to the systematic transformation of generic or globally trained personas into forms that reflect Indian socio-cultural realities. Without explicit Indianization, a persona such as a "climate change scientist" defaults to discourses around carbon markets and individual consumption, topics common in Western debates but misaligned with India's climate discourse, which foregrounds equity in development, historical responsibility of industrialized nations, and tensions between growth and sustainability. To operationalize Indianization, each generated persona undergoes a cultural compliance review conducted by an LLM-based agent. This review evaluates whether the persona reflects appropriate cultural values, avoids stereotypes, and embodies authentic Indian perspectives rather than Western projections. Personas failing this evaluation are recycled through modified prompts

until compliance is achieved. Validated personas are then assigned to functional tasks within relevant domains, ensuring both expertise alignment and cultural fidelity.

The example persona in this subsection highlights the granularity and cultural depth of PersonaHub. Rather than a neutral occupational profile, the poultry farmer is framed through national agricultural priorities, indigenous breeds, and socio-economic realities of small-scale farmers, perspectives that resonate strongly within Indian discourse but would be absent or flattened in generic datasets. By operationalizing Indianization at scale, PersonaHub establishes a foundation for culturally faithful synthetic data generation, enabling downstream models to reflect authentic Indian voices across diverse domains.

**Data Distillation via Indic PersonaHub: Constructing Culturally-Grounded Synthetic Population**

Figure 8: Framework for building **Indic Persona Hub**

### B.5.1 CULTURALLY-GROUNDED TEXT GENERATION: FROM PERSONA TO PRODUCTION

The complete distillation pipeline (Figure 8) implements a multi-stage architecture designed to combine the scale of synthetic generation with rigorous quality assurance. At every stage, feedback loops and cultural checks ensure that only thoroughly vetted material enters the corpus. This reflects a central principle: scale alone is insufficient, and large volumes of synthetic data must be continuously filtered for coherence, factual reliability, and cultural resonance. The process begins with population synthesis, where more than 300 million raw personas are generated across 1400+ domains. While this stage establishes the breadth of identities, it also introduces the challenge of refinement, since not all personas are equally coherent or contextually appropriate. To address this, persona–task pairs are evaluated for relevance, verifying that each persona is logically aligned with the task it is assigned. A Carnatic musician persona, for instance, cannot meaningfully generate content for Hindustani gharana traditions, just as a Vedic mathematics expert approaches teaching with assumptions that diverge from those of a computational number theory specialist. Pairs that fail this relevance check are returned for reassignment rather than advanced to later stages, preventing the accumulation of weak alignments that would otherwise dilute the corpus.

Once coherence is ensured, approved pairs move to action execution, where persona-guided prompts generate the primary textual outputs. These outputs undergo multiple layers of validation, checking for factual accuracy, internal consistency, and cultural alignment with the persona's framing. Validated material then proceeds to translation, where specialist agents produce high-quality Indic language text. This stage leverages ensemble methods and post-correction techniques from our translation pipeline, ensuring outputs read as natural, idiomatic language rather than mechanically translated text carrying cross-lingual artifacts. A final enrichment stage polishes fluency and coherence, ensuring that the stored corpus balances the advantages of synthetic scale with the standards of linguistic and cultural quality required for downstream use.

The pipeline's core innovation is treating Indianization as a structural design principle rather than a superficial layer. Cultural grounding is embedded across three stages. First, personas are Indianized, transforming generic global templates into contextually authentic actors situated in Indian socio-cultural environments. Second, tasks are reframed through reflective, domain-specific prompts that elicit reasoning grounded in Indian ethical, social, and political frameworks—for instance, posing bioethics questions in terms of India's regulatory, moral, and equity considerations rather than universalized Western norms. Finally, the generation stage produces long-form English that is fluent yet culturally resonant, incorporating familiar narrative conventions, societal norms, and locally meaningful modes of reasoning.

By embedding cultural interventions throughout the pipeline, synthetic generation becomes genuinely adaptive rather than merely grammatical. While translation yields correct Indic text, culturally grounded synthesis captures how Indians reason and communicate within their discourse traditions. This layered design treats cultural authenticity as a core quality criterion, ensuring the corpus scales without losing fidelity to the perspectives it aims to represent.

## B.6 SYNTHETIC AUGMENTATION AND REWRITING

Complementing persona-driven synthesis, our QA extraction pipeline transforms unstructured Indic text into high-quality instruction data through four-stage processing. Context-aware chunking segments raw text into 1000-4000 token spans, preventing mid-sentence breaks and preserving logical coherence. Each segment is interpretable as standalone unit, critical for question generation where questions must be answerable from chunk information alone. Chunking respects document structure, treating section boundaries, paragraph breaks, and functional definitions as natural segmentation points maintaining semantic integrity.

Each chunk undergoes relevance checking and domain classification, filtering ephemeral or Western-centric content while assessing cultural relevance to Indian contexts. Valid segments receive domain assignments (Healthcare, Finance, History, Culture, BFSI, Education, Governance, Law, News, Sports, Tourism) through multi-label classification recognizing content often spans domains, India's pharmaceutical industry bridges Healthcare, Business, and Governance simultaneously. The pipeline processed 1121 chunks from Wikipedia Indic articles, 619 from DharmaWiki covering religious and philosophical traditions, and 4775 from diverse sources including government reports and news archives, yielding corpus balancing encyclopedic knowledge with culturally specific content underrepresented in Western knowledge bases.

From validated chunks, the pipeline generates fully self-contained questions spanning general explanation (comprehension), commonsense reasoning (implicit cultural knowledge), causal reasoning (exploring relationships), and open-ended prompts (inviting analysis). This diversity ensures models develop multiple reasoning capabilities beyond fact retrieval. Each question receives two answer forms: crisp answers for fact-based queries, and detailed answers (3-5 sentences) supplying explanatory context connecting questions to broader domain knowledge. This multi-fidelity generation recognizes different use cases: conversational agents benefit from detailed contextually rich responses while fact-checking systems require concise verifiable statements.

### B.6.1 ABLATION EXPERIMENT: CONVENTIONAL VS DISTILLED DOWNSTREAM PERFORMANCE

The main idea behind this ablation is to introduce instruction-style data during pretraining to observe the kinds of signals it produces. In particular, we want to understand how our constructed instruction/SFT-style data influences model behavior when integrated directly into the pretraining mix. The empirical impact of comprehensive distillation, combining persona-driven synthesis with structured QA extraction, is shown in Table 13, comparing models fine-tuned on open source data against our in-house recipe. Across thirteen diverse benchmarks spanning commonsense reasoning, knowledge assessment, and truthfulness, our distilled data demonstrates consistent improvements. HellaSwag accuracy advances from 70.47 to 73.07, with Hindi version improving from 44.01 to 44.59. More dramatic gains appear on challenging tasks: MMLU Pro (Wang et al., 2024) improves from 5.23 to 8.73 exact match (67% relative improvement), while CommitmentBank (de Marneffe et al., 2019) advances from 30.36 to 57.14, nearly doubling performance. Indic-specific evaluations show MILU (Verma et al., 2025) Hindi improving from 28.87 to 32.26 and Sanskriti States

from 55.13 to 55.91. Consistency across diverse task types validates our hypothesis that culturally grounded, persona-driven synthetic data provides training signal qualitatively different from mechanically curated or translated datasets.

Table 13: Comparison of Open Source SFT and In-house SFT Data Recipe across different tasks.

| Task | Score Name | Open Source SFT | In-house SFT Data Recipe |
|---|---|---|---|
| hellaswag | acc_norm, none | 70.47 | 73.07 |
| hellaswag_hi | acc_norm, none | 44.01 | 44.59 |
| global_mmlu_full_en | acc, none | 37.89 | 37.40 |
| global_mmlu_full_hi | acc, none | 31.43 | 31.65 |
| mmlu_pro | exact_match, custom-extract | 5.23 | 8.73 |
| piqa | acc_norm, none | 78.24 | 79.22 |
| winogrande | acc, none | 62.04 | 62.19 |
| truthfulqa_gen | bleu_acc, none | 35.74 | 37.70 |
| truthfulqa_mc1 | acc, none | 27.17 | 29.74 |
| cb | acc, none | 30.36 | 57.14 |
| milu_English | acc, none | 35.95 | 37.19 |
| milu_Hindi | acc, none | 28.87 | 32.26 |
| sanskriti_states | acc, none | 55.13 | 55.91 |

By combining synthetic article generation anchored in culturally grounded personas with structured QA extraction from authentic Indic sources, the distillation pipeline achieves both breadth and depth. The synthetic component provides scale and domain coverage impossible through manual curation alone, while extraction ensures grounding in real-world Indian knowledge sources and linguistic patterns. The result is a resource that is factually grounded, instruction-ready, and culturally resonant, uniquely suited for fine-tuning language models for Indian contexts. This infrastructure represents not merely a data processing pipeline but a fundamental rethinking of training data construction for low-resource, culturally distinct linguistic communities in an era dominated by English-centric foundation models trained primarily on Western corpora.

## C INDIC MMLU

### C.1 INDIC MMLU CONSTRUCTION AND VALIDATION

Indic MMLU was constructed through a multistage translation and refinement pipeline (see Figure 9), beginning with first-pass machine translation and followed by iterative refinement from multilingual LLMs to ensure idiomaticity and domain fidelity. Validation combined *LLM-as-judge* ratings (math correctness, linguistic fluency, and coherence) with embedding-based semantic checks; Tables 14 and 15 summarize these results.

### C.2 DETAILS OF MULTISTAGE VALIDATION PIPELINE EMPLOYED FOR INDIC MMLU

#### C.2.1 CONSENSUS BASED LLM-AS-JUDGE RATINGS

To systematically evaluate translation quality, we employed large language models (LLMs) as judges, each operating under one of three carefully designed expert personas:

- **Math Expert Persona**
  Rated mathematical correctness, technical terminology usage, and fidelity of mathematical concepts.

- **Linguistic Expert Persona**
  Assessed fluency, grammatical correctness, idiomatic usage, and overall naturalness of the translation.

- **Coherence Expert Persona**
  Evaluated semantic alignment, logical flow, and clarity between the English source and the translated output.

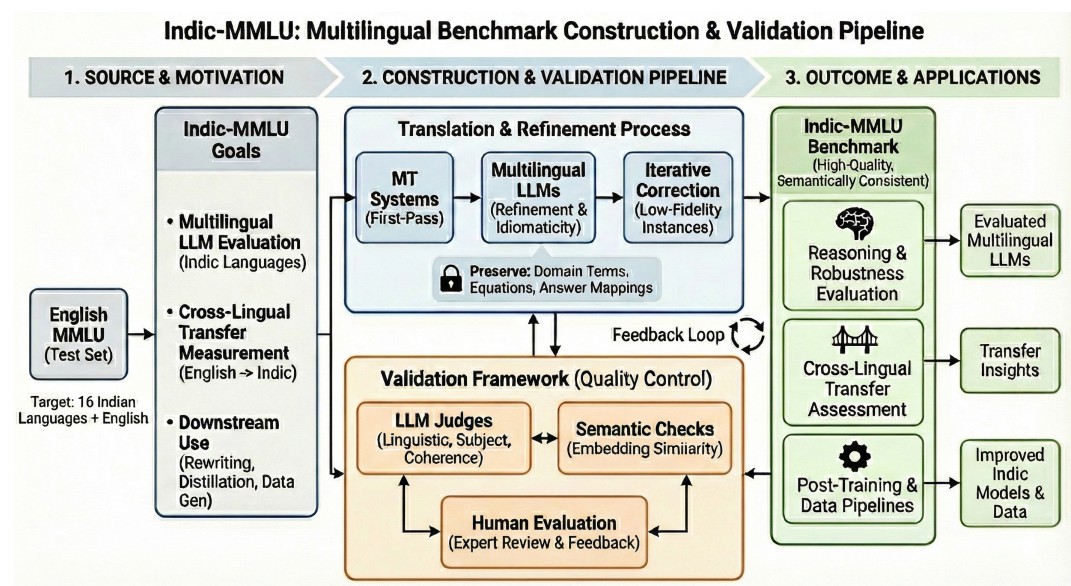

Figure 9: Indic MMLU — end-to-end construction pipeline (overview).

**Evaluation Dimensions Measured:**

- Language quality, fluency, and linguistic suitability.

- Mathematical correctness and preservation of technical content.

- Overall coherence and semantic consistency with the original English text.

Results of Math Judge, Coherence Judge and Linguist Judge can be seen in table 14

| Language | Avg Maths Rate | Avg Coherence Rate | Avg Linguist Rate | Total Records |
|----------|----------------|--------------------|--------------------|---------------|
| Hindi | 9.658029 | 9.722054 | 9.667891 | 14042 |
| Bengali | 9.182561 | 9.131505 | 8.202293 | 14042 |
| Nepali | 9.473793 | 9.331505 | 8.402293 | 14042 |
| Telugu | 9.220837 | 8.975929 | 7.986256 | 14042 |
| Odia | 9.264421 | 8.944096 | 8.220268 | 14042 |
| Punjabi | 9.348027 | 9.049138 | 8.253454 | 14042 |
| Assamese | 9.147201 | 9.039097 | 8.133243 | 14042 |
| Sanskrit | 8.338485 | 8.112520 | 6.390115 | 14042 |
| Kannada | 9.322746 | 9.111736 | 8.131819 | 14042 |
| Sindhi | 2.070645 | 3.109956 | 2.478137 | 14042 |
| Gujarati | 9.260647 | 9.206523 | 8.368965 | 14042 |
| Marathi | 9.301951 | 9.342045 | 8.381926 | 14042 |
| Tamil | 9.047856 | 9.042943 | 8.071144 | 14042 |
| Malayalam | 9.134952 | 8.998148 | 7.956488 | 14042 |
| Maithili | 9.263282 | 9.118146 | 8.218772 | 14042 |

Table 14: Summary of **LLM-as-Judge** ratings (scale 1–10). Higher values indicate better performance on mathematical correctness, coherence, and linguistic quality.

**LLM as a Judge Consensus Analysis**

- Ratings are reported on a **1–10 scale**, where higher values indicate better translation and enhancement quality.

- Most languages achieve **consistently high averages** (greater than 8.9), demonstrating strong overall quality after enhancement.

- A small number of outliers highlight **potential areas for further targeted refinement**.

### C.2.2 EMBEDDING-BASED SEMANTIC ANALYSIS

Cosine similarity provides an automated measure of **semantic closeness** between the translated or enhanced text and its original English counterpart.

- Typical mean similarity scores range from **0.76 to 0.85** across languages; higher values indicate better semantic preservation.
- Use these scores as a **triage mechanism**:
    - **Low similarity** → manual review recommended.
    - **High similarity** → likely semantically faithful.

| Language (embedding key) | Mean Similarity | Std Similarity | Min Similarity | Max Similarity |
|---|---|---|---|---|
| mmlu_as_in_qwen3_embed | 0.8045 | 0.0597 | 0.4644 | 1.0000 |
| mmlu_bn_in_qwen3_embed | 0.8133 | 0.0504 | 0.5276 | 1.0000 |
| mmlu_gu_in_qwen3_embed | 0.8211 | 0.0535 | 0.4016 | 1.0000 |
| mmlu_hi_in_qwen3_embed | 0.8472 | 0.0489 | 0.3688 | 1.0000 |
| mmlu_kn_in_qwen3_embed | 0.8106 | 0.0565 | 0.4660 | 1.0000 |
| mmlu_mai_in_qwen3_embed | 0.8226 | 0.0504 | 0.4936 | 1.0000 |
| mmlu_ml_in_qwen3_embed | 0.8158 | 0.0531 | 0.5109 | 1.0000 |
| mmlu_mr_in_qwen3_embed | 0.8129 | 0.0513 | 0.5281 | 1.0000 |
| mmlu_ne_in_qwen3_embed | 0.8242 | 0.0502 | 0.4802 | 1.0000 |
| mmlu_or_in_qwen3_embed | 0.8159 | 0.0555 | 0.4684 | 1.0000 |
| mmlu_pa_in_qwen3_embed | 0.8246 | 0.0516 | 0.5450 | 1.0000 |
| mmlu_sa_in_qwen3_embed | 0.7912 | 0.0574 | 0.4981 | 1.0000 |
| mmlu_sdd_in_qwen3_embed | 0.7646 | 0.0735 | 0.3633 | 0.9676 |
| mmlu_ta_in_qwen3_embed | 0.7964 | 0.0559 | 0.5242 | 1.0000 |
| mmlu_te_in_qwen3_embed | 0.8006 | 0.0524 | 0.5379 | 1.0000 |

Table 15: Cosine similarity statistics between translated/enhanced items and English originals (embedding model: Qwen3). Values close to 1 indicate stronger semantic preservation.

**Cosine Similarity Analysis**

Using the Qwen multilingual embedding model[16], we computed cosine similarities between English MMLU items and their translated counterparts. Overall results indicate strong semantic preservation across languages.

- **High mean similarity (0.80–0.85)** for most languages (e.g., Assamese, Bengali, Gujarati, Hindi, Maithili, Malayalam, Marathi, Nepali, Odia, Punjabi), reflecting faithful translation quality. Hindi achieves the highest mean (0.8472).
- **Moderate scores** for Sanskrit (0.79), Tamil (0.79), Telugu (0.80), and Kannada (0.81), likely due to greater morphological and syntactic divergence from English.
- **Sindhi (Devanagari)** shows the lowest alignment (mean 0.7646, highest variance), suggesting limited embedding coverage and areas for targeted refinement.
- Minimum similarities (0.36–0.53) highlight occasional divergences arising from complex or paraphrased items, while most languages reach a maximum of 1.0 for formulaic or short content.

Overall, these results indicate that the translation pipeline maintains strong semantic fidelity, with cosine similarity serving as an effective triage tool for identifying cases requiring manual review.

### C.3 FRONTIER OPEN SOURCE MODEL COMPARISON ON INDIC MMLU

We present the top-performing models on our Indic MMLU benchmark, organized by language. To facilitate readability, we first define concise abbreviations for each model, followed by a streamlined table summarizing the top-5 models per language. Refer to the Indic MMLU construction pipeline (Figure 9)). Table 17 shows Frontier LLM's performance over Indic MMLU.

**Model Abbreviations:** K2 corresponds to kimi-k2; DS3.1 represents DeepSeekV3.1; DS3 denotes DeepSeekV3-0324; DSR1 is DeepSeekR1-0528; L4M stands for

---

[16]https://huggingface.co/Qwen/Qwen3-Embedding-4B

LLaMA-4-Maverik; **Q-Instr** refers to `Qwen3-235B-A22B-Instruct-2507`; **Q-Think** denotes `Qwen3-235B-A22B-Thinking-2507`; **Q80** is `Qwen3-Next-80B-A3B-Instruct`; and **GLM4.5** corresponds to `GLM 4.5`. See Table 16 for per-language top-5 lists.

Table 16: Top-5 Models by Indic MMLU for Each Language (Abbreviations)

| Language | Top-5 Models | Language | Top-5 Models |
|---|---|---|---|
| English | K2, DS3.1, Q-Instr, DS3, L4M | Nepali | Q-Instr, Q-Think, L4M, Q80, GLM4.5 |
| Hindi | L4M, Q-Instr, DS3, DSR1, DS3.1 | | |
| Marathi | L4M, Q-Instr, Q-Think, DS3, DS3.1 | Malayalam | L4M, Q-Instr, Q-Think, DS3, Q80 |
| Gujarati | L4M, Q-Instr, DS3, DSR1, DS3.1 | Tamil | L4M, Q-Instr, Q-Think, DS3, DS3.1 |
| Bengali | L4M, Q-Instr, DS3, Q-Think, DS3.1 | Telugu | L4M, Q-Instr, Q-Think, DS3, DSR1 |
| Assamese | Q-Instr, L4M, Q-Think, DS3, DS3.1 | Kannada | L4M, Q-Instr, Q-Think, DS3, DS3.1 |
| Punjabi | L4M, Q-Instr, Q-Think, GLM4.5, K2 | Maithili | Q-Instr, L4M, Q-Think, DS3, DSR1 |
| | | Santhali | Q-Instr, L4M, Q-Think, DS3, DSR1 |
| Sindhi | Q-Instr, L4M, Q-Think, K2, Q80 | **Average** | L4M, Q-Instr, Q-Think, DS3, DS3.1 |

| Model | Parameters | Indic MMLU |
|---|---|---|
| DeepSeekR1-0528 | 685B | 0.67 |
| DeepSeekV3-0324 | 685B | 0.68 |
| DeepSeekV3.1 | 685B | 0.68 |
| Gemma-3 27B | 27B | 0.63 |
| gpt-oss-120B High | 120B | 0.48 |
| gpt-oss-120B Med | 120B | 0.48 |
| gpt-oss-120B Low | 120B | 0.48 |
| gpt-oss-20B High | 20B | 0.52 |
| gpt-oss-20B Med | 20B | 0.52 |
| gpt-oss-20B Low | 20B | 0.52 |
| kimi-k2 | 1T | 0.66 |
| Qwen3-235B-A22B-Instruct-2507 | 235B | 0.72 |
| Qwen3-235B-A22B-Thinking-2507 | 235B | 0.70 |
| Qwen3-0.6B | 0.6B | 0.32 |
| GLM 4.5 | 358B | 0.66 |
| Gemma-7B | 7B | 0.24 |
| Mistral-7B-v0.1 | 7B | 0.28 |
| Mistral-7B-v0.2 | 7B | 0.29 |
| Mistral-7B-v0.3 | 7B | 0.27 |
| LLaMA-4-Scout | 109B | 0.64 |
| LLaMA-4-Maverik | 402B | 0.72 |
| Mistral-Small-24B-Instruct-2501 | 24B | 0.41 |
| LLaMA-3.1-70B | 70B | 0.60 |
| Qwen3-Next-80B-A3B-Thinking | 80B | 0.65 |
| Qwen3-Next-80B-A3B-Instruct | 80B | 0.67 |
| Mistral-Large-Instruct-2411 | 24B | 0.54 |

Table 17: Indic MMLU scores across a range of open-source frontier and small-scale models.

## C.4 INDIC MMLU SCORES BY LANGUAGE

We benchmarked every tier of frontier open-source models on the Indic MMLU. The language-wise results for each language are shown below: Assamese (Figure 13a), Bengali (Figure 13b), English (Figure 14a), Gujarati (Figure 14b), Hindi (Figure 15a), Kannada (Figure 15b), Maithili (Figure 16a), Malayalam (Figure 16b), Marathi (Figure 17a), Nepali (Figure 17b), Oriya (Figure 18a), Punjabi (Figure 18b), Sanskrit (Figure 19a), Sindhi (Figure 19b), Tamil (Figure 20a), and Telugu (Figure 20b). Two images are stacked per page for improved readability, with spacing between them to avoid crowding. Additional details: G

# D  DATA ACQUISITION AND GOVERNANCE

Building equitable representation for India's 22 official languages demands more than incremental improvements to existing pipelines. It requires rethinking data acquisition from the ground up. The challenge is not simply one of scale, but of diversity, authenticity, and cultural grounding. Most large language models today are trained on web-scraped corpora that privilege high-resource languages, leaving Indic languages with a fraction of the data volume and virtually none of the domain-specific depth needed for real-world applications.

The foundation of MILA draws from established multilingual corpora similar to Pile (Gao et al., 2021), RedPajama (Weber et al., 2024), and C4 (Raffel et al., 2023), incorporating approximately 5 billion words gathered through multi-source web crawling of multilingual websites, forums, and academic repositories, apart from over 1700 open datasets from HuggingFace[17]. However, web-scraped data alone cannot address the cultural and linguistic gaps inherent in translated or synthetic corpora. To mitigate these limitations, we prioritized curated book collections, amounting to approximately 32 billion words across 16 languages, sourced primarily from Archive.org[18] and the National Digital Library of India[19]. These collections offer authentic content and culturally grounded materials that complement the breadth of crawled and open-source data, ensuring that MILA captures not just linguistic patterns but also the educational and cultural contexts that define how these languages are actually used.

Collecting these books and academic papers posed significant systems challenges that extended far beyond simple download automation. Standard pipelines frequently stalled, exceeded resource limits, or required weeks of wall-clock time, making large-scale acquisition impractical without fundamental architectural innovations. The heterogeneity of sources demanded source specific solutions that could operate at unprecedented concurrency while maintaining provenance tracking, license compliance, and fault tolerance. We therefore designed custom infrastructure and optimization layers that reduced runtime and compute usage by 40–70%, enabling sustained acquisition at scales that standard tools could not achieve. This section details our approach across four major acquisition domains: Archive.org, NDLI, Wikimedia, and the underlying infrastructure that made large-scale harvesting feasible.

## D.1  ARCHIVE.ORG

Archive.org represents one of the most valuable yet underutilized resources for Indic language modeling. Unlike web-scraped text, which often suffers from quality inconsistencies, duplication, and cultural decontextualization, Archive.org provides digitized books that have undergone editorial processes, domain specialization, and cultural embedding. For Indic languages in particular, Archive.org hosts extensive collections of historical texts, classical literature, government documents, and subject-specific materials that are virtually absent from standard web corpora. However, accessing this wealth of content at scale required overcoming substantial technical and organizational challenges related to metadata discovery, download orchestration, and quality assurance.

As seen in Table 18, Archive.org, we extracted over 1,012,198 digitized PDFs spanning multiple Indic languages and government documents, totaling 82,104,639 pages. This collection forms one of the richest curated corpora for Indic language model training. Bengali and Sanskrit provide substantial depth, contributing 10.95 million and 10.09 million pages, respectively, encompassing 3.10 billion and 2.68 billion words. Bengali materials include both modern literature and historical documents from the Bengal Renaissance, while Sanskrit holdings feature philosophical treatises, grammatical texts, and mathematical manuscripts spanning millennia. Hindi offers the highest number of individual PDFs at 396,120, with 7.53 million pages and 4.15 billion words, reflecting contemporary textbooks, popular literature, and government publications. Malayalam and Marathi further enhance the corpus, with Malayalam's 65,030 PDFs spanning 2.18 million pages (1.06 billion words) and Marathi's 124,220 PDFs covering 3.02 million pages (1.26 billion words), providing a mix of traditional, historical, and contemporary literary forms. Government documents across multiple languages supplement these sources, offering structured policy-oriented content, including legal codes,

---

[17]https://huggingface.co
[18]https://archive.org
[19]https://ndl.iitkgp.ac.in

Table 18: Archive.org Corpus by Language (Books, Pages, and Word Counts)

| Language | # PDFs | # Pages | Word Count |
|---|---|---|---|
| Hindi | 396.12 K | 7.53 M | 4.15 B |
| Marathi | 124.22 K | 3.02 M | 1.26 B |
| Malayalam | 65.03 K | 2.18 M | 1.06 B |
| Telugu | 77.86 K | 5.93 M | 1.53 B |
| Tamil | 43.59 K | 5.28 M | 1.44 B |
| Kannada | 41.71 K | 4.08 M | 1.01 B |
| Sanskrit | 44.49 K | 10.09 M | 2.68 B |
| Bengali | 41.25 K | 10.95 M | 3.10 B |
| Urdu | 126.03 K | 32.15 M | 10.03 B |
| English (Maths, Ayurveda) | 45.10 K | 2.57 M | 0.89 B |
| **Total** | **1 M** | **84.00 M** | **27.15 B** |

census reports, and administrative guidelines, grounding language models in governance-specific discourse.

To enhance subject coverage beyond general literature and government documents, we further targeted subject collections such as Mathematics, Ayurveda, and Agriculture. These domain-focused subsets represent critical areas where Indic-language expertise exists but is poorly represented in standard training corpora. Mathematical texts include both classical Indian mathematics such as works on algebra, geometry, and astronomy from historical mathematicians, and modern textbooks covering calculus, statistics, and applied mathematics. Ayurveda collections encompass classical texts, commentaries, materia medica, and clinical guidelines, providing comprehensive coverage of traditional medical knowledge. Agriculture materials include crop management guides, soil science texts, veterinary manuals, and rural development documents that reflect India's agricultural diversity and traditional farming knowledge. These domain-focused subsets were subjected to optical character recognition post-processing and correction via large language models, improving text quality and making the data more usable for supervised fine-tuning.

The acquisition process for Archive.org materials required navigating several technical challenges. Archive.org exposes its metadata through paginated search endpoints, each capped at approximately 10,000 results. For large collections spanning hundreds of thousands of items, this pagination limit necessitates sophisticated query strategies that partition the search space into manageable chunks. Metadata quality is heterogeneous across collections, with inconsistencies in language tagging, missing bibliographic information, and ambiguous licensing statements. Long-running download jobs are fragile, often stalling due to network issues, rate limiting, or server-side errors. Naïve sequential ingestion approaches required several days even on high-bandwidth machines, making comprehensive collection infeasible within reasonable time frames. The details on how these challenges were tackled are given in D.4.

## D.2 NDLI

The National Digital Library of India represents a fundamentally different acquisition paradigm compared to Archive.org. Where Archive.org provides broad historical and literary coverage, NDLI offers curriculum-aligned, licensed, and provider-attributed materials explicitly designed for educational use. This structural difference makes NDLI uniquely valuable for training language models that must operate in pedagogical contexts, answer curriculum-based questions, or generate educational content aligned with Indian educational standards. NDLI materials span two disjoint strata: school and state-boards covering K-12 education, and higher education encompassing undergraduate, postgraduate, and diploma programs. We report each stratum on its own axes to avoid conflating counts across heterogeneous groupings, allowing us to preserve the pedagogical and institutional context of each item while facilitating targeted OCR and post-correction workflows adapted to Indic scripts and educational content.

State-board and NCERT materials provide grade-sequenced, syllabus-aligned content in Indic languages and English, making this stratum essential for curriculum-grounded pretraining and super-

Table 19: NDLI School / State-Boards: counts by language (items).

| Language | Count |
|----------|-------|
| Hindi | 4,114 |
| English | 3,785 |
| Urdu | 1,064 |
| Telugu | 755 |
| Sanskrit | 657 |
| Kannada | 557 |
| Tamil | 552 |
| Marathi | 481 |
| Gujarati | 429 |
| Malayalam | 210 |
| Bengali | 99 |
| Oriya/Odia | 44 |
| Assamese | 19 |
| Garo | 10 |
| Bodo/Boro | 4 |
| Nepali | 1 |
| Manipuri | 1 |

Table 20: NDLI School / State-Boards: counts by class/level (items).

| Class/Level | Count |
|-------------|-------|
| Class X | 1,926 |
| Class XI | 1,611 |
| Class IX | 1,533 |
| Class XII | 1,483 |
| Class VIII | 1,351 |
| Class VII | 1,202 |
| Class VI | 1,168 |
| Class V | 635 |
| Class III | 603 |
| Class IV | 593 |
| Class I | 351 |
| Class II | 313 |

vised fine-tuning on pedagogy-aligned tasks such as worked solutions, syllabus-based question answering, and instructional content generation. The curriculum alignment is not merely topical but structural materials follow prescribed syllabi, use standardized terminology, and progress through concepts in pedagogically validated sequences. This makes NDLI school content particularly valuable for applications like automated tutoring systems, homework assistance, and educational content generation that must respect both subject matter and grade-appropriate presentation.

Table 21: NDLI School / State-Boards: counts by content provider (items).

| Provider | Count |
|----------|-------|
| SCERT Telangana | 4,247 |
| Raj-eGyan | 2,606 |
| Punjab School Education Board | 2,465 |
| Gujarat Secondary & Higher Secondary Education Board | 788 |
| Jammu & Kashmir State Board of School Education | 694 |
| Board of Secondary Education, Madhya Pradesh | 520 |
| Karnataka Secondary Education Examination Board | 416 |
| NCERT | 357 |
| SCERT Kerala | 317 |
| SCERT Tripura | 100 |
| A. P. Open School Society, Amaravati | 85 |
| Assam Higher Secondary Education Council | 59 |
| Odisha Primary Education Programme Authority | 42 |
| Board of School Education Haryana | 33 |
| NCERT — Vocational Education | 26 |
| Board of Secondary Education, Odisha | 23 |
| Kendriya Vidyalaya ASC Centre(S) | 3 |
| Kendriya Vidyalaya Devlali (No. 1) | 1 |

The distribution of NDLI school and state-board content reveals the linguistic and institutional landscape of Indian education, as detailed in Tables 19, 20, and 21. As shown in Table 19, Hindi dominates with 4,114 items, reflecting both its status as a widely taught language and the extensive digitization efforts by Hindi-medium state boards. English follows closely with 3,785 items, representing both English-medium schools and English as a subject across state boards.On te other hand,

several smaller languages including Garo with 10 items, Bodo with 4 items, Nepali with 1 item, and Manipuri with 1 item highlight the uneven digitization across states, with implications for equitable language model development.

The distribution by class level, presented in Table 20, shows relatively balanced coverage across grades with some concentration in secondary and higher secondary levels. Class X leads with 1,926 items, reflecting the significance of board examinations at this level and corresponding digitization priority. Class XI follows with 1,611 items, Class IX with 1,533 items, and Class XII with 1,483 items. These four grades collectively account for the bulk of materials, corresponding to the secondary education phase where standardized curricula are most rigorously defined and assessment is most formal. Middle and primary levels are also represented, though less prominently, ensuring coverage across the full curricular progression. This stratification supports the design of training pipelines that respect pedagogical sequencing and enables creation of evaluation sets that test a model's ability to generate grade-appropriate explanations without oversimplification or unnecessary complexity.

The provider distribution, presented in Table 21, highlights both the institutional diversity of the collection and the uneven levels of digitization commitment across Indian states. SCERT Telangana leads with 4,247 items, reflecting the state's strong investment in digital curriculum resources as part of recent education initiatives. Raj-eGyan (Rajasthan) and Punjab School Education Board follow with 2,606 and 2,465 items respectively, underscoring the momentum of state-led digitization efforts in northern India. Other contributors such as Gujarat, Jammu and Kashmir, and NCERT provide substantial but comparatively smaller shares, while several states and institutions add more modest numbers. Collectively, the distribution illustrates that while some states have achieved large-scale digitization, others remain underrepresented, pointing to regional disparities in access to digital educational content.

These tables provided complementary perspectives on the same corpus: languages highlight linguistic diversity and the multilingual nature of Indian education, grade levels ground pedagogy and enable curriculum-sequenced model training, and providers reveal provenance and institutional commitment to open education. Importantly, these dimensions were treated as orthogonal, items that were bilingual or cross-listed across grades were preserved with multiple metadata labels and deduplicated only at the item-ID level. This schema-first organization ensures that a single mathematics textbook available in both Hindi and English, or a multi-grade resource spanning Classes IX and X, is counted once but tagged with all applicable metadata. This approach enables flexible querying and sampling strategies during training while preventing artificial inflation of dataset statistics.

Table 22: NDLI Higher Education: counts by content provider (items).

| Content Provider | Count |
| --- | --- |
| LibreTexts | 7,591 |
| e-Adhyayan | 6,902 |
| Botanical Survey of India (BSI) | 976 |
| Knowledge Unleashed in Multiple Bharatiya Languages (e-KUMBH) | 478 |

Table 23: NDLI Higher Education: counts by education level (items).

| Level | Count |
| --- | --- |
| Post Graduate | 6,901 |
| Under Graduate | 1,259 |
| Diploma | 146 |

Higher education holdings from NDLI provide domain depth in Mathematics, Botany, Chemistry, Medicine, and Engineering, along with provider-curated texts amenable to precision supervised fine-tuning on derivations, definitions, proofs, and technical procedures. These materials differ fundamentally from school content in their assumed prior knowledge, technical depth, and specialized vocabulary. Higher education content is valuable not just for training models to understand advanced topics but also for enabling retrieval over structured knowledge, where precise definitions, formal proofs, and established methodologies must be accurately represented and retrievable.

The provider distribution for higher education, shown in Table 22, is dominated by LibreTexts with 7,591 items, reflecting its multi-institutional effort to curate open educational resources across STEM disciplines. e-Adhyayan follows closely with 6,902 items, highlighting a major Indian initiative aligned with national curricula. Other contributors include the Botanical Survey of India with 976 items, offering authoritative botanical references, and e-KUMBH with 478 items, which ex-

Table 24: NDLI Higher Education: top subjects (items).

| Subject | Count |
|---|---|
| Mathematics | 2,884 |
| Plants (Botany) | 900 |
| Commerce, Communications & Transportation | 875 |
| Chemistry & Allied Sciences | 757 |
| Medicine & Health | 755 |
| Engineering & Allied Operations | 194 |
| Computer Science, Information & General Works | 65 |
| Civil Engineering | 59 |
| Other Branches of Engineering | 59 |
| Plants noted for characteristics & flowers | 45 |
| Others | 208 |

pands access to higher education materials in Indian languages. The distribution by education level, presented in Table 23, shows a clear emphasis on postgraduate content with 6,901 items, followed by 1,259 undergraduate and 146 diploma-level resources. This concentration at the postgraduate level underscores the advanced and specialized nature of the collection, making it particularly valuable for training models on technical reasoning, research-oriented writing, and domain-specific knowledge.

The disciplinary specialization, shown in Table 24, is led by Mathematics with 2,884 items, offering extensive coverage across core and advanced topics that are especially valuable for developing models with strong reasoning capabilities. Botany follows with 900 items, reflecting India's rich biodiversity and the strong institutional contributions in this field. Commerce and related areas contribute 875 items, underscoring the practical relevance of economic and infrastructural studies. Other disciplines such as chemistry, medicine, and engineering add further breadth, ensuring the corpus is not only quantitatively rich but also balanced across technical, scientific, and applied domains. This subject diversity, when considered alongside provider and level distributions, enables the construction of evaluation-ready subsets tailored to curriculum progression and disciplinary expertise.

The NDLI pipeline illustrates a methodology that extends beyond mere content acquisition to systematic organization that preserves pedagogical and institutional context. Unlike Archive.org's historical focus or Wikimedia's encyclopedic coverage, NDLI provides materials explicitly designed for learning, with clear curricular alignment, grade-level appropriateness, and institutional provenance. This makes NDLI content particularly valuable for educational applications of language models, where generating pedagogically sound content, respecting curricular sequences, and providing grade-appropriate explanations are critical requirements that web-scraped data cannot reliably support.

### D.3 WIKIMEDIA

Wikimedia[20] projects form one of the largest open-access multilingual resources for Indian languages, capturing encyclopedic, cultural, educational, and archival text that complements the historical depth of Archive.org and the curricular structure of NDLI. Where Archive.org provides edited books and NDLI offers curriculum-aligned materials, Wikimedia contributes community-maintained, collaboratively edited content that reflects contemporary knowledge, cultural perspectives, and living linguistic practices. The distributed nature of Wikimedia projects—spanning Wikipedia, Wikisource, Wikibooks, and numerous specialized initiatives—provides diverse textual genres and knowledge domains, making it an essential component of comprehensive multilingual training corpora.

The language-wise distribution of Wikimedia content, presented in Table 25, reveals both the platform's multilingual breadth and the persistent resource imbalances across languages. English dominates overwhelmingly with 2.80 billion words across 3.65 million files, reflecting Wikipedia's origins as an English-language project and the continued predominance of English in online knowledge production. This massive English presence, while valuable for multilingual models that must handle

---

[20]https://commons.wikimedia.org/

Table 25: Wikimedia Corpora: Language-Wise Summary

| Language | Words (in Millions) | Files |
|---|---|---|
| English | 2.80B | 3.65M |
| Bengali | 383.53M | 996K |
| Hindi | 159.89M | 355K |
| Tamil | 152.45M | 681K |
| Telugu | 108.38M | 261K |
| Urdu | 103.67M | 168K |
| Sanskrit | 89.76M | 187K |
| Malayalam | 99.08M | 326K |
| Gujarati | 25.88M | 70K |
| Marathi | 54.13M | 149K |
| Kannada | 52.87M | 121K |
| Oriya/Odia | 18.52M | 61K |
| Punjabi | 40.48M | 112K |
| Assamese | 24.15M | 74K |
| Kashmiri | 1.53M | 3.9K |
| Nepali | 21.20M | 43K |
| Others (Manipuri, Garo, Bodo, etc.) | < 1M | < 2K |

code-switching and cross-lingual tasks, also highlights the scale of resource disparity that MILA aims to address.

Bengali emerges as the strongest Indic language with nearly 384 million words, supported by active editor communities and institutional digitization initiatives that have enabled systematic content creation. Hindi and Tamil follow with sizable volumes, though Hindi's output remains modest relative to its vast speaker base, illustrating the persistent digital divide even among widely spoken languages. Mid-resource languages such as Telugu, Malayalam, Marathi, Kannada, and Punjabi contribute substantial content, yet still represent only a fraction of the available English corpus. At the other end of the spectrum, languages like Gujarati, Assamese, Nepali, and Oriya remain underrepresented, while Kashmiri and several smaller languages, including Manipuri, Garo, and Bodo, contribute only marginal volumes. This stark disparity between high- and low-resource Indic languages underscores the uneven digital landscape and highlights the urgent need for targeted curation efforts to bridge linguistic inequities.

This language-wise view emphasizes both the relative strengths of English and high-resource Indic languages and the severe under-representation of low-resource languages. The long-tail distribution has profound implications for language model training: while English and Bengali content can support robust monolingual models, languages like Kashmiri, Manipuri, and Bodo require cross-lingual transfer, synthetic augmentation, and careful low-resource techniques to achieve even basic competence. The stark disparities also underscore why curated collections from Archive.org and NDLI are essential, web-crawled and community-maintained sources alone cannot provide the volume and quality needed for equitable language modeling across all Indic languages.

The project-wise distribution, summarized in Table 26, reveals how Wikimedia content is distributed across different knowledge domains and textual genres. Wikisource dominates with 2.09 billion words across 4.95 million files, providing archival and historical texts. Wikisource's mission is to collect and transcribe source documents—original texts, historical documents, literary works, and primary sources, that are in the public domain or permissively licensed. For Indic languages, Wikisource is particularly valuable because it hosts digitized versions of classical literature, historical chronicles, religious texts, and early modern works that are often unavailable in other digital formats. The dominance of Wikisource in total word count reflects both the length of these source documents and the systematic digitization efforts by language communities.

The Wikimedia ecosystem contributes a total of 4.42 billion words across 8.93 million files, forming one of the most diverse open repositories for multilingual content. Wikisource dominates in scale due to its digitized literary and historical texts, while Wikipedia, despite contributing fewer words, provides unparalleled topical breadth, structured knowledge, and contemporary relevance.

Table 26: Wikimedia Corpora: Project-Wise Summary

| Project | Words (in Billions) | Files (in Millions) |
|---|---|---|
| Wikisource | 2.09B | 4.95M |
| Wikipedia | 0.15B | 0.44M |
| Wikibooks | 0.10B | 0.09M |
| Wikiquote | 0.099B | 0.07M |
| Wikinews | 0.015B | 0.03M |
| Wikiversity | 0.051B | 0.04M |
| Wikivoyage | 0.044B | 0.03M |
| Others (Kidatawiki, Iwiki, Ecieswiki) | 0.17B | 1.1M |
| **Grand Total** | **4.42B** | **8.93M** |

Educational projects such as Wikibooks and Wikiversity add structured pedagogical materials, offering valuable resources for instructional and fine-tuning tasks. Smaller projects like Wikiquote, Wikinews, and Wikivoyage, though modest in size, enrich the corpus with idiomatic expressions, journalistic writing, cultural context, and descriptive language. Collectively, these repositories complement one another: literary depth from Wikisource, encyclopedic coverage from Wikipedia, didactic clarity from Wikibooks and Wikiversity, and domain-specific perspectives from smaller projects; ensuring broad linguistic and thematic diversity for model training.

The distribution highlights both the strengths and limitations of Wikimedia's community-maintained content. Wikisource contributes archival text with temporal depth and literary richness, while Wikipedia offers encyclopedic coverage and structured factual grounding. Together they support historical and contemporary linguistic research as well as information-seeking and question-answering tasks. Smaller projects such as Wikibooks, Wikiquote, Wikinews, Wikiversity, and Wikivoyage add genre diversity, exposing models to instructional writing, quotations, journalism, and travel description. At the same time, coverage is highly uneven: English and a handful of Indic languages dominate with billions of words, while many others remain under-represented. This imbalance necessitates complementary resources—curated books from Archive.org, curriculum materials from NDLI, and synthetic augmentation through the Indic-Persona Hub (Section 3.3.3). Wikimedia alone, though valuable, cannot provide sufficient representation for low-resource languages or the domain depth needed for specialized areas like agriculture, Ayurveda, or law.

Quality also varies substantially across languages. High-resource languages benefit from active editor communities, established guidelines, and systematic patrolling, while low-resource languages often face smaller communities, inconsistent editing, and greater vulnerability to low-quality contributions. To address this, we apply language-specific quality filtering based on article length, structural completeness, reference density, and community quality markers such as featured or good article status. In combination, Wikimedia's contemporary breadth and community perspectives, Archive.org's historical depth, and NDLI's institutional framing provide MILA with the linguistic diversity, domain coverage, temporal range, and cultural authenticity needed for equitable multilingual language modeling across India's diverse linguistic landscape.

### D.4 Infrastructure and Optimisation

Beyond the content and organizational strategies detailed in the previous sections, the acquisition of MILA required fundamental innovations in systems infrastructure and optimization. Collecting large-scale academic corpora from heterogeneous sources such as Archive.org and NDLI was not merely a data challenge but also a systems challenge demanding custom-built solutions that could operate at unprecedented scale while maintaining reliability, provenance, and efficiency. Standard pipelines frequently stalled, exceeded resource limits, or required weeks of wall-clock time, making comprehensive acquisition impractical. We therefore designed custom infrastructure and optimization layers that reduced runtime and compute usage by 40–70%, enabling sustained acquisition at scales that generic tools could not achieve. This section details the technical architecture, optimization strategies, and governance frameworks that made large-scale multilingual corpus construction feasible.

Table 27: BitTorrent performance comparison: standard clients vs. our optimized engine.

| Metric | Standard BT | Our Engine | Improvement |
|---|---|---|---|
| Zero-leech speed | ∼50 KB/s | ∼200 KB/s | 3× |
| Max connections | ∼200 | 30,000 | 150× |
| Cache usage | ∼5% RAM | 60% RAM | 12× |
| Concurrent downloads | 3–5 | 30 | 6× |
| Stall recovery | Manual | Auto (5–10s) | — |

**Large-Scale Corpus Acquisition: A Metadata-First Architecture.** Large-scale corpus acquisition presents a fundamental trade-off: high throughput demands aggressive parallelism but increases failures and complicates provenance tracking, while reliability requires conservative allocation and checkpointing that reduces throughput. Governance adds further overhead through metadata tracking, license verification, and audit trails. We resolve this through a metadata-first architecture that treats metadata as the primary object for discovery, governance, and deduplication before text processing. This enables license-aware filtering before download, record-level deduplication via stable identifiers, and targeted discovery by subject, language, and education level, while reducing costs by eliminating redundant downloads.

**Unified Metadata Schema and Hierarchical Deduplication.** At the scale of tens of millions of documents, direct text-based deduplication or OCR is infeasible as a first pass. We therefore construct a unified metadata schema that normalizes identifiers (ISBN, DOI, handle, archive ID), bibliographic data (title, author, publisher, year, edition), technical attributes (filesize, extension, pages), governance information (license, rights, timestamps), and integrity checks (MD5, URL, cover image). This consistent representation enables license-aware filtering and prepares the ground for robust deduplication. Deduplication proceeds hierarchically by combining multiple signals to maximize accuracy while minimizing false positives. Records are marked as duplicates if any hard key—canonical URL, DOI, ISBN, or MD5—matches exactly. Otherwise, soft matching bundles normalized title, author, and publication year, with corroborating attributes like filesize and page count, to identify duplicates within tolerance thresholds. All decisions are logged with record IDs and triggering signals for auditability. This pipeline effectively eliminates redundancies, particularly in Archive.org where popular books often reappear across collections, mirrors, and formats (PDF, EPUB, DJVU).

**Governance-First Licensing.** Since many collections contain restrictive or ambiguous licenses, we enforce governance as a primary constraint. Each item is tagged with licensing metadata and provenance snapshots. Non-permissive or uncertain items are quarantined for research-only use, contributing to coverage statistics but excluded from training and redistribution. Only permissively licensed content from Archive.org public domain collections, NDLI open textbooks, and Wikimedia projects enters the training pipeline.

**Source-Specific Challenges.** Archive.org provides rich bibliographic metadata including stable identifiers, collection memberships, and language tags, but exhibits occasional language field errors with Hindi mislabeled or Indic content lacking tags. Technical challenges include paginated search endpoints capped at 10,000 results per query and fragile long-running jobs. We addressed this via adaptive query planning that slices queries by date ranges, subjects, and languages to bypass result windows, feeding asynchronous multi-semaphore crawlers with pause-resume checkpoints. Separate semaphores for metadata fetching, downloading, and post-processing allow each stage to proceed at its natural rate. This reduced ingestion time from 7 days to 24 hours on comparable hardware, saving 40–70% in compute overhead. NDLI supplies structured metadata tied to education levels and subject facets, enabling curriculum-aligned slicing. Challenges include multilingual misclassification, sparse ISBN coverage, and inconsistent subject tagging. We apply normalization layers mapping provider vocabularies to standardized taxonomies and flag ambiguous records for manual review.

Metadata-first processing has documented limitations: pervasive language misclassification (Hindi as English, script-metadata mismatches), licensing gaps requiring collection-based heuristics, and edition ambiguities from multiple ISBNs or minor reprints. Future iterations will integrate OCR-based post-processing: text-based deduplication using MinHash/SimHash to detect edition-level

reprints; OCR-based language identification with script-aware classifiers (fastText, CLD3) to correct mislabels; and confidence scoring combining metadata with OCR predictions to quantify uncertainty. Yet, this strategy provides a governed, efficient, and auditable baseline enabling acquisition at scales content-first approaches cannot achieve. While OCR-based deduplication and language identification will further improve quality, this hybrid approach (metadata for scale and governance, content analysis for quality) maximizes the value of large collections for Indic-focused LLM pretraining and domain-specific fine-tuning. The infrastructure represents a foundational contribution extending beyond our immediate needs, providing reusable patterns for equitable language technology development across linguistic diversity.

## E  Data Organisation

Building a multilingual Indic dataset spanning 7.5 trillion tokens presents governance challenges that fundamentally differ from those encountered in conventional English corpus construction, as emphasized by foundational work on dataset documentation and transparency (Gebru et al., 2021; Jernite et al., 2022). Unlike English corpora, which benefit from decades of standardization efforts, established digitization practices, and relatively uniform encoding conventions, Indic data confronts structural fragmentation across multiple dimensions simultaneously. Sources span digitized textbooks with varying OCR quality, newspapers employing inconsistent orthographic conventions, government documents using legacy encoding schemes, social media content mixing scripts and languages within single posts, and historical archives where material has been digitized under different technical standards across decades. This heterogeneity manifests not merely as noise to be filtered but as fundamental diversity requiring preservation—the very linguistic variation that makes low-resource languages distinct risks being erased through overly aggressive normalization. The challenge extends beyond scale to encompass control: ensuring that truly low-resource languages are preserved in the long tail of the distribution rather than being overwhelmed by higher-resource languages, verifying that Unicode normalization operations do not inadvertently collapse phonologically distinct characters that appear visually similar across scripts, and maintaining complete auditability of every transformation applied to source material so that downstream model behaviors can be traced back to specific data processing decisions.

Without rigorous governance infrastructure and systematic taxonomic organization, a trillion-token Indic dataset risks becoming simultaneously too brittle for production deployment and too opaque for scientific reproducibility. Brittleness emerges when licensing restrictions are inadequately tracked, causing models trained on the corpus to inherit legal liabilities; when personally identifiable information leaks through inadequate filtering; or when quality degradation in specific language-domain combinations goes undetected because monitoring lacks the granularity to surface issues affecting small subpopulations. Opacity arises when transformation lineage is lost, making it impossible to debug model behaviors by examining training data provenance; when versioning is ad-hoc, preventing reproducible experimentation; or when metadata is incomplete, leaving researchers unable to construct domain-specific subsets or balance corpus composition across linguistic and topical dimensions. These governance failures compound in low-resource language contexts, where the community lacks the scale to absorb quality issues through redundancy and where each dataset artifact represents irreplaceable cultural and linguistic resources that cannot be easily regenerated if corrupted or lost.

### E.1  Lakehouse Architecture: Unifying Storage, Metadata, and Governance

To address complex governance requirements, we implement a petabyte-scale AI data lakehouse architecture that unifies storage, lineage tracking, metadata cataloging, governance enforcement, and versioning. Unlike traditional data lakes, which offer scale without governance, or warehouses, which enforce governance but lack flexibility, the lakehouse paradigm combines their strengths, supporting ACID transactions and schema enforcement alongside schema-on-read adaptability for diverse machine learning corpora. The storage layer, built on JuiceFS with MinIO object storage, is organized into three zones reflecting data maturity. The Raw zone ingests source material in original form, preserving even malformed or duplicate documents to safeguard scarce Indic resources. The Curated zone stores cleaned, deduplicated, and standardized data with full metadata, while the Feature Store holds processed features such as tokenized sequences and embeddings, versioned with

their generating code for reproducibility. These tiers enforce governance checkpoints and allow safe experimentation without contaminating production data.

Lineage tracking employs OpenLineage and Marquez to capture transformation events across heterogeneous tools such as Spark, Airflow, and Kafka. OpenLineage provides a vendor-neutral specification, while Marquez aggregates lineage into a queryable graph. This enables forward queries (which artifacts depend on this source?) and backward queries (which sources produced this feature?), supporting debugging, provenance verification, and compliance. When a model exhibits anomalies on specific Indic examples, lineage reveals the complete processing path, feature engineering, cleaning, translation, or OCR, back to the source document. It also ensures auditors can verify that sensitive or licensed content is correctly handled across pipeline stages.

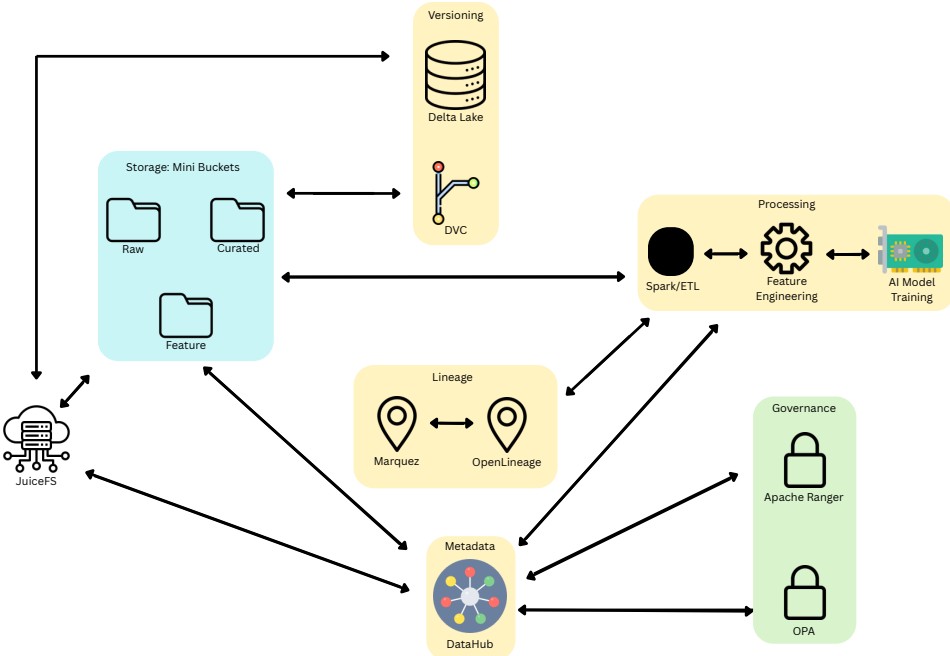

Figure 10: End-to-end data governance pipeline showing flow from raw ingestion through curated and feature layers. MinIO-backed JuiceFS storage provides the foundation, with Spark ETL pipelines processing data while emitting lineage events captured by OpenLineage and Marquez. DataHub maintains the metadata catalog, while Apache Ranger and OPA enforce governance policies at promotion boundaries. Delta Lake and DVC provide versioning and reproducibility guarantees throughout.

### E.2 METADATA CATALOGING AND TAXONOMIC ORGANIZATION

Metadata cataloging through DataHub transforms raw storage into a searchable, navigable knowledge graph where every asset is annotated with rich descriptive, operational, and governance-aligned metadata. DataHub serves as the central catalog organizing not merely individual documents but the full corpus topology, capturing relationships between source collections, processed datasets, derived features, and trained models in a unified graph structure. Every asset in our corpus is annotated with a comprehensive metadata schema spanning multiple dimensions simultaneously. Domain annotations classify content into categories including Agriculture, Culture, Education, News, Business, Healthcare, Sports, Law, Governance, Tourism, and BFSI (Banking, Financial Services, and Insurance), enabling construction of domain-specific subsets for targeted pretraining or evaluation. Language and script metadata distinguish between related but distinct linguistic varieties: Hindi in Devanagari script versus Hindi transliterated to Latin script versus Urdu in Perso-Arabic script, while also capturing code-mixed content where multiple languages appear within single documents. Modality annotations identify whether content consists of plain text, PDF documents requiring OCR,

images with embedded text, audio transcriptions, or code mixed with natural language documentation.

Quality tier metadata encodes multiple dimensions of data fidelity, including OCR confidence scores for digitized documents, readability metrics assessing linguistic complexity, and completeness indicators flagging truncated or corrupted content. License and sensitivity tags ensure safe promotion for downstream use by explicitly tracking intellectual property restrictions, personally identifiable information, and content requiring special handling due to cultural sensitivity or regulatory constraints. Source provenance captures origin information, which institutional archive, web domain, or digitization project contributed each document, enabling attribution and supporting partnerships with data providers who require usage tracking. Stage metadata indicates each asset's position in the processing pipeline, distinguishing raw ingested material from cleaned corpus ready for training from experimental features under development. Lineage metadata links assets to their processing history, capturing complete transformation graphs that enable reproducibility and debugging.

This rich metadata infrastructure serves multiple critical functions beyond basic organization. First, it enables precise corpus composition for targeted training objectives: constructing a legal domain corpus requires selecting by domain tag while filtering by quality tier and license compatibility. Second, it supports fairness and representation analysis by enabling quantitative assessment of corpus composition across languages, domains, and sources, revealing when certain linguistic communities or topical areas are underrepresented. Third, it facilitates automated dataset card generation, producing comprehensive documentation that satisfies emerging best practices for dataset transparency and responsible AI development. Fourth, it provides the substrate for governance policy enforcement, as policies can reference metadata predicates to conditionally permit or deny operations based on asset characteristics. DataHub's web interface transforms this metadata into powerful search and discovery capabilities, enabling researchers to navigate the corpus through faceted browsing, full-text search across metadata fields, and visual exploration of lineage graphs that reveal data provenance and downstream usage patterns.

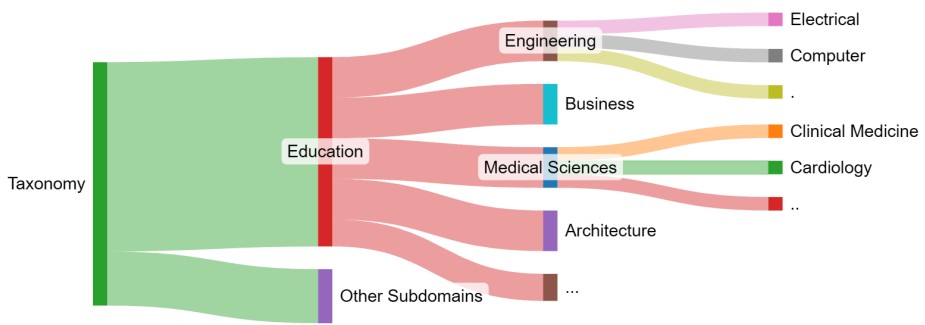

Figure 11: Sample Taxonomy Division of Education (1 of 12 Broad Domains)

Complementing the metadata infrastructure, our comprehensive taxonomy spanning over 1400 domains provides consistent structure and semantic coverage across the full range of tasks, languages, and knowledge areas represented in MILA. This taxonomy serves as a controlled vocabulary for domain classification, ensuring consistent annotation across different sources and processing stages while providing hierarchical organization that captures both broad categories and fine-grained specializations. A representative subset of the taxonomy appears in the appendix, illustrating how domain categories are organized into hierarchies—Healthcare branches into Ayurveda, Allopathic Medicine, Public Health, Traditional Healing Practices, and Medical Ethics, each with further subdivisions capturing specialized subdomains. This taxonomic structure enables not only consistent classification but also intelligent corpus subset construction: researchers can select content at varying levels of granularity, from all healthcare-related material to specifically Ayurvedic pharmaceutical texts, with the taxonomy automatically including appropriate subcategories. The taxonomy also guides synthetic data generation by ensuring comprehensive domain coverage in persona creation and providing structured prompts that elicit domain-appropriate reasoning patterns and knowledge.

### E.3 GOVERNANCE POLICY ENFORCEMENT AND COMPLIANCE

The governance layer operationalizes metadata and lineage into enforceable policies that mitigate legal, ethical, and quality risks while supporting controlled experimentation. We adopt a two-tier design: Apache Ranger manages role-based access, masking, and audit logging at scale, while Open Policy Agent (OPA) enforces fine-grained, context-aware rules. Ranger defines which teams can access which zones, the operations permitted on asset classes, and conditions for data promotion, for example, allowing only quality-approved, license-compliant assets to move from Curated to Feature Store. OPA implements these rules through programmable Rego policies, evaluating decisions at key points such as ingestion, transformation, feature sampling, and artifact publication. This policy-as-code approach enables versioning, automated testing, and auditable logs of all enforcement decisions.

License governance is particularly critical for Indic corpora, which span government releases, copyrighted works under agreements, web scrapes of uncertain status, and user-generated content with varied terms. Each source carries explicit license metadata, with conservative inheritance rules applied across transformations so that derived data inherits the most restrictive license. Policies prevent mixing of incompatible licenses and automatically filter datasets to license-compatible subsets. Compliance reports and audit trails document licensing status and guarantee that downstream models are trained only on permitted materials. Privacy protections address risks from large-scale crawls and user content. Multi-stage PII detection combines pattern matching, named entity recognition, and heuristic filters, triggering policies that range from full exclusion to redaction or metadata flagging for review. Domain-specific handling tailors enforcement: medical records require stricter filtering than news, social media identifiers demand careful redaction, and legal texts balance privacy with preservation of case law.

### E.4 VERSIONING, REPRODUCIBILITY, AND PRODUCTION OPERATIONS

The versioning and reproducibility layer ensures scientific rigor and production reliability for a continuously evolving corpus. Delta Lake provides ACID transactions and time travel, enabling atomic commits, rollbacks, and point-in-time queries that reconstruct historical corpus states. Each modification to curated or feature store data generates a new version capturing both the changes and metadata describing transformation logic, configuration, and execution context. This allows precise reconstruction of any prior corpus state for replication or analysis of how changes affect model behavior. Data Version Control (DVC) extends versioning to the full pipeline, tracking data artifacts alongside code, configuration, and dependencies. Integrated with Git, DVC enables collaborative workflows, branching experiments, and environment reconstruction without storing petabyte-scale data in Git itself. Researchers can retrieve specific pipeline versions, including processing code and corresponding data pointers, facilitating systematic experimentation with OCR heuristics, translation ensembles, and filtering strategies while preserving reproducibility.

Figure 10 illustrates the integrated workflow. Raw data enters via JuiceFS into the Raw zone with minimal processing. Apache Spark ETL pipelines apply cleaning, standardization, and quality filters while emitting OpenLineage events. DataHub builds lineage graphs enriched with processing provenance, and OPA enforces governance policies for promotion to the Curated zone based on quality, license, and domain balance. Curated data supports feature engineering, producing tokenized sequences, instruction tuning pairs, or domain-classified samples. Feature Store assets undergo final validation before training, with policies ensuring balanced sampling, exclusion of low-quality or sensitive material, and license compliance. Delta Lake captures atomic commits at every stage, DVC tracks code and configuration, and Marquez maintains the complete lineage from raw sources to final models. This architecture ensures that every token in MILA's 7.5 trillion token corpus has traceable provenance, verifiable quality, documented licensing, and reproducible processing history, transforming the data collection into a rigorously managed resource suitable for scientific investigation and production deployment. By embedding governance, quality, and reproducibility by design, the pipeline underpins both compliance and the scientific validity of downstream model evaluations.

## F  Human-in-the-Loop Linguistic Validation

A critical insight emerging from our work on MILA is that achieving true linguistic quality for low-resource Indic languages requires moving beyond automated metrics and model-driven evaluation to systematically incorporate expert human judgment throughout the data curation pipeline. While automated evaluation provides essential scalability, enabling assessment of billions of tokens, it fundamentally cannot capture the subtle dimensions of linguistic naturalness, cultural appropriateness, and contextual fidelity that distinguish genuinely high-quality Indic language data from mechanically correct but culturally hollow text. This limitation proves particularly acute for low-resource languages where automated metrics are themselves calibrated on inadequate reference corpora, potentially rewarding outputs that conform to limited training distributions while penalizing linguistically rich variations that fall outside narrow statistical norms. The challenge extends beyond surface-level grammaticality to encompass deeper questions of register appropriateness, dialectal variation, cultural resonance, and the preservation of linguistic features that automated systems—trained predominantly on high-resource languages—may incorrectly flag as errors.

Our approach integrates rigorous human-in-the-loop linguistic validation across all major pipeline components: OCR postprocessing, synthetic data generation, translation, and data distillation. Native language experts and trained linguists evaluate outputs iteratively across multiple complementary dimensions that together capture linguistic quality more comprehensively than any single metric. Fluency assessment examines whether text reads naturally and smoothly without awkward phrasing, unnatural sentence structures, or constructions that betray mechanical generation. Adequacy evaluation verifies that translated or generated content fully captures intended meanings without omissions, additions, or semantic distortions that alter propositional content. Grammar analysis identifies syntactic errors, morphological inconsistencies, and agreement violations that undermine linguistic correctness. Tone assessment ensures that register, formality level, and stylistic choices align appropriately with content type and target audience. Vocabulary richness evaluation measures whether texts employ diverse, expressive lexical choices rather than repetitive, simplified vocabulary characteristic of poor-quality synthetic data. Cultural appropriateness checking identifies content that, while linguistically correct, employs cultural references, examples, or framing inconsistent with Indian contexts. Readability assessment determines whether average speakers of each language can easily comprehend the text without specialized training or unusual linguistic sophistication.

This multidimensional evaluation framework enables identification of failure modes invisible to automated metrics. A translation might achieve high BLEU scores through literal word-for-word correspondence while producing text that native speakers judge unnatural or culturally inappropriate. Conversely, a high-quality translation employing idiomatic expressions and culturally appropriate adaptations might score lower on automated metrics precisely because it deviates from literal correspondence in service of naturalness. By systematically collecting expert judgments across these dimensions, we identify which processing pipelines, model configurations, and postprocessing strategies produce genuinely high-quality outputs for each language rather than merely optimizing automated metrics that may not align with human quality perceptions. Low-quality outputs flagged through this evaluation process are not discarded but rather corrected and reintegrated through iterative refinement loops, with pipelines rerun and configurations adjusted until consistently high scores across all dimensions are achieved for each language and task combination.

### F.1  Quantitative Pipeline Selection Through Human-Calibrated Metrics

The practical value of human-centered evaluation becomes evident when comparing alternative processing pipelines to select optimal configurations for each language. Figure 12 presents a representative case study comparing two translation approaches: Mistral-24B-Instruct (Jiang et al., 2023) versus an ensemble combining IndicTrans2 (Khan et al., 2024) and NLLB (Team et al., 2022) across multiple Indic languages using readability as an illustrative metric. The results reveal dramatic performance heterogeneity across languages: Mistral-24B-Instruct excels for Assamese achieving 84 percent readability, Bengali at 70 percent, and Hindi at 76 percent, while the IndicTrans2-NLLB ensemble demonstrates superior performance for English at 84 percent, Gujarati reaching 98 percent, and Telugu at 88 percent. This language-specific performance variation reflects fundamental differences in training data availability, script complexity, and morphological richness across lan-

guages, validating our decision to employ pipeline selection strategies rather than applying uniform processing to all languages.

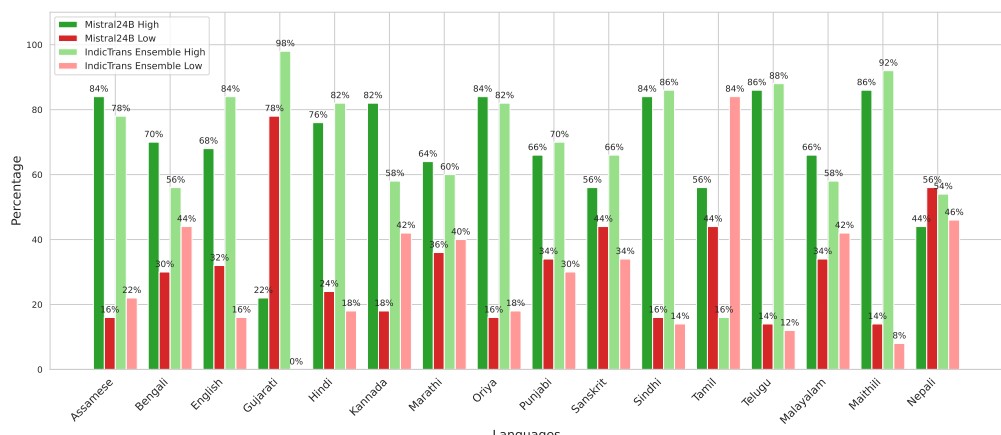

Figure 12: **Readability comparison** between Mistral-24B-Instruct and IndicTrans2-NLLB ensemble across Indic languages, demonstrating language-specific performance heterogeneity that motivates adaptive pipeline selection. Each language achieves optimal results with different model configurations, with Mistral excelling for Assamese, Bengali, and Hindi while the ensemble performs better for English, Gujarati, and Telugu.

Such comparisons guide model selection for each language, ensuring outputs are not only syntactically correct but also culturally and contextually aligned with native speaker expectations. The evaluation process extends well beyond readability scores to encompass comprehensive assessment across all quality dimensions. For OCR outputs, evaluators assess not only character-level accuracy but whether reconstructed text preserves semantic coherence across line breaks, whether ligature decomposition produces linguistically valid sequences, and whether layout analysis correctly identifies reading order in multi-column documents with embedded figures. For translation, assessment examines whether target language outputs preserve subtle pragmatic meanings, maintain appropriate register and formality levels, and employ vocabulary natural to the domain rather than literal dictionary translations that sound stilted. For synthetic data generation, evaluation verifies that persona-driven outputs genuinely reflect Indian cultural contexts rather than superficially adapted Western content, that reasoning patterns align with domain-specific norms within Indian professional and academic contexts, and that generated examples employ culturally appropriate scenarios and references.

The scoring process operates through iterative refinement loops where low-quality outputs are not merely flagged but actively corrected by linguists, with corrections fed back to improve processing pipelines. When evaluation reveals systematic errors, such as consistent mistranslation of domain-specific terminology, inappropriate register choices for particular content types, or recurring grammatical patterns characteristic of mechanical translation, these patterns inform targeted improvements to models, prompts, and postprocessing rules. Multiple evaluation rounds continue until all metrics consistently meet high standards, with each iteration incorporating lessons from previous rounds to progressively elevate quality. This ensures that each language and task combination ultimately leverages a specialized, validated pipeline preserving linguistic integrity, cultural context, and factual fidelity rather than accepting mediocre outputs from generic processing approaches. The result is a corpus where quality is not assumed based on automated metrics but actively verified and refined through expert judgment, with continuous improvement driven by systematic analysis of failure modes and targeted interventions addressing identified weaknesses.

## F.2 STRUCTURED EVALUATION PROTOCOLS AND CRITERIA STANDARDIZATION

The effectiveness of human evaluation depends critically on providing evaluators with clear, standardized criteria and systematic protocols that ensure consistency across annotators, languages, and evaluation rounds. We developed comprehensive evaluation guidelines that operationalize abstract quality dimensions into concrete assessment procedures with explicit decision rules and illustrative

examples. Evaluators receive detailed instructions specifying how to identify and categorize different error types, what severity levels to assign based on impact on comprehensibility and naturalness, and how to provide actionable feedback enabling targeted corrections. This standardization proves essential for maintaining evaluation reliability as the project scales across 16 languages with different evaluator teams, ensuring that a score of 4 out of 5 for fluency carries consistent meaning whether applied to Assamese translations or Telugu synthetic data.

| Evaluation Criterion | Description |
|---|---|
| Fluency & Readability | Does the translation read naturally and smoothly in your language, without awkward phrasing, unnatural sentence structures, or grammatical errors? |
| Adequacy & Meaning Preservation | Does the translated text fully capture the meaning of the original English sentence without omitting, adding, or distorting information? |
| Use of Rich & Appropriate Vocabulary | Does the translation use a rich and diverse vocabulary that feels natural and expressive in your language? |
| Cultural & Contextual Appropriateness | Are there any cultural inconsistencies, unnatural phrases, or word choices that feel out of place or confusing in your language? |
| Grammar & Sentence Structure | Are the grammar, syntax, and sentence structures well-formed and correct in your language? |
| Consistency & Tone Matching | Does the translation maintain the same tone, formality, and style as the original English text? |
| Readability & Ease of Understanding | Is the translation easy to read and understand for an average speaker of your language? |

Table 28: Sample Translation Quality Evaluation Criteria with Descriptions

The evaluation protocol structures assessment around clearly defined, answerable questions for each quality dimension. As illustrated in Table 28, for instance, fluency and readability are evaluated by asking whether the translation reads naturally and smoothly in the target language, without awkward phrasing, unnatural sentence structures, or grammatical errors. Ratings are then assigned on a scale from 1 (very unnatural) to 5 (perfectly natural). Similar rating procedures are applied across all other evaluation criteria, and the resulting scores are subsequently used to provide feedback for enhancing the translation pipeline.

Beyond numerical ratings, evaluators provide critical written feedback detailing specific issues identified in assessed samples. This qualitative feedback proves invaluable for diagnosing systematic problems and guiding targeted improvements. Representative feedback examples from our evaluation process illustrate the types of insights human judgment provides that automated metrics miss entirely. One evaluator noted that "Hindi translation is not proper, uses overly formal Sanskritized vocabulary inappropriate for the conversational tone of the source text," identifying a register mismatch invisible to most automated metrics. Another flagged content as "not depicting true picture, contains anti-national sentiment," catching politically sensitive framing that requires cultural knowledge to identify. Concerns about regional bias appeared in feedback like "North-South divide should be avoided" and "Why mention a particular state? Shows regional bias," ensuring generated examples don't inadvertently reinforce stereotypes. Terminology choices received scrutiny: "Instead of 'bhedbhaav' it should be 'indifference', passage tone should suggest solutions rather than expressing anger, needs softer and more polite terminology." Factual accuracy checking emerged in feedback such as "Doesn't seem to be reality—fact check percentage cited" and "Are these statistics verified?" Authenticity concerns surfaced in notes like "Indian name pronunciation issues—proper authentic usage of Indian vocabulary should be present."

This feedback directly informs pipeline improvements through systematic categorization and analysis. Common feedback patterns indicate where models consistently struggle—such as inappropriate register choices, regional bias, or factual inaccuracies—enabling targeted interventions. For translation pipelines, feedback revealing consistent terminology issues for specific domains motivates development of domain-specific glossaries and constraints. For synthetic generation, feedback identifying cultural inappropriateness guides refinement of persona specifications and generation

prompts to better capture Indian contexts. The evaluation framework implements a zero-data-loss policy where low-quality data is corrected and updated based on feedback rather than simply discarded, ensuring continuous improvement while maximizing the value extracted from expensive human annotation. Multiple evaluation rounds with iterative refinement continue until outputs consistently achieve high scores across all dimensions, with each iteration incorporating lessons from previous feedback to progressively eliminate failure modes.

### F.3    ADDRESSING DIALECTAL VARIATION AND PRACTICAL USABILITY

Standard evaluation approaches fall short when targeting non-urban, monolingual populations who speak dialects diverging from formal standard varieties. Evaluations conducted by native speakers of standard dialects—typically taught in schools and represented in digital corpora, do not reflect whether dialect speakers of varying literacy levels can understand or use model outputs. This limitation is especially consequential for agricultural advisory systems, where users predominantly speak regional dialects with vocabulary, pronunciation, and grammatical features absent from standard Hindi. A system generating flawless standard Hindi may be incomprehensible or culturally alien to a Bhojpuri-speaking farmer, whereas outputs incorporating dialectal features might score lower on standard metrics precisely because they deviate from formal norms. Indic languages exhibit profound dialectal diversity that conventional corpora and evaluation methods erase. Hindi alone varies dramatically across regions: Haryanvi, Punjabi-influenced Hindi, Bihari, and Jharkhandi dialects differ in vocabulary, morphology, and syntax, particularly for everyday agricultural objects and practices. Folk terms for vegetables, clothing, tools, and farming processes carry rich pragmatic and cultural associations, learned orally rather than through formal education. Standard language models trained primarily on written corpora lack this dialectal vocabulary and cultural grounding. Even when dialectal text is included, the dominance of standard forms causes models to favor these over less frequent dialectal alternatives, creating a disconnect between model outputs and the needs of rural users.

For example, most farmers in target demographics are illiterate or semi-literate, with even literate individuals preferring folk vocabulary over standard language in practical agricultural discussions. Furthermore, farmers express strong preference for speech-based interaction over text, reflecting both literacy constraints and the practical reality that speech is more natural and efficient for real-time agricultural decision-making. Text-based systems, regardless of linguistic quality, exclude large portions of the target population, while speech interfaces employing dialectal vocabulary and natural prosody enable genuine accessibility. To address this gap, we propose targeted evaluation assessing dialectal appropriateness and practical usability for non-urban populations, starting with pilot experiments before broader deployment. The pilot focuses on agriculture and two Hindi dialects, including Bhojpuri, chosen for its wide geographic spread, rich folk vocabulary, and large farming population. Native Bhojpuri-speaking agricultural workers serve as evaluators, judging whether model outputs use vocabulary, grammar, and cultural references natural to their variety rather than imposing formal Hindi.

Evaluation extends beyond standard quality dimensions to capture real-world usability. Vocabulary naturalness considers folk terms for crops, tools, and processes. Cultural resonance ensures examples and scenarios reflect rural lived experience. Comprehensibility for low-literacy users evaluates sentence structures and discourse organization suitable for limited formal education. Speech interface suitability examines whether outputs would sound natural when rendered orally with local pronunciation and prosody. These criteria complement standard measures of grammar and factual accuracy. Beyond agriculture, dialectal evaluation redefines linguistic quality for low-resource language technology. Conventional frameworks privilege formal, written varieties, marginalizing dialects spoken by millions. Incorporating dialectal variation fosters inclusive systems that respect linguistic diversity and folk registers systematically erased by standard corpora and automated metrics. Human-in-the-loop evaluation thus becomes essential for ensuring MILA and related systems reflect India's full linguistic richness.

Crucially, building a high-quality Indic multilingual dataset relied on rigorous human-in-the-loop validation across OCR, synthetic generation, translation, and data distillation. Native experts iteratively evaluated fluency, adequacy, grammar, tone, vocabulary richness, cultural appropriateness, and readability. Low-quality outputs were corrected and reintegrated, with pipelines rerun until consistently high scores were achieved, ensuring optimal quality for every language and task.

# G INDIC MMLU SCORES BY LANGUAGE

We benchmarked every tier of frontier open-source models on the Indic MMLU. The language-wise results are shown below: Assamese (Figure 13a), Bengali (Figure 13b), English (Figure 14a), Gujarati (Figure 14b), Hindi (Figure 15a), Kannada (Figure 15b), Maithili (Figure 16a), Malayalam (Figure 16b), Marathi (Figure 17a), Nepali (Figure 17b), Oriya (Figure 18a), Punjabi (Figure 18b), Sanskrit (Figure 19a), Sindhi (Figure 19b), Tamil (Figure 20a), and Telugu (Figure 20b). Each float contains two stacked images with spacing between them, so the next float continues naturally on the next page.

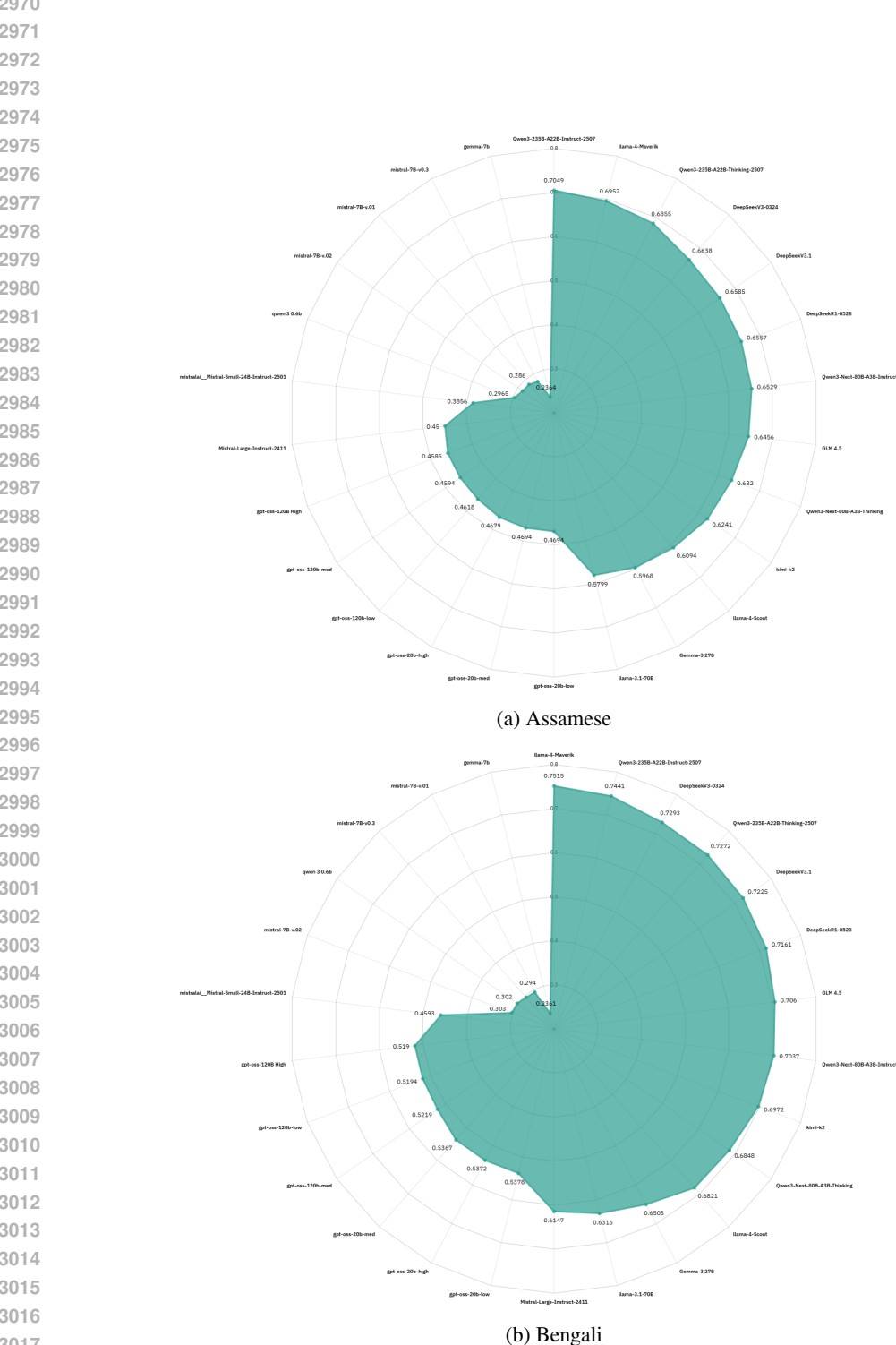

(a) Assamese

(b) Bengali

Figure 13: Indic MMLU scores by language: a) Assamese; b) Bengali.

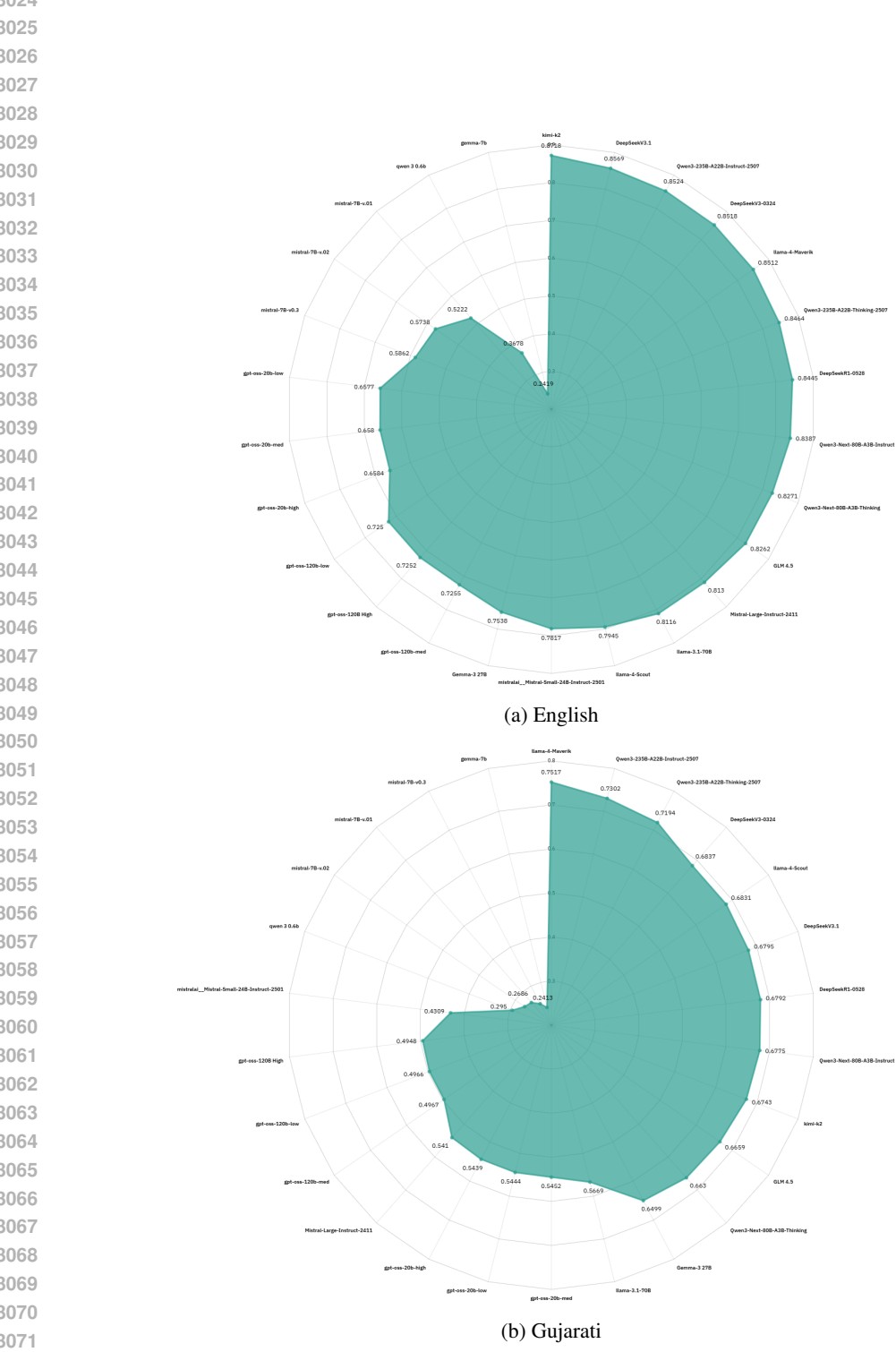

(a) English

(b) Gujarati

Figure 14: c) English; d) Gujarati.

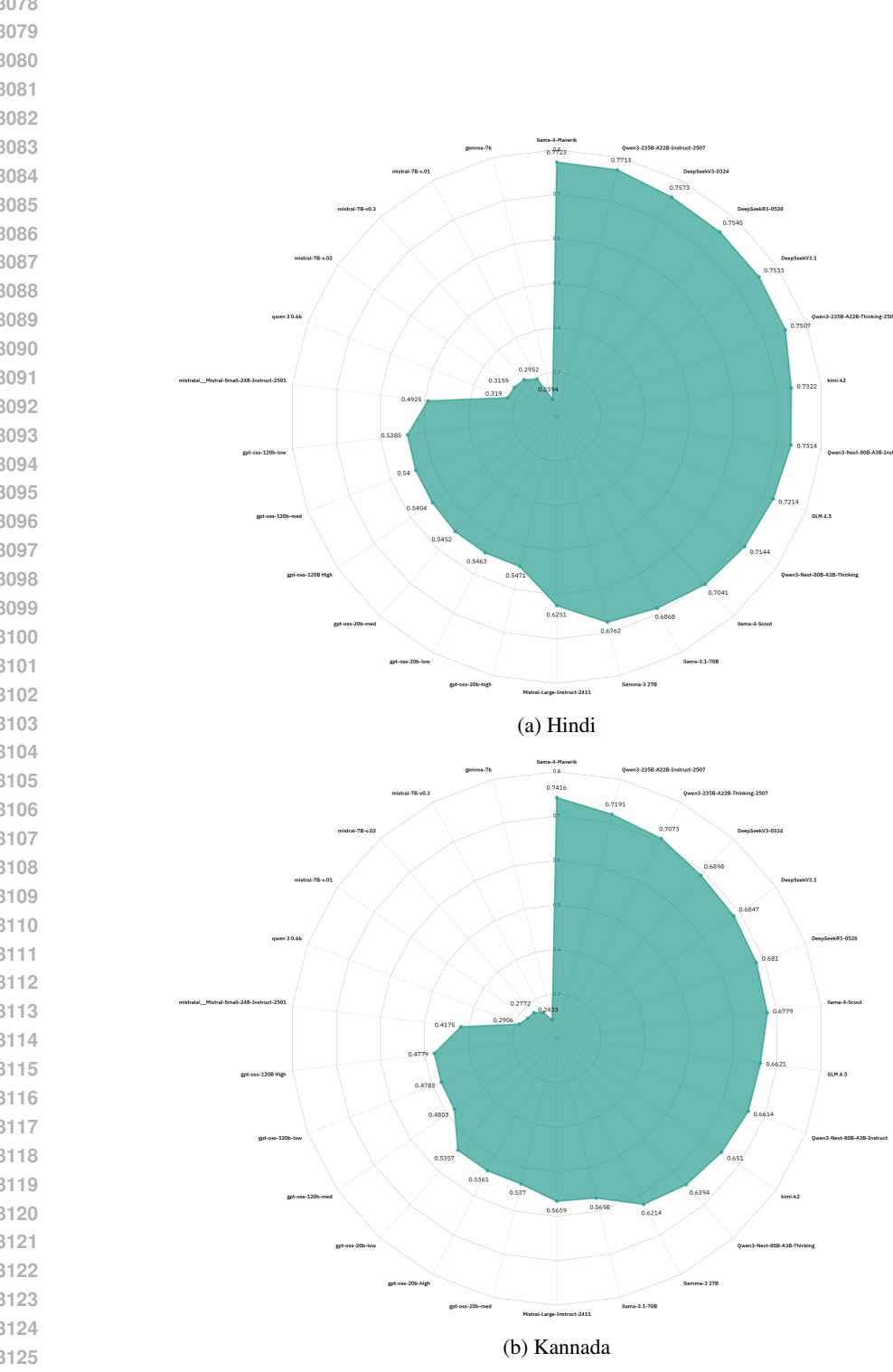

(a) Hindi

(b) Kannada

Figure 15: e) Hindi; f) Kannada.

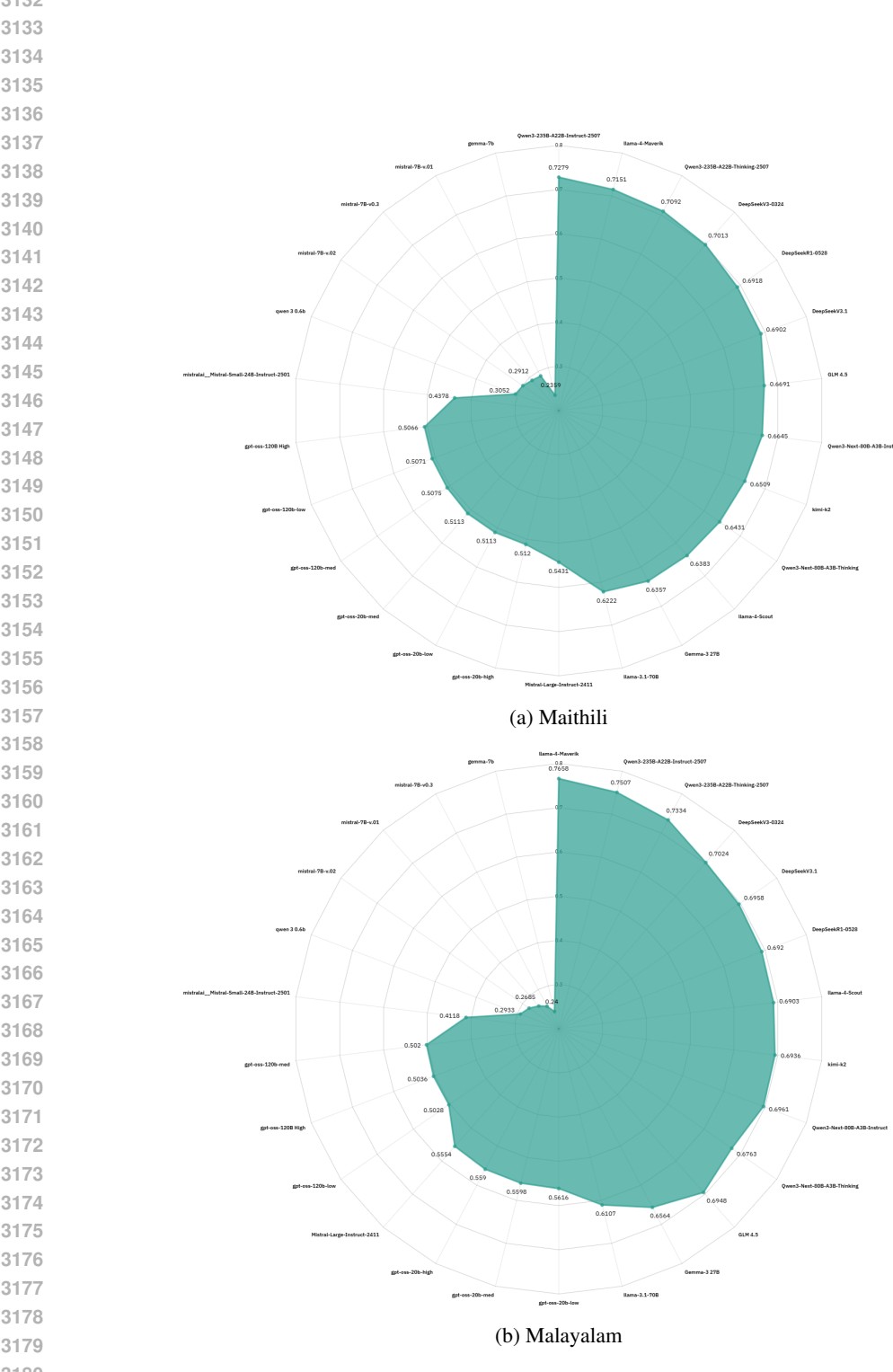

(a) Maithili

(b) Malayalam

Figure 16: g) Maithili; h) Malayalam.

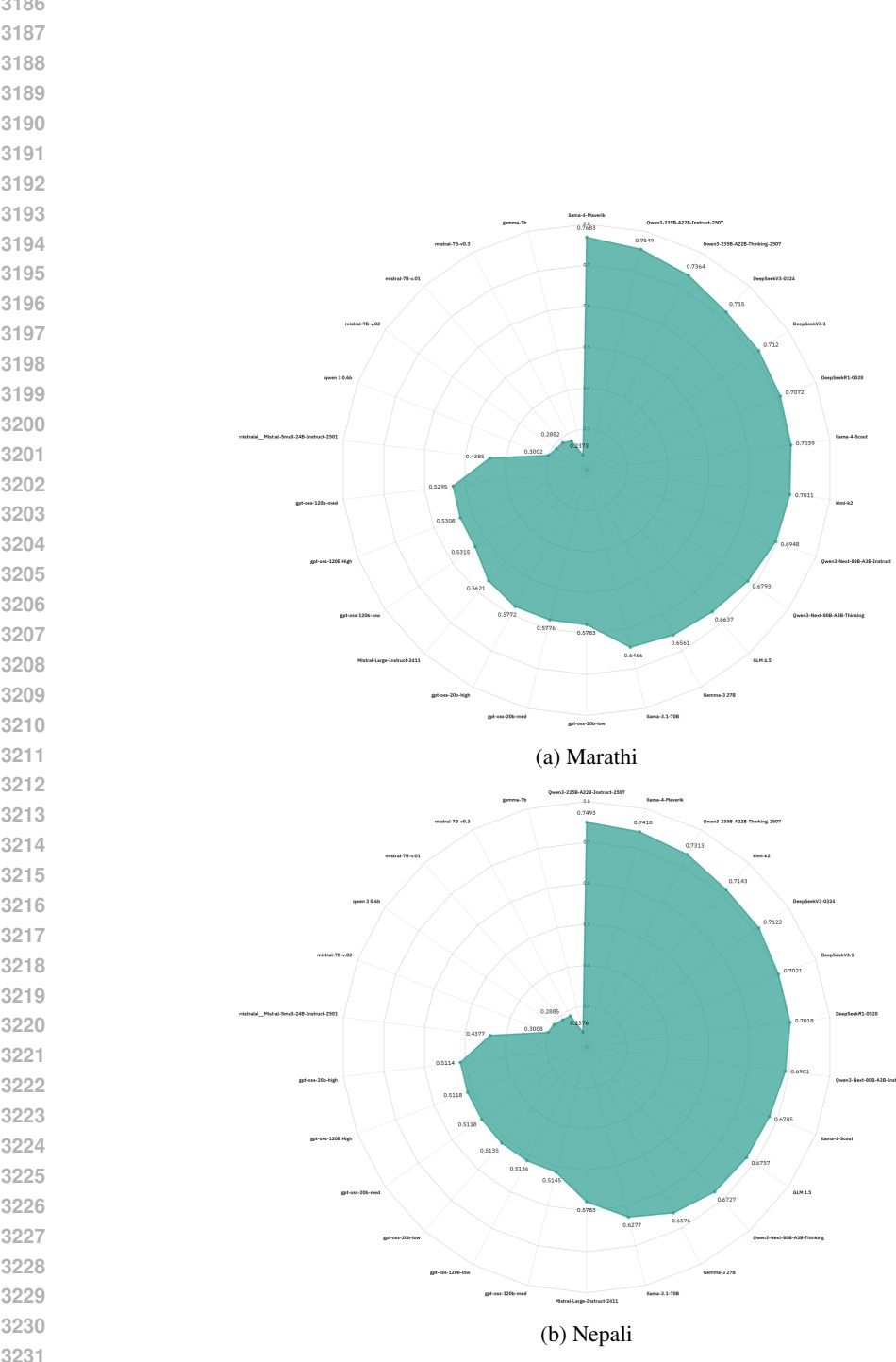

(a) Marathi

(b) Nepali

Figure 17: i) Marathi; j) Nepali.

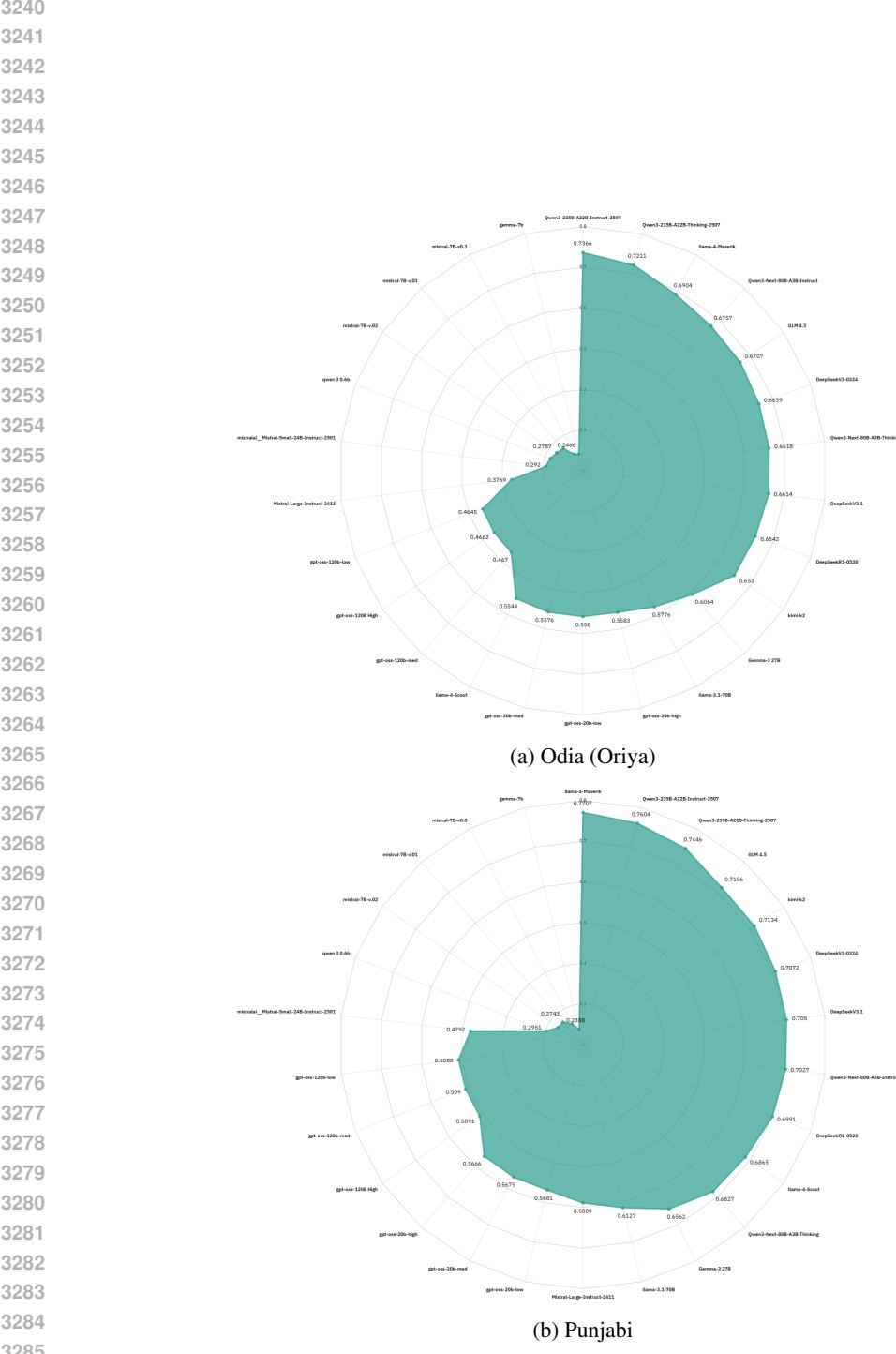

(a) Odia (Oriya)

(b) Punjabi

Figure 18: k) Oriya; l) Punjabi.

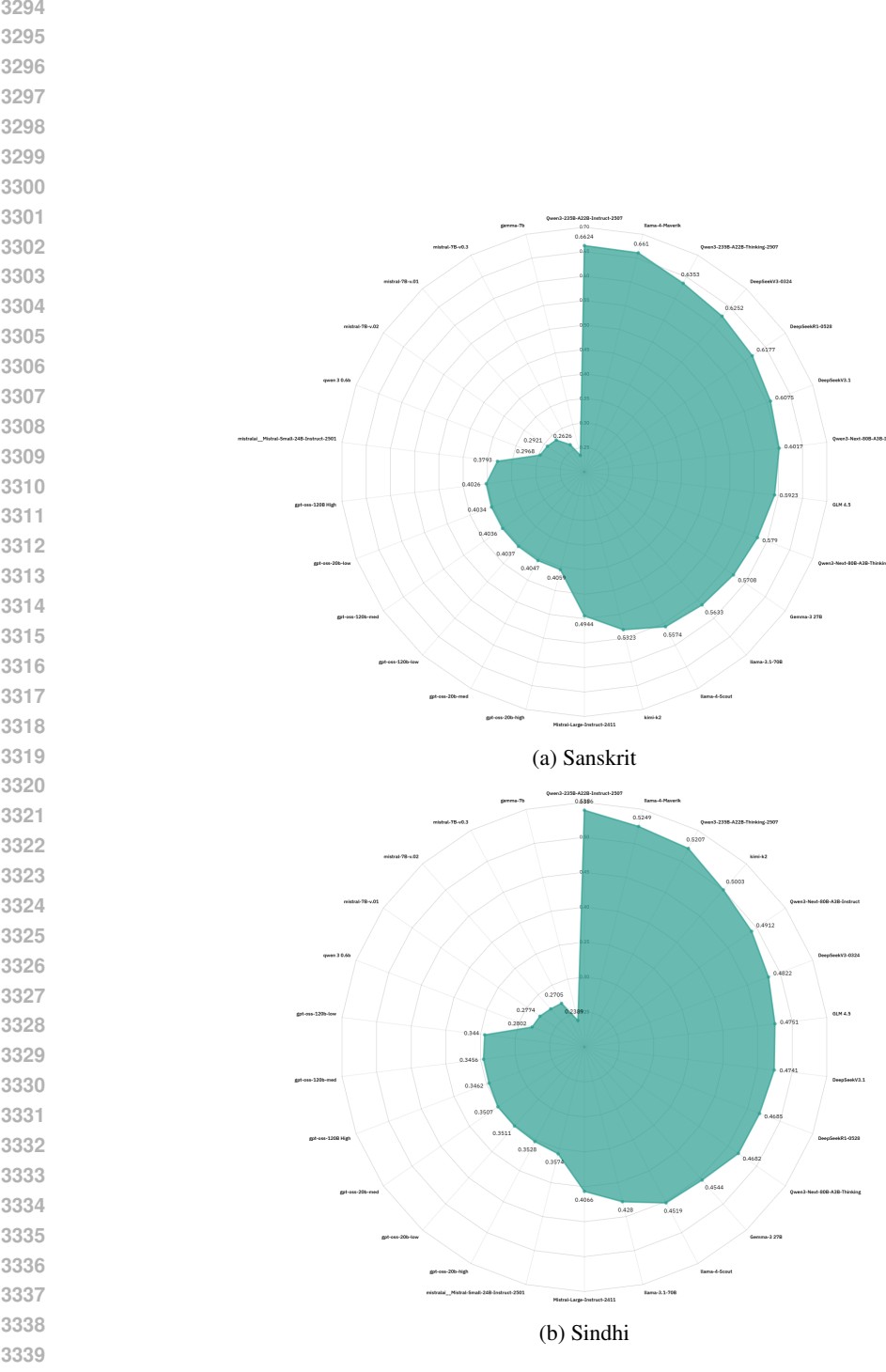

(a) Sanskrit

(b) Sindhi

Figure 19: m) Sanskrit; n) Sindhi.

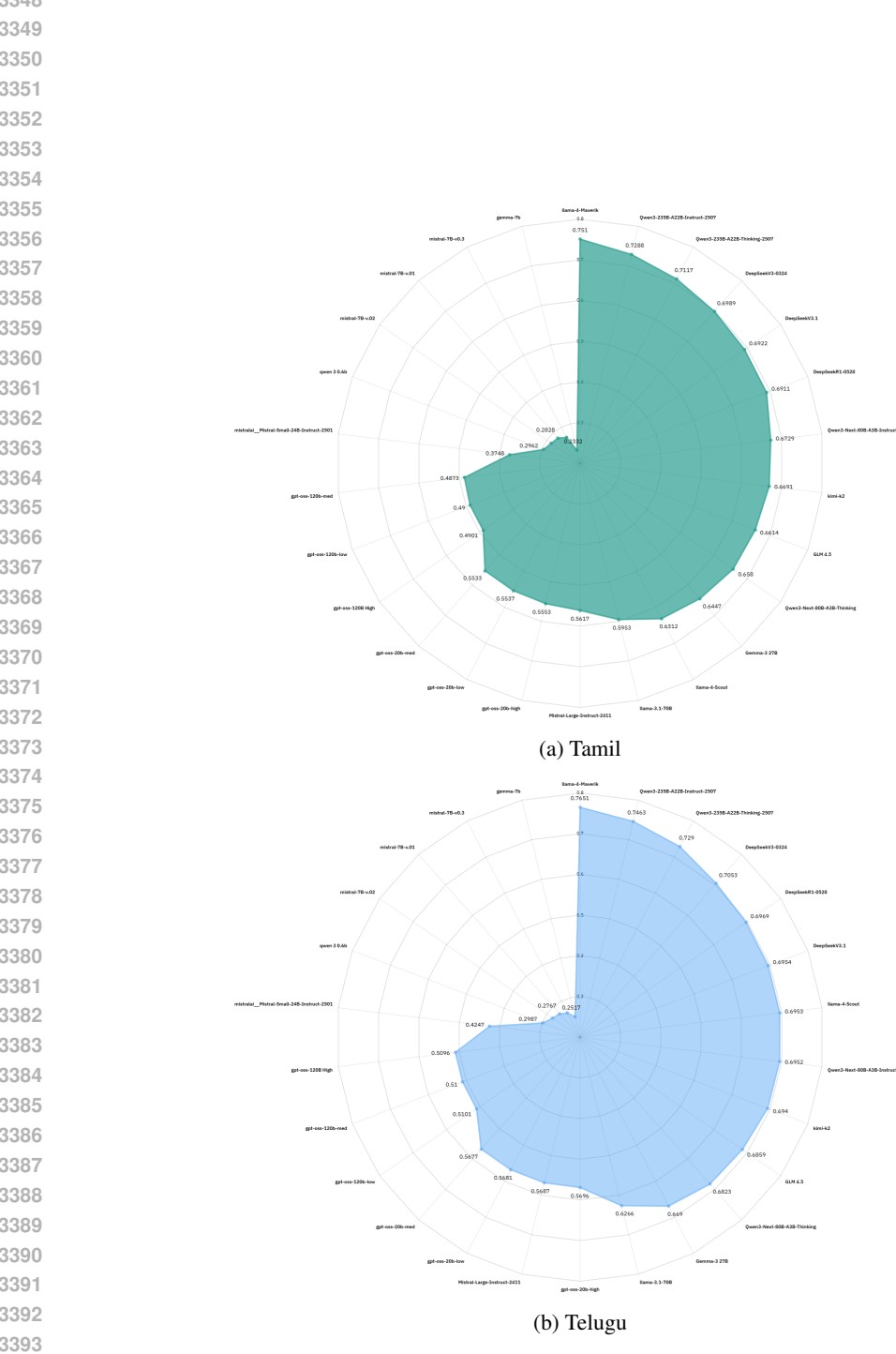

(a) Tamil

(b) Telugu

Figure 20: o) Tamil; p) Telugu.

# H  TRANSLATION BENCHMARK RESULTS

## H.1  EVALUATION OF BASELINE MT AND LLMS ON INDIC LANGUAGES

We evaluated the general translation capabilities of current open-source models on Indic languages using **ai4bharat/IN22-Gen** and **google/IndicGenBench_flores_in**. Both baseline MT systems and large language models (LLMs) were tested using CHRF, CHRF++, and SACREBLEU metrics.

Results show that LLMs generally provide more fluent and context-aware translations, especially for morphologically rich languages, while baseline MT models perform well for high-resource languages but lag on low-resource or complex languages. Performance varies across language pairs, highlighting the uneven support for Indic languages in current open-source models.

### H.1.1  RESULTS FOR AI4BHARAT/IN22-GEN

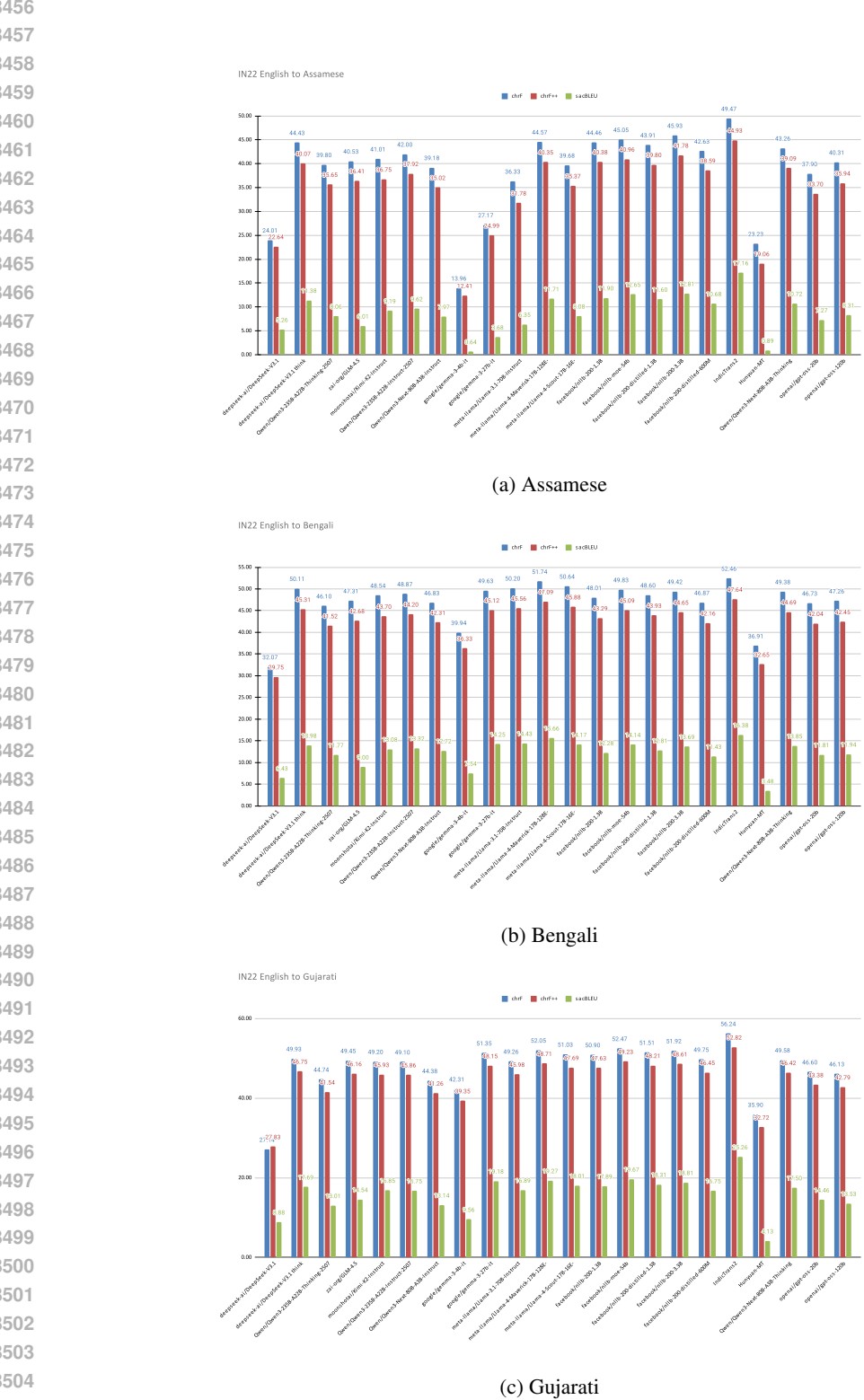

Figure 21: Evaluation of **ai4bharat/IN22-Gen** across Assamese, Bengali, and Gujarati. Standalone open-source LLM and MT outputs (no ensembles or post-processing); metrics: chrF, chrF++, sacre-BLEU. — Part I

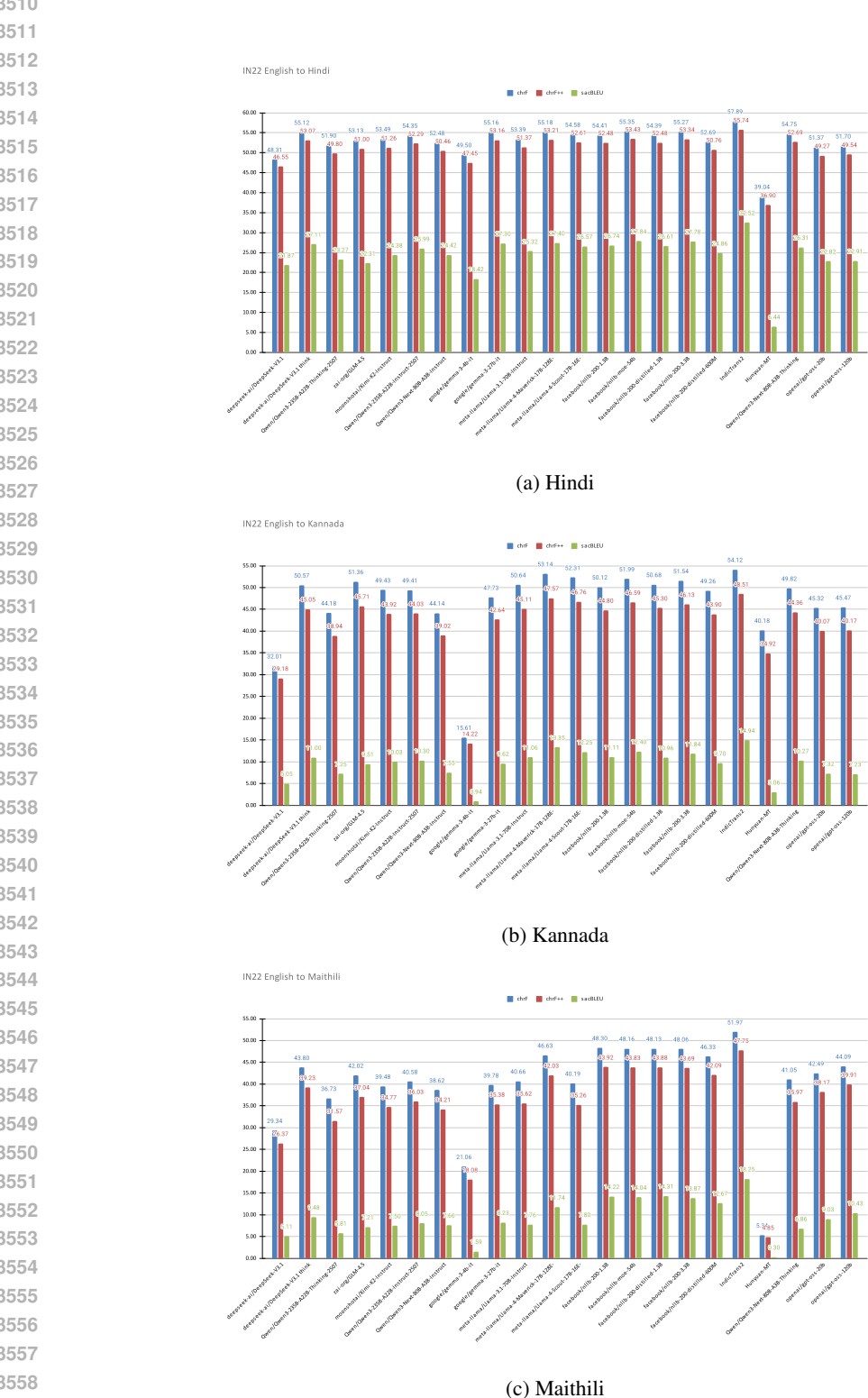

(a) Hindi

(b) Kannada

(c) Maithili

Figure 22: Evaluation of **ai4bharat/IN22-Gen** across Hindi, Kannada, and Maithili. Standalone open-source LLM and MT outputs (no ensembles or post-processing); metrics: chrF, chrF++, sacre-BLEU. — Part II

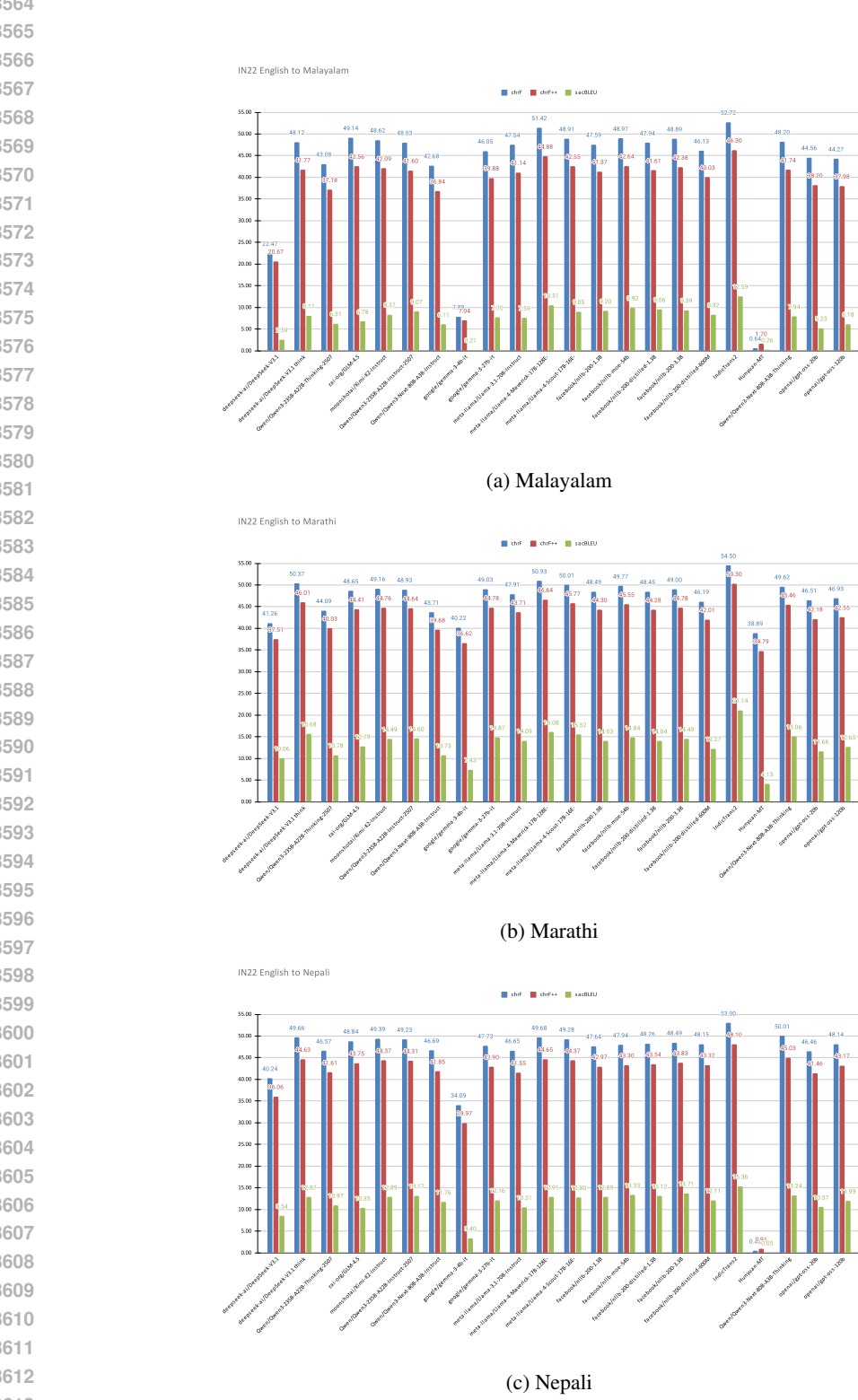

(a) Malayalam

(b) Marathi

(c) Nepali

Figure 23: Evaluation of **ai4bharat/IN22-Gen** across Malayalam, Marathi, and Nepali. Standalone open-source LLM and MT outputs (no ensembles or post-processing); metrics: chrF, chrF++, sacre-BLEU. — Part III

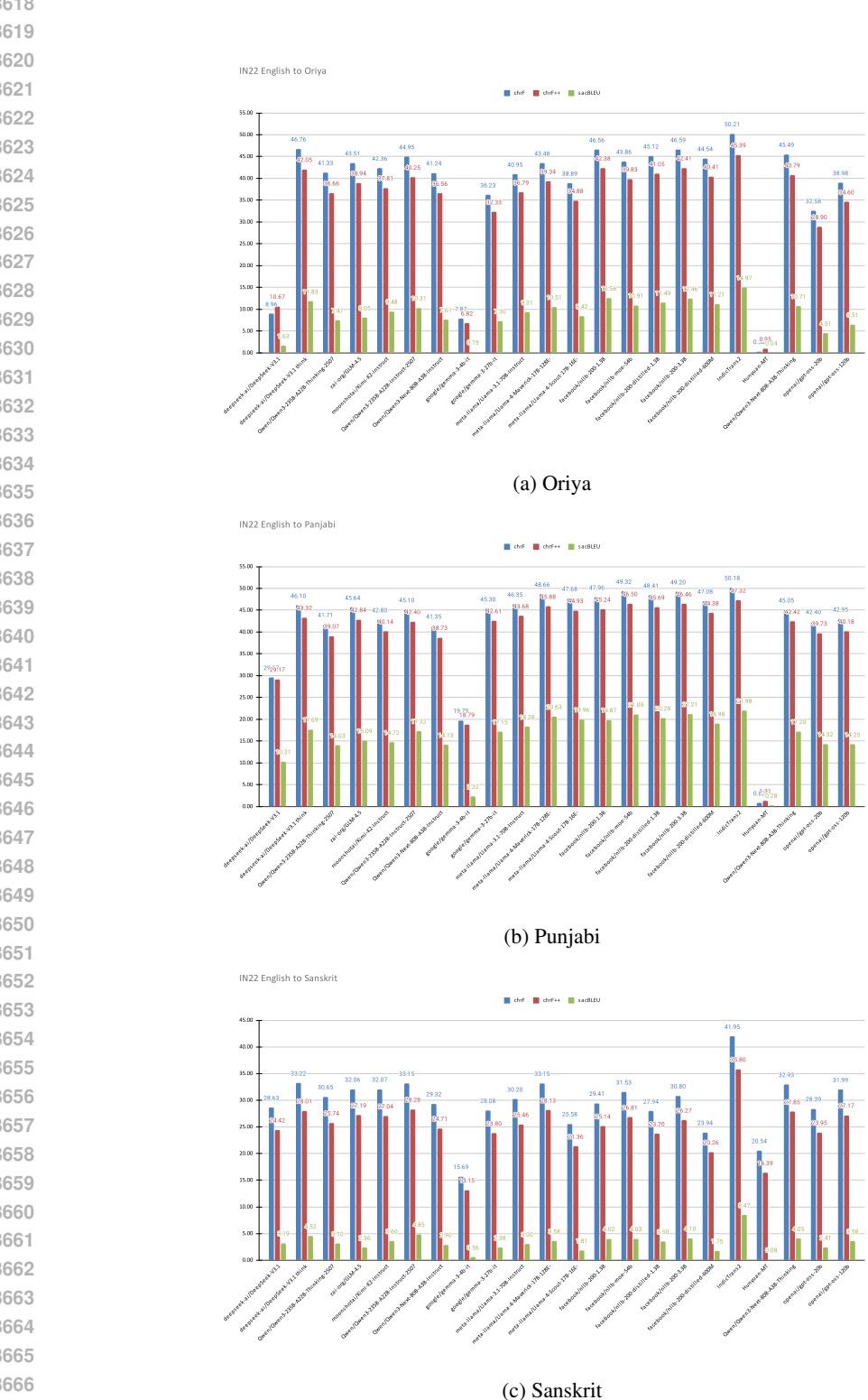

Figure 24: Evaluation of **ai4bharat/IN22-Gen** across Oriya, Punjabi, and Sanskrit. Standalone open-source LLM and MT outputs (no ensembles or post-processing); metrics: chrF, chrF++, sacre-BLEU. — Part IV

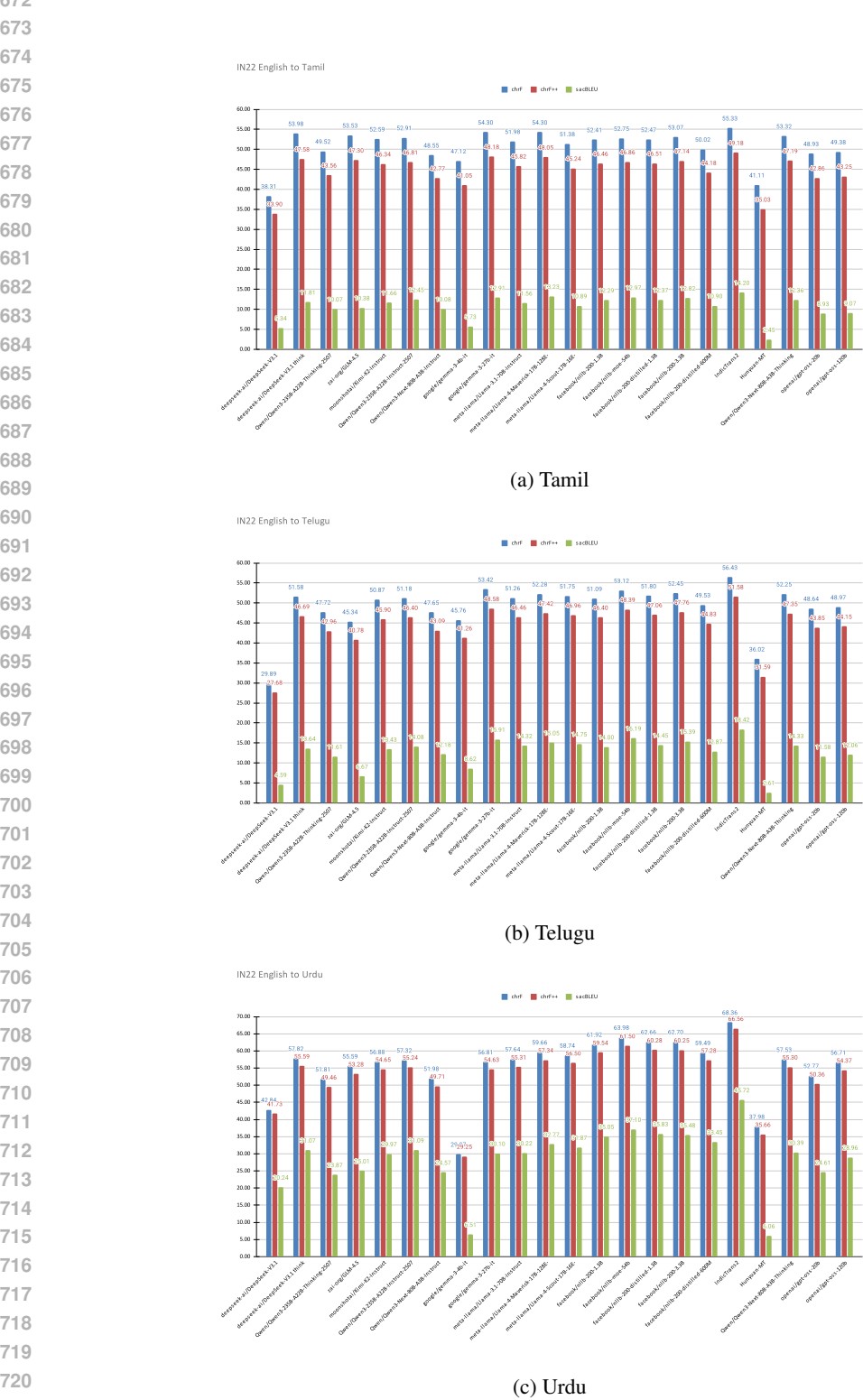

(a) Tamil

(b) Telugu

(c) Urdu

Figure 25: Evaluation of **ai4bharat/IN22-Gen** across Tamil, Telugu, and Urdu. Standalone open-source LLM and MT outputs (no ensembles or post-processing); metrics: chrF, chrF++, sacreBLEU. — Part V

### H.1.2 RESULTS FOR GOOGLE/INDICGENBENCH_FLORES_IN

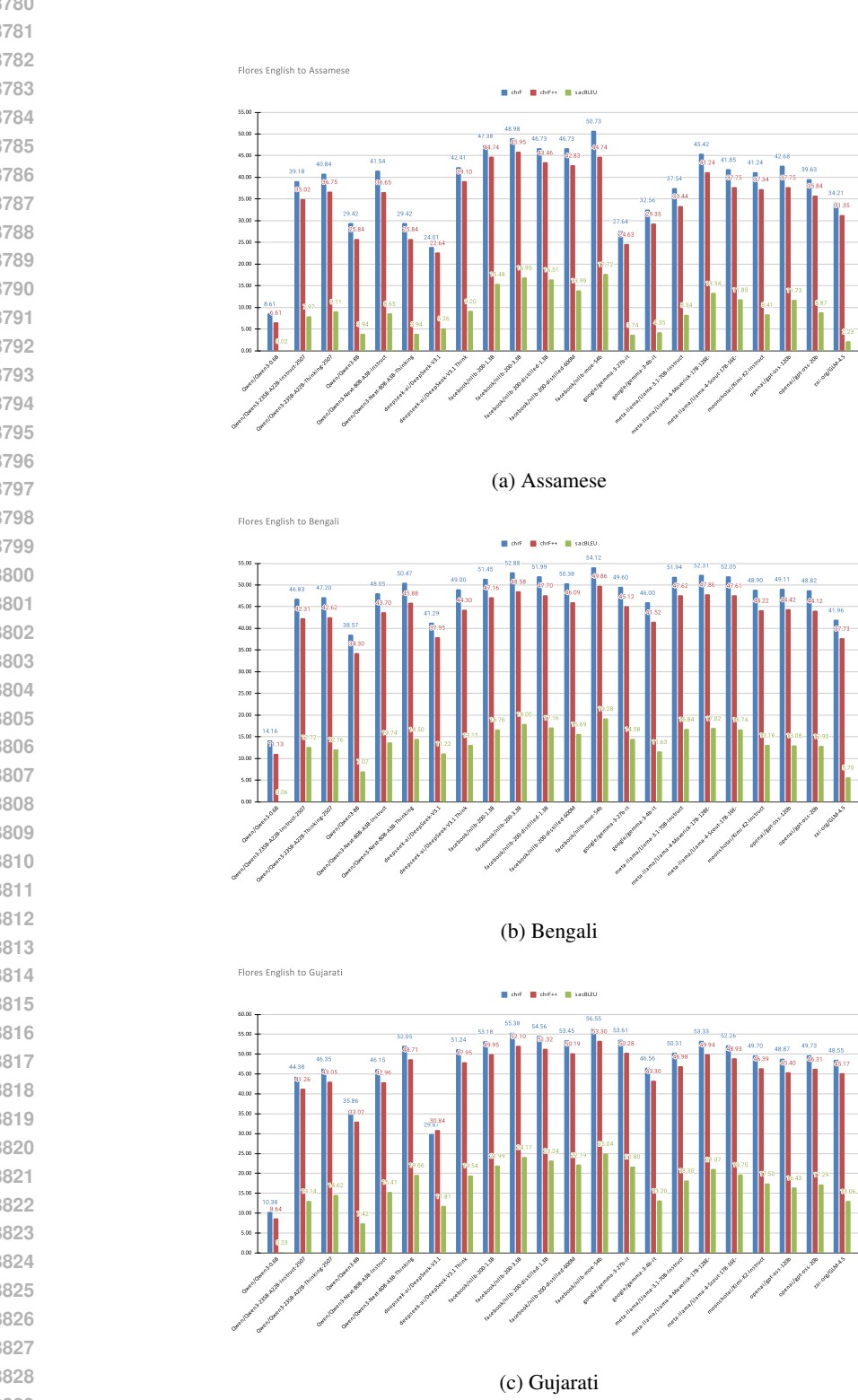

(a) Assamese

(b) Bengali

(c) Gujarati

Figure 26: Evaluation of **google/IndicGenBench_flores_in** across Assamese, Bengali, and Gujarati. Standalone open-source LLM and MT outputs (no ensembles or post-processing); metrics: chrF, chrF++, sacreBLEU. — Part I

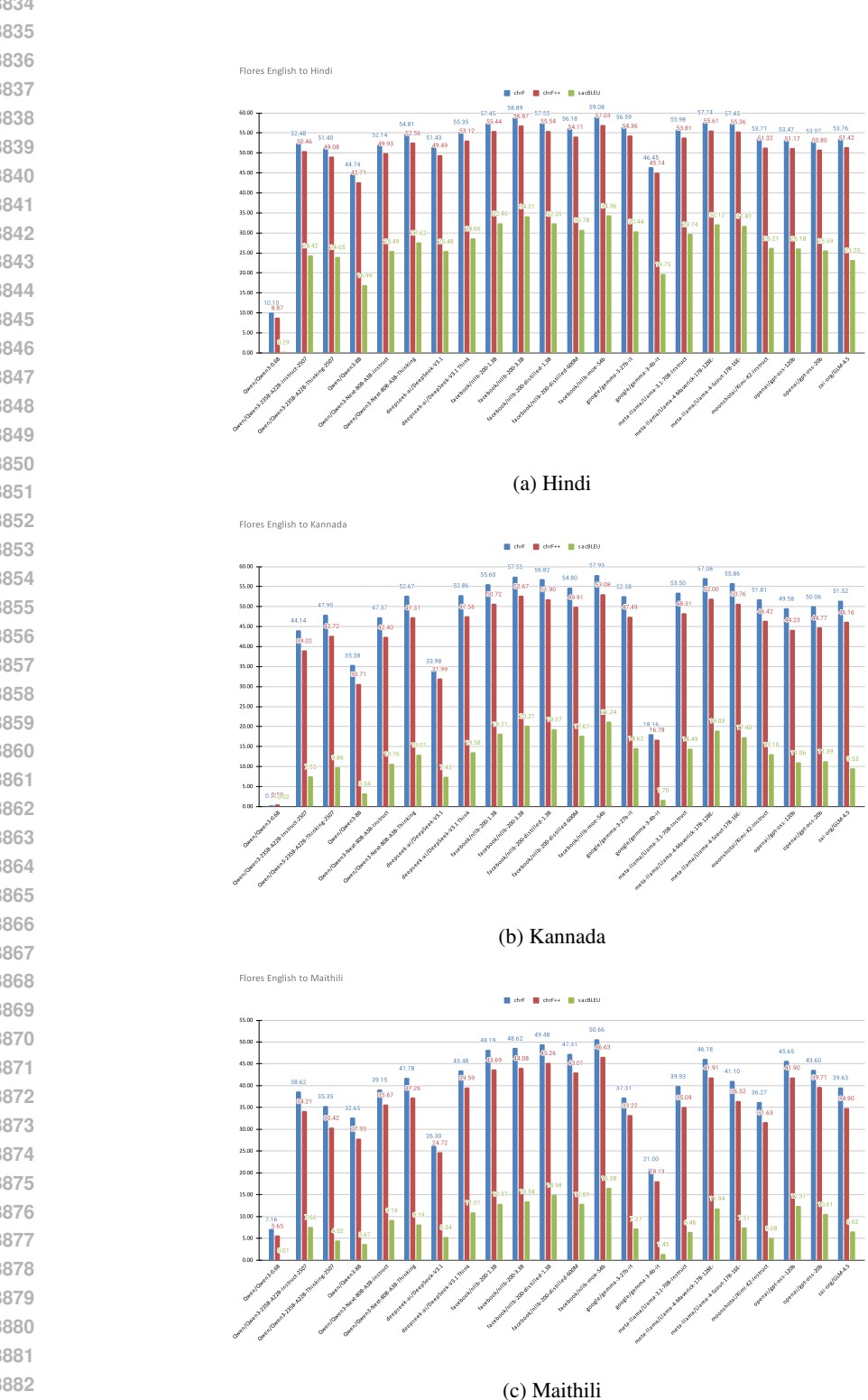

(a) Hindi

(b) Kannada

(c) Maithili

Figure 27: Evaluation of **google/IndicGenBench_flores_in** across Hindi, Kannada, and Maithili. Standalone open-source LLM and MT outputs (no ensembles or post-processing); metrics: chrF, chrF++, sacreBLEU. — Part II

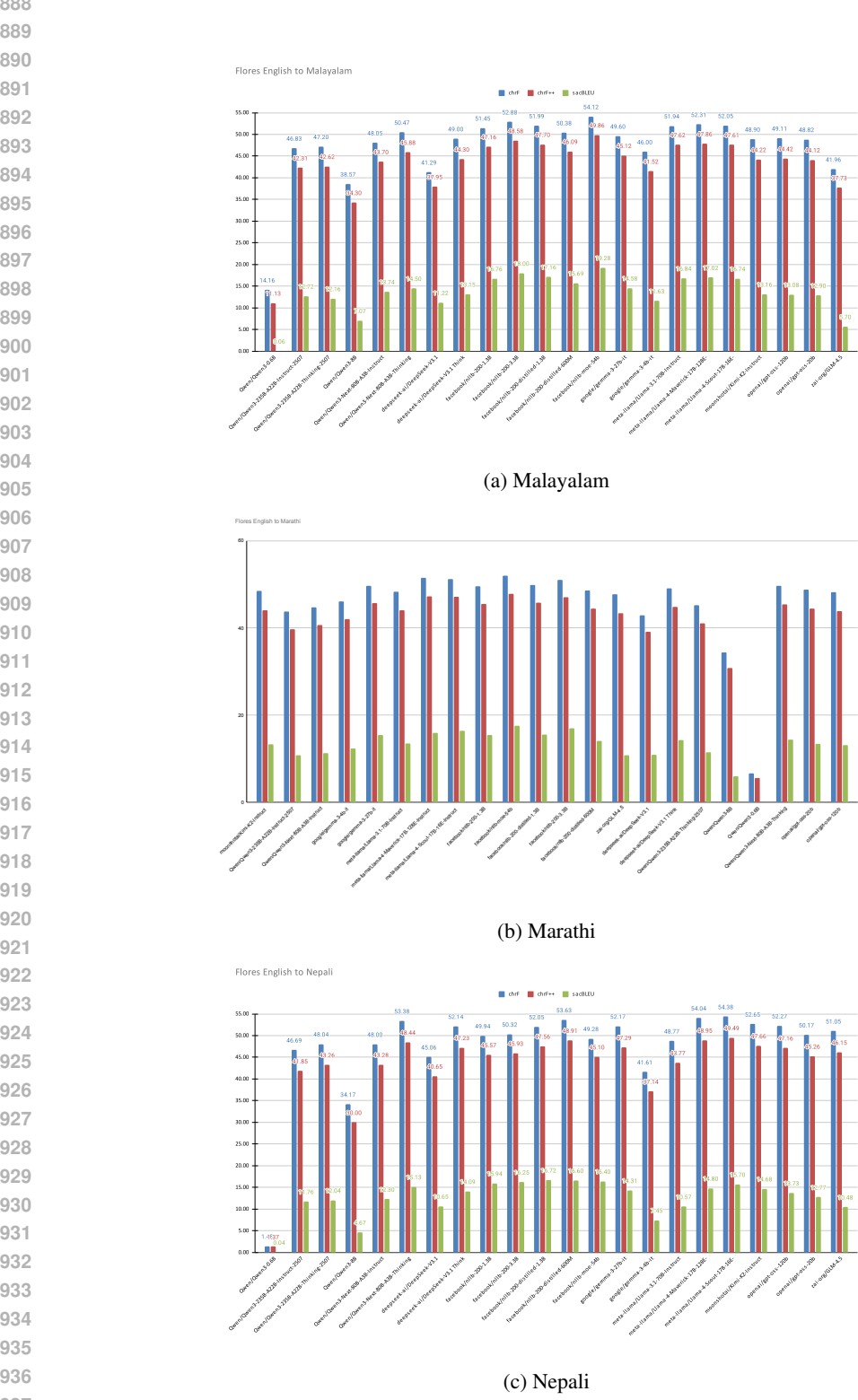

(a) Malayalam

(b) Marathi

(c) Nepali

Figure 28: Evaluation of **google/IndicGenBench_flores_in** across Malayalam, Marathi, and Nepali. Standalone open-source LLM and MT outputs (no ensembles or post-processing); metrics: chrF, chrF++, sacreBLEU. — Part III

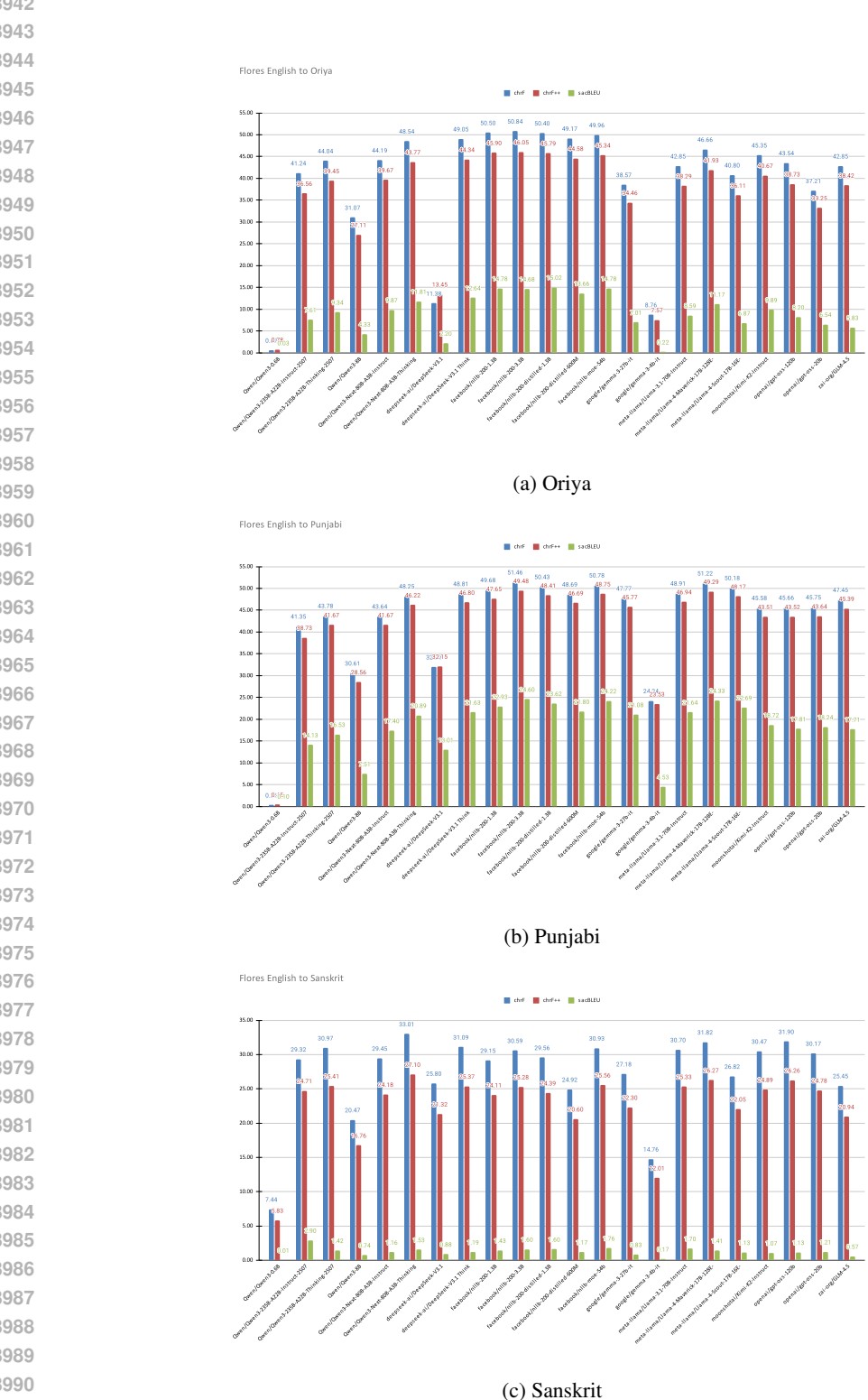

Figure 29: Evaluation of **google/IndicGenBench_flores_in** across Oriya, Punjabi, and Sanskrit. Standalone open-source LLM and MT outputs (no ensembles or post-processing); metrics: chrF, chrF++, sacreBLEU. — Part IV

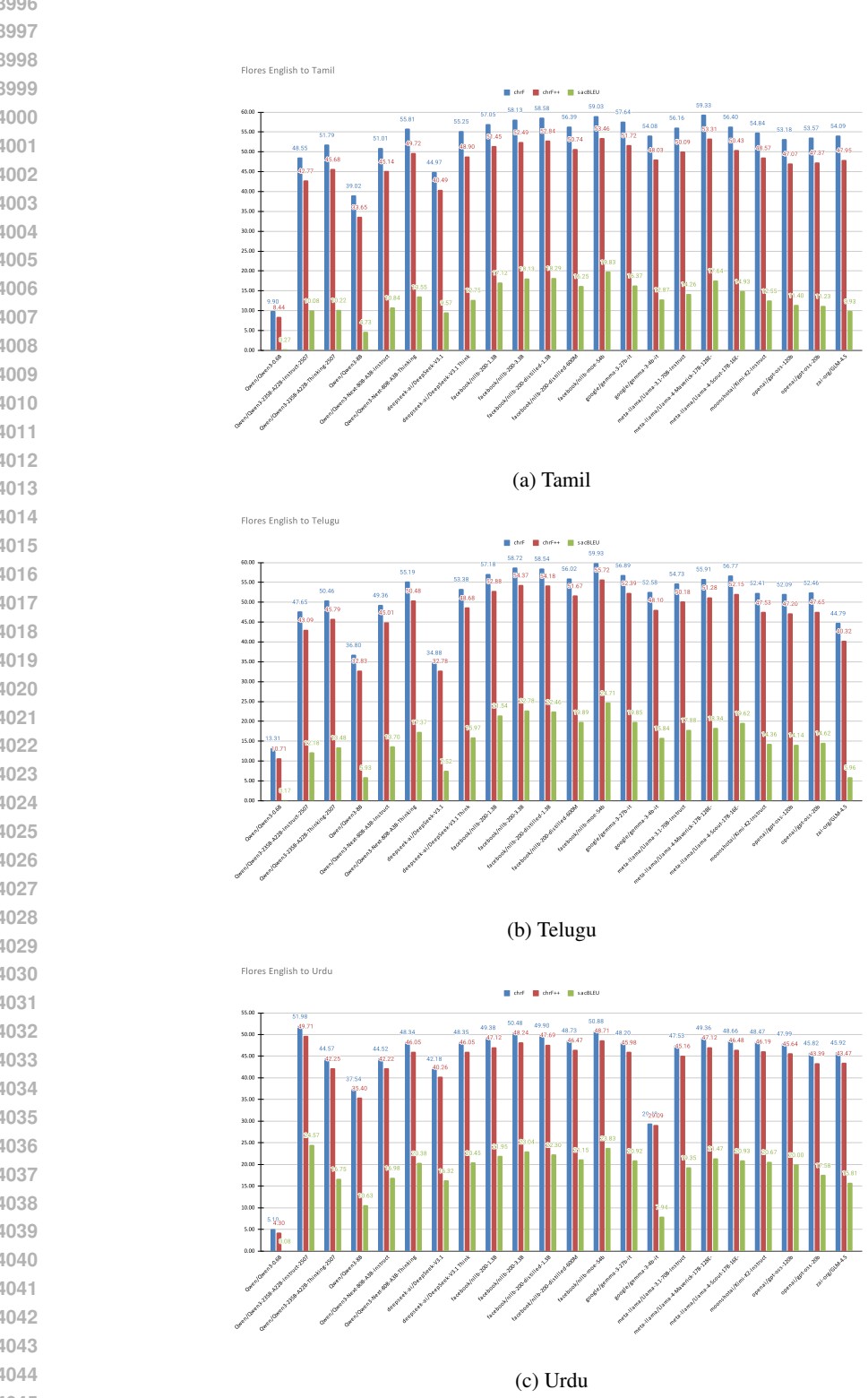

(a) Tamil

(b) Telugu

(c) Urdu

Figure 30: Evaluation of **google/IndicGenBench_flores_in** across Tamil, Telugu, and Urdu. Standalone open-source LLM and MT outputs (no ensembles or post-processing); metrics: chrF, chrF++, sacreBLEU. — Part V

## H.2 EVALUATION OF THE ENSEMBLE (MT + LLM) FOR INDIC LANGUAGES

We evaluated an ensemble translation pipeline that employs IndicTrans2 as the primary MT system and leverages open-source LLMs (chosen based on strong Indic MMLU performance) for targeted post-editing. In this setup, the MT output provides a lexical and structural scaffold, while the LLM performs contextual disambiguation, corrects inflection and agreement errors, and improves overall fluency. The experiments use the same test sets as the baseline evaluation (**ai4bharat/IN22-Gen** and **google/IndicGenBench_flores_in**) and apply CHRF, CHRF++, and SACREBLEU for direct comparability.

The results show that the IndicTrans2 + LLM ensemble consistently enhances contextual fluency and reduces overly literal or forced translations when compared to standalone IndicTrans2. The most substantial improvements occur in morphologically rich and low-resource Indic languages. For high-resource language pairs, the ensemble yields modest but reliable fluency gains while maintaining word-level adequacy. Although the method introduces additional computational overhead and requires a carefully defined post-editing policy to avoid occasional LLM-induced semantic drift, it offers a strong practical balance between literal fidelity and naturalness.

### H.2.1 RESULTS FOR AI4BHARAT/IN22-GEN USING ENSEMBLE (MT + LLM)

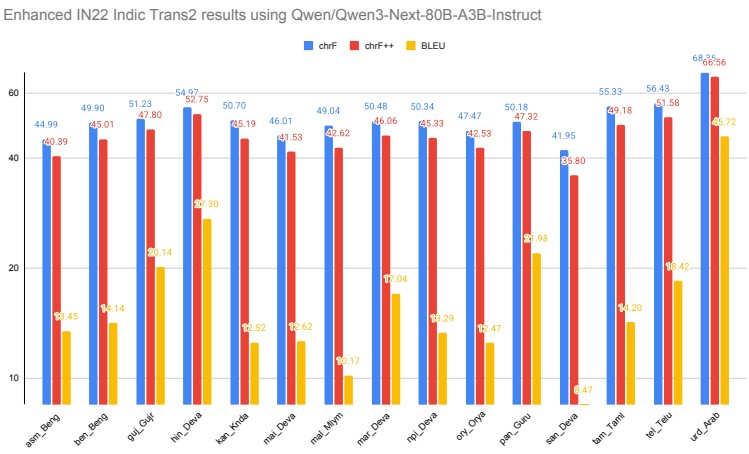

(a) Post-editing with Qwen/Qwen3-Next-80B-A3B-Instruct

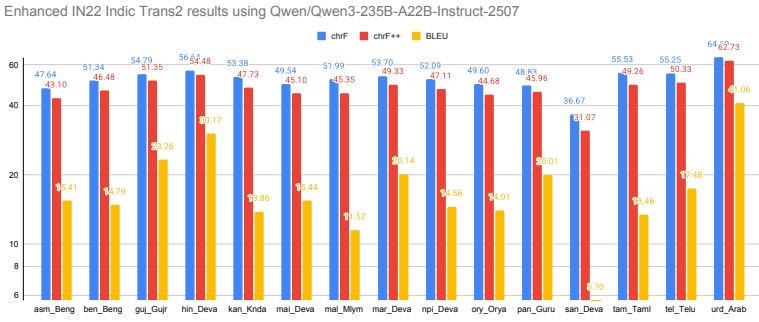

(b) Post-editing with Qwen/Qwen3-235B-A22B-Instruct-2507

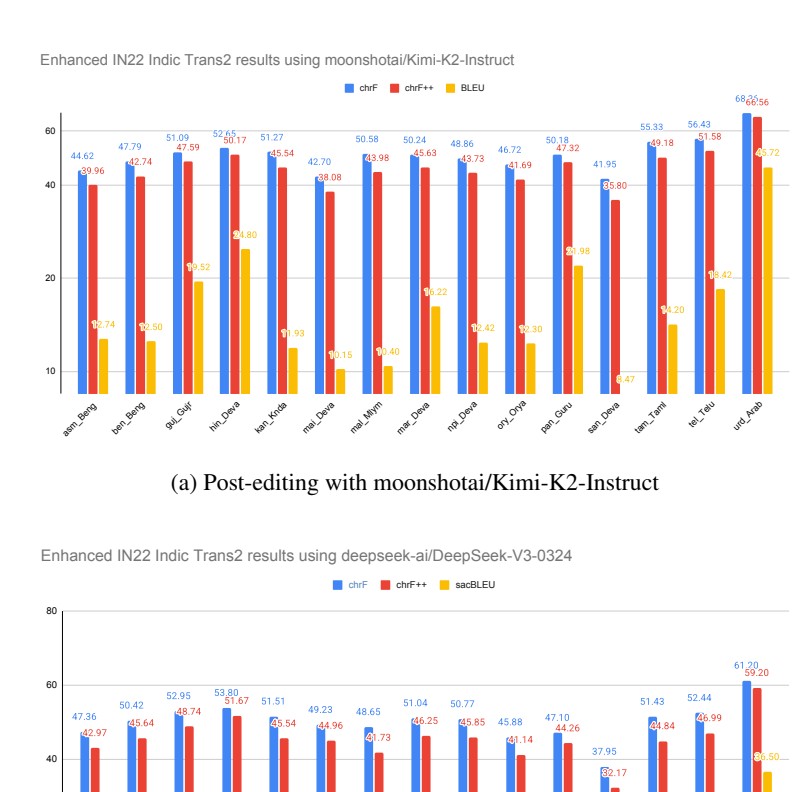

(a) Post-editing with moonshotai/Kimi-K2-Instruct

(b) Post-editing with deepseek-ai/DeepSeek-V3-0324

Figure 33: Evaluation of post-enhanced IndicTrans2 translations on the **AI4BHARAT/IN22-GEN** benchmark assessed using CHRF, CHRF++, and SACREBLEU.

## H.2.2 RESULTS FOR GOOGLE/INDICGENBENCH_FLORES_IN USING ENSEMBLE (MT + LLM)

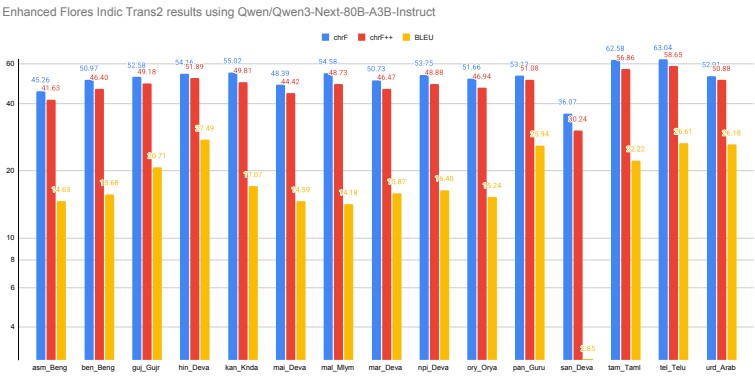

(a) Qwen/Qwen3-Next-80B-A3B-Instruct

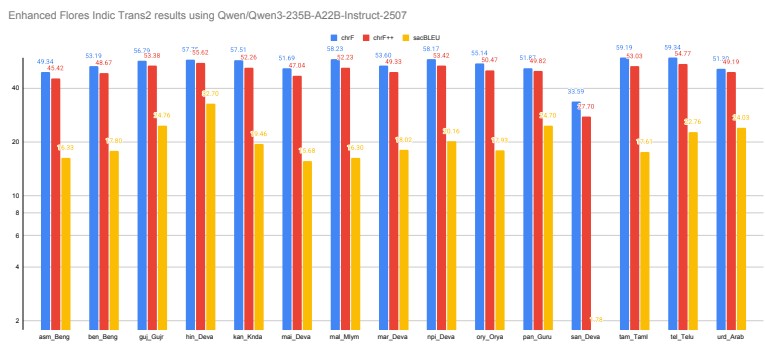

(b) Qwen/Qwen3-235B-A22B-Instruct-2507

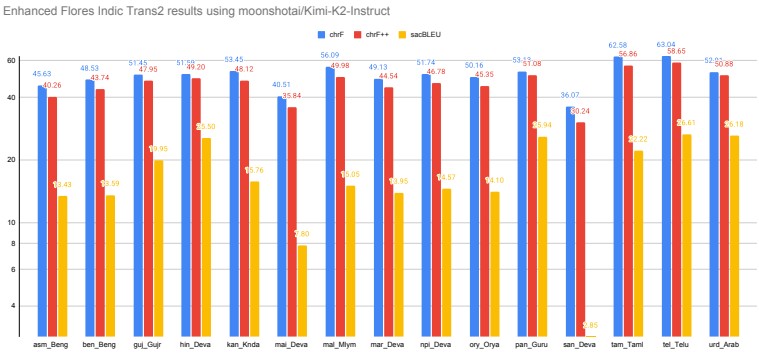

(c) moonshotai/Kimi-K2-Instruct

Figure 34: Evaluation of post-enhanced IndicTrans2 translations on the **google/IndicGenBench_flores_in** benchmark assessed using CHRF, CHRF++, and SACRE-BLEU.

# I  OCR BENCHMARK RESULTS

The following figures present representative Models average performance across benchmarks by Indic scripts and English for open source OCR/VLM Models multiple.

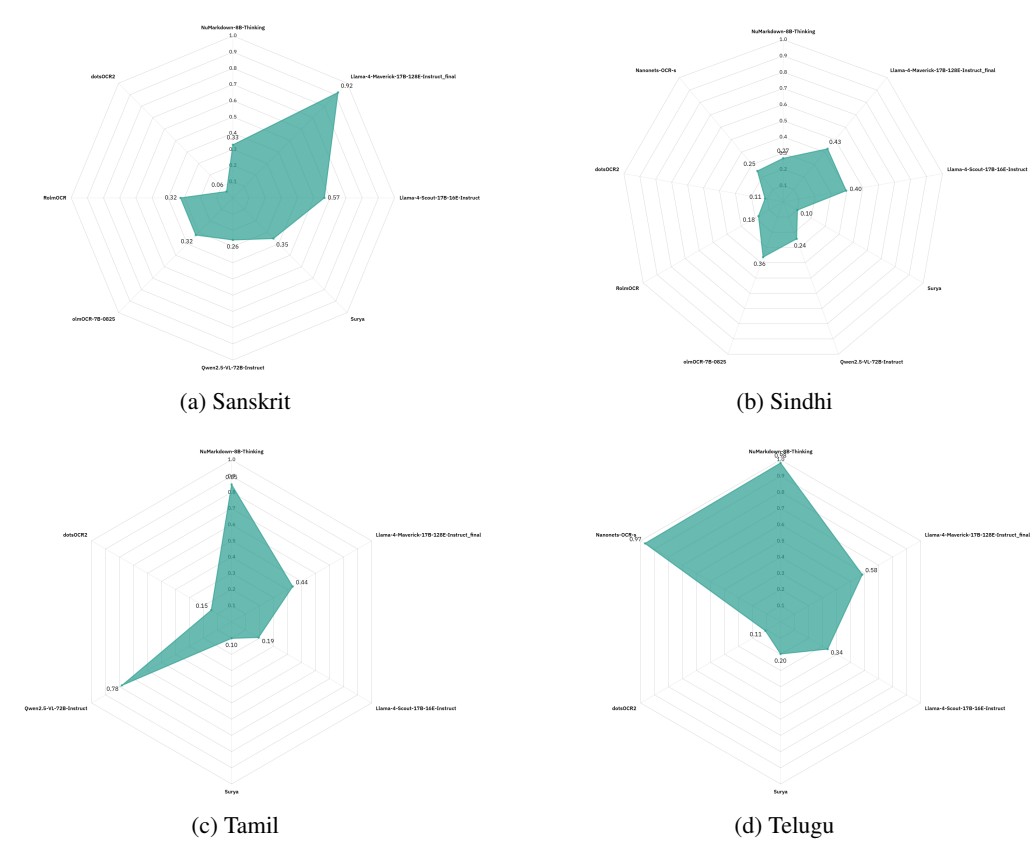

(a) Sanskrit

(b) Sindhi

(c) Tamil

(d) Telugu

Figure 35: Representative OCR outputs: Sanskrit, Sindhi, Tamil, Telugu.

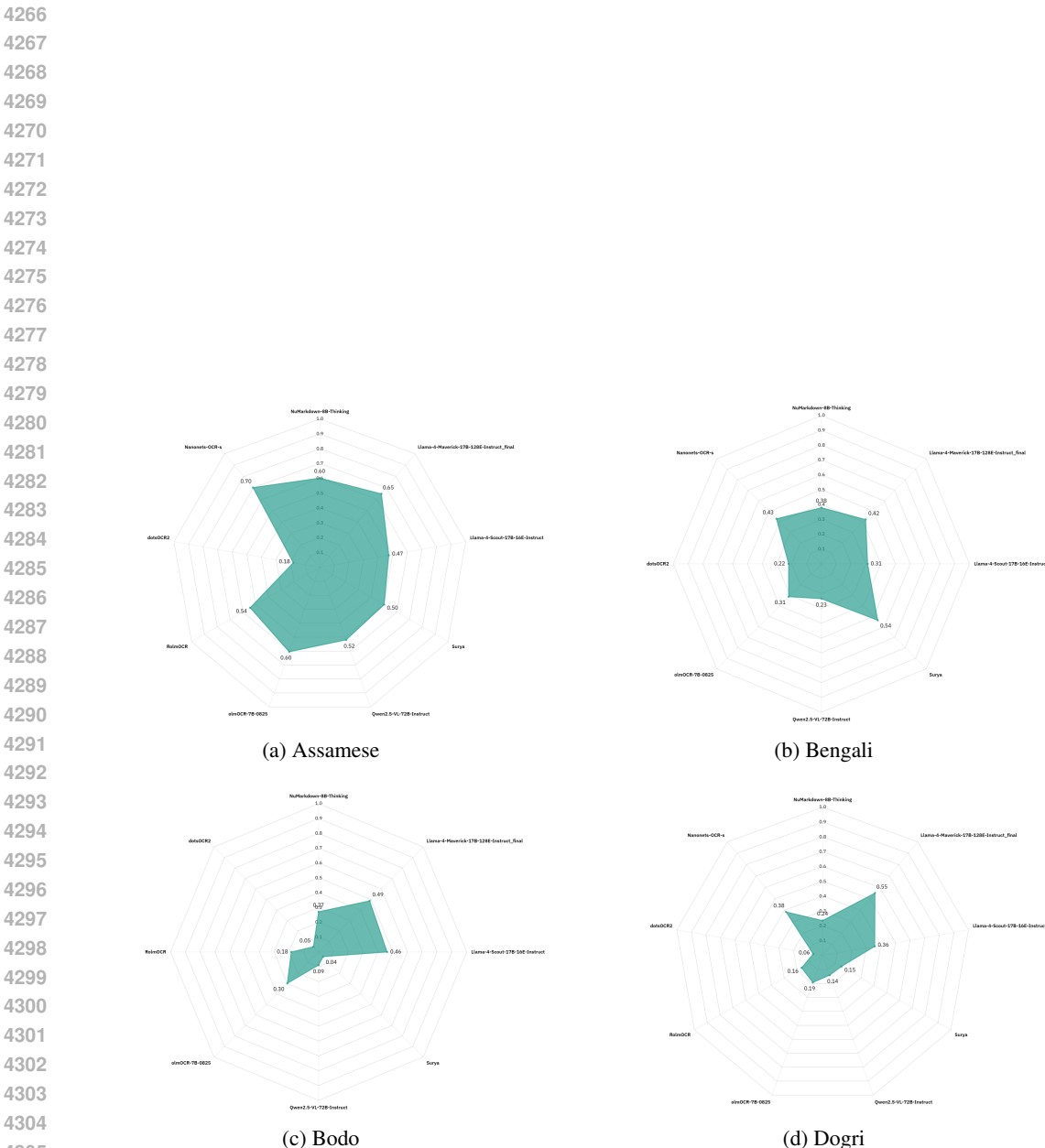

(a) Assamese

(b) Bengali

(c) Bodo

(d) Dogri

Figure 36: Representative OCR outputs: Assamese, Bengali, Bodo, Dogri.

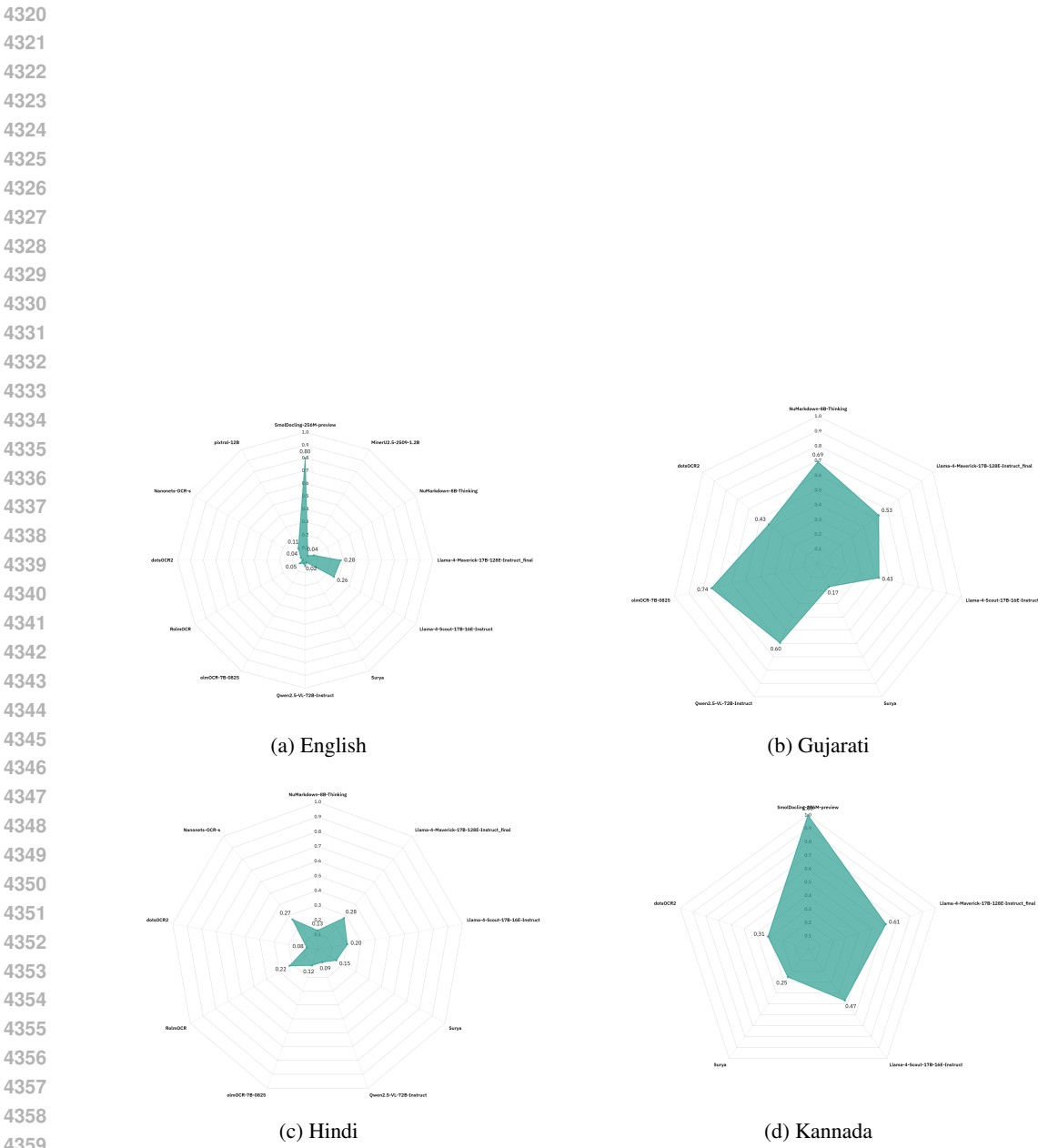

(a) English

(b) Gujarati

(c) Hindi

(d) Kannada

Figure 37: Representative OCR outputs: English, Gujarati, Hindi, Kannada.

