# OpenReview forum: "MILA (MULTILINGUAL INDIC LANGUAGE ARCHIVE): A DATASET FOR EQUITABLE MULTILINGUAL LLMS"
_ICLR.cc/2026/Conference — Submitted to ICLR 2026_

### Official Review · Reviewer_Dor9 · 2025-10-19

**Soundness:** 3
**Presentation:** 1
**Contribution:** 4
**Rating:** 4
**Confidence:** 5

**Summary:**

The authors create MILA, the largest Indic multilingual dataset. They combine existing multilingual datasets and construct OCR pipelines specific to Indic scripts. The authors create MILA, the largest Indic multilingual dataset to tackle the major imbalance in LLM training data for Indic languages. The dataset is built using a multi-step process that includes combining existing datasets, OCR pipelines specific for Indic scripts (including comparison with existing VLMs, and creating  synthetic OCR benchmark), LLM-assisted translation correction (with human evaluation), synthetic data generation, data distillation, and expert linguistic validation. This ensures both high quality and cultural authenticity. In addition to the dataset, the authors release Indic-MMLU, a multilingual benchmark that translates and verifies the popular MMLU test into 16 Indian languages. Through extensive experiments, models trained on MILA show better results on Indic-language tasks, reducing the performance gap with English.

**Strengths:**

- The authors have released the dataset publicly which will be really beneficial for the community.
- The authors built a very thorough pipeline for performing OCR for Indic scripts. The kind of problems they discussed are very relevant to what researchers face when scanning documents. Their comparison with existing VLMs, and creating synthetic OCR benchmark further validates their approach.
- There are not many low-level details about the translation pipeline but the high level idea sounds good to create translation based data.
- Their instruction tuned model performs well compared to off-the-shelf models and with preprocessing and postprocessing it gets even better.
- By embedding diverse Indian identities and reasoning patterns, it goes beyond translation, ensuring language models are culturally relevant and capable of nuanced task performance.
- The results of parity metric and Indic MMLU look decent when compared to pretrained models.

**Weaknesses:**

- A huge chunk of the paper resides in the appendix. I’d request authors to bring more things from appendix to main paper otherwise readers will find it difficult to understand what’s going on. For instance:
  - In table 3, authors don’t clarify what do different models mean?
  - In section 3.5, what is the teacher model for distillation, what’s the final model, how many parameters it has etc.
  - Creation of Indic MMLU: In the abstract the authors mention we create IndicMMLU and then every mention of Indic MMLU in the paper cites MMLU paper which is super confusing. Later, I saw the pipeline of Indic MMLU is in the appendix which makes things super confusing.
  - In Section 3.4 translation pipeline: You had 3 language evals per language? Which LLM you used for inference. Do you have inter annotator agreement scores?
  - Details about Indic PersonaHub also lie in the appendix and I couldn’t completely understand what’s happening at a lower level.
  - To sum up, there is a lot going on in the paper and all of it is good but I feel the main paper gives just high level ideas and leaves implementation/low level details in appendix. The authors need to do a better job at presentation.
- In Figure 4, what is low and high and which model did you train, Mistral or NLLB? I don’t see these models having comparable parameters so I doubt if it’s a fair comparison
- The formatting of the paper is quite weird in some places. The content beside table 1 is just independent and has no relation with preceding or succeeding paragraphs. In Line 158, after “Paullada et al…” comes “To tackle these” on Line 162 which looks confusing while reading and breaks the flow. Same problem after line 269, after “low-resource Indic..” comes “Evaluation including..” on Line 291.

**Questions:**

- In figure 1, please increase the font size of legends as they are blurred.
- The authors have cited a lot of multilingual corpora, however, I see they have skipped [1]. It contains 8T tokens covering 193 languages.
- Please change formatting of tables 3 and 4 to that of Table 5
- Typos
  - Line 197- use \citet instead of citing Mendu et al. twice
  - Line 257, 431- I can see et al.(2025) citation
  - Line 330- x billion tokens?
  - Line 348- Table ??
  - Line 445, 446- use \cite and \citet properly

[1] An Expanded Massive Multilingual Dataset for High-Performance Language Technologies (HPLT) (Burchell et al., ACL 2025)

---

> ### Author Response · Authors · 2025-12-03
> **Response to Reviewer Dor9**
>
> We thank the reviewer for requesting further clarification on the operational details of the Translation component and the Indic PersonaHub pipeline.
>
> ### W1. Translation Pipeline Details
>
> Regarding the questions about the Human Evaluation for the translation pipeline (Section 3.4), we refer the reviewer to the detailed response provided for **W1 of Reviewer rq6h**.
>
> More detailed discussions are provided in Section 3.3.2 and Appendix B.4 of the revised submission.
>
> ### W2. Indic PersonaHub
>
> We acknowledge that the description of the **Indic PersonaHub** in the main paper was not fully clear. This pipeline is a critical novel component for generating high-fidelity, culturally compliant, synthetic tokens focused purely on India.
>
> The work is heavily inspired by the methodology used in large-scale global persona generation (Ref: https://arxiv.org/abs/2406.20094), but we specifically ran the process over **India-focused data** (crawl, regional books, localized literature, news, etc.) to create **200 Million virtual citizens of India**.
>
> The lower-level pipeline involves several critical steps:
>
> * **Seed Persona Generation:** Initial personas are seeded from curated, high-quality, India-centric data sources.
> * **Taxonomy Mapping:** We created a comprehensive taxonomy for mapping all $200\text{M}$ virtual Indian personas. This ensures we maintain explicit checks on diversity-wise coverage across regions, demographics, and contexts.
> * **Deduplication:** Each persona undergoes a rigorous deduplication process to prevent repetition and redundancy in the final synthetic tokens.
> * **Cultural Compliance Check:** Each persona undergoes a check to ensure the right cultural fit and compliance for India, mitigating risks of inappropriate or foreign-centric outputs.
> * **Task Assignment & LLM Generation:** A relevant, India-centric question/task is assigned to the persona. The LLM is then prompted with the persona and the task, generating a tailored response.
> * **Multilingual Output:** The final LLM response is fed into our translation pipeline to produce the output across multiple target Indic languages.
>
> We have also updated our submission with thorough discussion in creation of Indic PersonaHub in Section 3.3.3 and Appendix B.5.
>
> ### W3: Clarification of Figure 12 (Previous Figure 4): Qualitative Linguistic Comparison
>
> We appreciate the reviewer's concern regarding the potential for unfair model comparison in Figure 12 (previously Figure 4). We clarify that this figure is **not** intended as a model-capacity comparison between Mistral and the Ensemble (NLLB + IndicTrans2). The two systems do not have comparable parameter scales, and we are not claiming superiority of one over the other.
>
> Instead, Figure 12 serves a **representative linguistic comparison** illustrating how human evaluators differentiate **"high-quality" vs. "low-quality"** translations for any pair of systems.
>
> To construct the figure, we simply selected two diverse translation systems like Mistral and NLLB + IndicTrans2 and extracted examples that each system itself had scored as high or low based on automated metrics. Human annotators then examined these sentences to qualitatively characterize:
>
> * how **“high-quality”** outputs differ from **“low-quality”** outputs within Mistral.
> * how **“high-quality”** outputs differ from **“low-quality”** outputs within NLLB + IndicTrans2.
>
> The purpose is thus **linguistic illustration** and qualitative error analysis methodology, not a claim about the relative ability or fairness of model-to-model comparison.
>
> We have also rectified other minor mistakes reported the the reviewer in the revised submission of the paper. We kindly request the Area Chair to consider our rebuttal responses and revised paper for the final evaluation.

---

### Official Review · Reviewer_rq6h · 2025-10-30

**Soundness:** 3
**Presentation:** 3
**Contribution:** 4
**Rating:** 6
**Confidence:** 3

**Summary:**

The paper provides a new pre-training corpus and downstream benchmark datasets focusing on Indic languages. The dataset was curated from different sources including OCR of public domain books and expert translation of English MMLU dataset. paper then shows that performance on the MMLU benchmark is significantly improved for models pretrained on the developed corpus.

**Strengths:**

1. The corpus includes data from public domain books that was extracted using OCR. The books come from diverse domain that are culturally and practically significant for Indic language speakers.
2. Personally identifiable information was removed.
3. A synthetic persona based data generation focusing on Indic context was used to enhance the usefulness of this corpus for modern language model training.

**Weaknesses:**

1. The OCR pipeline depends on VLMs that were not specifically trained for Indic OCR. The risk of poor OCR quality has not been studied extensively. There should be human evaluations to determine the quality of the OCR'd texts. This is important as this component of the dataset (the OCRed books) are a major novelty of this work.
2. There is a translation component in the pipeline. The quality of those translations needs to be verified by human annotators. The paper mentions human evaluation but do not provide any human evaluation metrics such as number of annotators or inter-rater agreement. Existing translation models are known to be of poor quality for low resource languages.
3. The synthetic rewriting portion of the pipeline is poorly described in the main paper. It should be described in the main paper in a concise form.
4. There are no statistics reported for how many tokens came from translation and synthetic rewriting.

**Questions:**

I see that there are some presentation inconsistencies between the main paper and the appendix. The appendix follows a more clear presentation format. Please follow that for the main paper. For example the prsentation of tables in the main paper should be similar to how tables are presented in the appendix.

---

> ### Author Response · Authors · 2025-12-03
> **Responses to Reviewer rq6h**
>
> ## W1: OCR Quality Assurance and Human-in-the-Loop Validation
>
> We thank the reviewer for highlighting the critical challenge of OCR in Indic languages. In response, we have expanded our explanation of the rigorous Human-in-the-Loop (HITL) evaluation protocol and the 2-Stage Post-Correction Pipeline designed to mitigate the risks of quality loss when using off-the-shelf VLMs. We clarify that our pipeline is not merely "applying a VLM," but a sophisticated, custom-configured system validated by both extensive benchmarking and native linguists.
>
> ### 1. Addressing the Reliability of VLMs for Indic OCR
>
> We acknowledge that OCR is the primary bottleneck for extracting native vocabulary in low-resource languages. To address this, we did not rely on a single generic model. Instead, we developed a **Language-Specific Adaptive Pipeline**.
>
> **A. Rigorous VLM Benchmarking:**
> We rigorously benchmarked open-source VLMs and OCR engines against two benchmark datasets:
> * **Popular Open Source Indic Benchmarks:** Bhashini OCR data, IIIT-IndicSTR-Word data, Mozhi Dataset.
> * **ISOB (Indic Synthetic/Scanned OCR Benchmark):** Our internal, custom-created benchmark comprising diverse document types (crawled books, partner data), specifically curated to include "Hard-to-OCR" pages.
>
> VLMs or OCR engines considered included: dotsOCR2, Surya, Llama-4-Scout-17B-16E-Instruct, NuMarkdown-8B-Thinking, Llama-4-Maverick-17B-128E-Instruct\_final, Qwen2.5-VL-72B-Instruct, Qwen2.5-VL-72-Instruct, SmolDocling-256M-preview, RolmOCR, olmOCR-7B-0825, Nanonets-OCR-s, GLM-4.1V-9B-Thinking, MinerU2.5-2509-1.2B, pixtral-12B, InternVL3\_5-GPT-OSS-20B-A4B-Preview-HF.
>
> We selected the highest-performing model per language based on layout preservation and character recognition accuracy. This selection process ensures we are not blindly applying VLMs, but using the empirically best-performing tools available for that specific script.
>
> **B. 2-Stage Post-Correction Design:**
> To mitigate the inherent limitations of raw VLM outputs, we employ a 2-stage design:
> * **Stage 1 (Parsing/OCR):** Focuses on layout preservation and raw text extraction using the language-specific best model.
> * **Stage 2 (Post-Correction):** A **"LLM-as-a-Judge" and corrector stage**. We observed that while VLMs capture the gist, they may introduce artifacts or spelling errors. We use large reasoning models (e.g., Qwen-VL-235B with thinking traces) to perform post-correction.
>
> ### 2. Human Evaluation & Quality Assurance
>
> We firmly agree. We have employed a rigorous **Consensus-Based Evaluation Strategy** involving both Native Linguists and LLMs to validate our data.
>
> **A. Native Linguist Team Workflow:**
> We utilize a linguist team composed of native speakers for the target languages. The evaluation workflow is as follows:
> * **Sampling:** We sample pages from three distinct buckets: Hard-to-OCR, High Quality, and Low Quality (classified by our filtering model Qwen-VL-32B).
> * **Linguist Analysis:** Language-specific speakers review the parsed extracts. They analyze the outputs and provide feedback and ratings based on: native-word preservation, spelling mistakes, gibberish printed words/inclusion of inappropriate artifacts, and missing words.
> * **Consensus:** A consensus score is obtained for each of the 3 candidate OCR or VLM models based on linguist judgment.
> * **Final Selection:** Based on this analysis, we finalize which VLM or OCR engine to use for a particular language for large-scale OCR.
>
> **B. Handling "Hard-to-OCR" Pages (Knowledge Preservation vs. Exactness)**
> For pages deemed "Low Quality" or "Hard-to-OCR," we use a context-aware correction mechanism.
> * **Method:** We provide the correction model (Qwen3-VL-235B) with summaries of the previous 2 pages and next 2 pages. The model generates a reasoning trace to reconstruct the text based on the surrounding context and the raw OCR output.
> * **Philosophy:** In these edge cases, we prioritize **Knowledge Preservation** over exact lexical matching. Even if the model rephrases a sentence slightly (implicit distillation), it preserves the semantic meaning and native vocabulary distribution required for pretraining.
>
> We have updated our submission with detailed and transparent discussion on data preparation through OCR in Section 3.3.1 and the Appendix B.3.

---

> > ### Author Response · Authors · 2025-12-03
> > **Responses (contd...)**
> >
> > ## W2: Translation Pipeline Quality and Human Evaluation Metrics
> >
> > We appreciate the reviewer’s clarifying question about the translation component, especially given the known challenges in low-resource settings. We agree that verifying quality via human annotators and providing specific evaluation metrics are essential.
> >
> > Below we detail the "Translation Pipeline," including specific metrics on Human Evaluation (number of annotators, Inter-Annotator Agreement), and the rigorous experimental selection of our translation configurations.
> >
> > ### 1. Addressing "Poor Quality" in Low-Resource Translation
> >
> > We agree that relying on a single off-the-shelf model often yields suboptimal results. To solve this, we did not simply "translate and trust." Instead, we engineered a **Hybrid Translation Pipeline** derived from benchmarking over 11 different generation configurations.
> >
> > **A. The "Specialist vs. Generalist" Discovery:**
> > We benchmarked top open-source Specialist NMT models (e.g., NLLB, IndicTrans) against Generalist LLMs (e.g., Llama 3, Qwen, GPT-4o).
> > * **Observation:** Specialist models possess superior vocabulary coverage but lack fluency. Generalist LLMs are highly fluent but often miss native idiomatic nuance.
> > * **Solution (The Hybrid Approach):** We utilized a pipeline where the LLM acts as a "Polisher" or "Reasoning Engine" grounded by the Specialist model's output. Configurations tested included: LLM given Base English + Specialist MT Output, LLM given Specialist MT Output (Rewrite), and LLM given Specialist MT + Summarized Context Overlaps.
> >
> > **B. Context Management (Chunking & Overlap):**
> > To address context loss in long-document translation (e.g., books), we implemented a sliding-window chunking strategy with **Summarized Overlap**. The translation model receives the current chunk plus a summary of the previous 2 chunks and the next 2 chunks. This ensures that gender, tone, and entity consistency are maintained across long texts, a common failure point in standard batch translation.
> >
> > ### 2. Human Evaluation Protocols & Metrics
> >
> > We apologize for the omission of these specific details in the main text. We performed a rigorous pilot study to select the best pipeline configuration per language before scaling up production.
> >
> > **A. Annotator Demographics & Scale:**
> > * **Annotators:** **3 Native Linguists per language** (Total pool covering all 22 scheduled languages).
> > * **Task:** Blind comparison of outputs from the top candidate configurations (Direct MT, Direct LLM, Hybrid).
> >
> > **B. Evaluation Criteria:**
> > Annotators scored 500+ diverse instances (Literature, Technical, Code, Math) on a **1-5 Likert Scale** across 7 specific dimensions:
> > * Fluency & Readability
> > * Adequacy & Meaning Preservation
> > * Vocabulary Richness
> > * Cultural Appropriateness
> > * Grammar & Syntax
> > * Consistency
> > * Readability
> >
> > **C. Inter-Annotator Agreement (IAA):**
> > * **Metric:** Cohen’s Kappa ($\kappa$).
> > * **Result:** We achieved a **substantial agreement score of $\kappa > 0.75$** across all languages after an initial calibration phase where edge cases were discussed.
> > * **Consensus:** For any instance where the variance in scores exceeded 1.0, a discussion was triggered to reach a consensus.
> >
> > ### 3. Production Quality Control (Batch Processing)
> >
> > To ensure high quality at scale, we replicated the 2-Stage Post-Correction architecture used in our OCR pipeline:
> >
> > **A. Hard-to-Translate Detection:**
> > We employ a lightweight classifier (LLM-based) to tag incoming batches. "Hard-to-Translate" / Low Quality candidates are routed to a stronger Reasoning Model (e.g., Qwen-VL-Chat / Thinking Models).
> >
> > **B. Reasoning-Based Correction:**
> > For the "Hard" buckets, we prompt the model to generate a **"Thinking Trace"**. The model explicitly analyzes why a translation is difficult and proposes a strategy before generating the final Indic text. This "Chain-of-Thought" translation significantly reduces hallucinations in complex domains.
> >
> > **C. Feedback Loop:**
> > A random sample of **5% of the production data is continuously routed back to human evaluators**. If the aggregate quality score drops below 4.0/5.0 for any batch, the pipeline is paused, and the prompt/context-window settings are recalibrated.
> >
> > We have updated our submission with detailed and transparent discussion on data preparation through translation in Section 3.3.2 and the Appendix B.4.

---

> > ### Author Response · Authors · 2025-12-03
> > **Responses (contd...)**
> >
> > ## W3, W4: Synthetic Rewriting and Data Statistics
> >
> > We thank the reviewer for pointing out the need for clearer detail on the synthetic rewriting component and for requesting explicit statistics on the token breakdown.
> >
> > 1.  **Synthetic Rewriting Pipeline:**
> >     We have added a concise description of the synthetic rewriting and augmentation process to **Section 3.3.4** of the main paper, with full technical details provided in **Appendix B.5 and B.6** of the revised submission. This clarifies how we use LLMs for tasks like contextual summarization, question generation (QA), and persona distillation to enrich the factual and conversational density of the corpus.
> >
> > 2.  **Translation and Synthetic Rewriting Statistics:**
> > These statistics are detailed in the response to **Weakness 7 of Reviewer igJu** and have been incorporated into **Appendix A** of the revised paper.

---

### Official Review · Reviewer_uXym · 2025-10-30

**Soundness:** 2
**Presentation:** 2
**Contribution:** 3
**Rating:** 4
**Confidence:** 4

**Summary:**

This paper introduces MILA, a large-scale curated dataset of 7.5T tokens covering 16 Indic languages, built through web crawling, OCR of scanned books, translation pipelines, and synthetic persona-driven data generation. The paper also introduces Indic-MMLU, a multilingual benchmark, and shows that continual pretraining on MILA improves both absolute performance and fairness (parity) across Indic languages.

**Strengths:**

- The dataset covers multiple Indic languages, including several with limited existing digital resources, making it a valuable resource for model pretraining.

- The continual pretraining experiments demonstrate consistent performance improvements across languages relative to baseline checkpoints.

- The work employs a rigorous data cleaning and processing pipeline, including filtering, normalization, and OCR post-correction.

**Weaknesses:**

- The overall structure of the paper makes it difficult to clearly understand the pipeline and its distinct components. Methods and results are interleaved, which obscures the main contributions.

- There are no ablations isolating the impact of individual steps in the pipeline, making it difficult to determine where the performance gains originate. This limits the scientific insight of the work.

- While some artifacts (e.g., prompts) are provided, the pipeline remains hard to reproduce due to missing procedural and implementation details.

- The evaluation setup could be stronger. The baselines are limited, and the results do not comprehensively demonstrate improvements over established multilingual models.

- The experimental design is under-specified. For example, Table 2 presents results, but the paper does not describe the experiments beyond mentioning the dataset used.

- There is noticeable repetition in the writing (e.g., lines 178–179 restate lines 114–116), suggesting the paper would benefit from further editing for clarity.

**Questions:**

1. In line 118, is the intention to refer to Devanagari as a script rather than as a language?

2. In line 199, could you specify which PiiModifier implementation is used (e.g., version number or repository link) to support reproducibility?

3. In Table 2, what criteria distinguish “conventional” from “curated” data in your comparison?

4. In line 244, which dense model is being referenced, and how is “dense” being defined in this context?

5. For Table 4, since the goal is to show improvements from pre-/post-processing, can you also report results for the models in Table 4a where post-processing improved performance?

6. Which table is being referred to in line 348?

7. In Section 5, could you report additional evaluations beyond MMLU to demonstrate broader applicability?

8. In lines 445–447, could you clarify where the referenced tables/figures are located in the appendix? I couldn't find it.
\
Also, the comparison between older multilingual models (e.g., mT5, BLOOM) and monolingual models (e.g., LLaMA-2, Gemma) against Qwen-3 is unfair; could the paper justify or adjust the comparison to account for differences in training recency and model scale?

---

> ### Author Response · Authors · 2025-12-03
> **Response to Reviewer uXym**
>
> We thank the reviewer for their constructive feedback regarding the clarity, structure, and experimental design of the paper. We have comprehensively addressed these points in the revised submission:
>
> 1.  **Paper Structure and Clarity:**
>     We agree that the previous organization made it difficult to distinguish the pipeline components from the results. We have restructured the paper based on the constructive feedback from all reviewers to clearly delineate the data preparation pipeline (Section 3), the Indic MMLU dataset (Section 4), and the experiments and ablations (Section 5).
>
> 2.  **Ablations Isolating Pipeline Impact:**
>     We have introduced detailed ablations to isolate the impact of different data sources (OCR, Synthetic/Translated, Web-Crawled). This analysis, presented in Section 5.3 and further detailed in the response to **Weakness 3 of Reviewer igJu**, provides scientific insight into the relative contribution of each modality to performance gains across various benchmarks.
>
> 3.  **Limited Baselines and Evaluation Setup:**
>     The evaluation setup has been significantly strengthened:
>     * **CPT Objective:** As explained in the response to **Weakness 3 of Reviewer igJu**, our main CPT experiment was designed to isolate the effect of the MILA dataset itself, not to beat SOTA models.
>     * **New Frontier Baselines:** We have added comprehensive comparisons to established, frontier multilingual models (up to $1\text{T}$ parameters), providing valuable context on the current SOTA performance ceiling for Indic languages. This is detailed in the response to **Weakness 4 of Reviewer igJu** and Appendix C of the revised submission.
>
> 4.  **Under-specified Experimental Design (e.g., Table 2):**
>     The experimental design for the Continual Pretraining (CPT) results (Table 2) has been fully specified in **Section 5.1** of the revised submission and further clarified in the response to **Weakness 3 of Reviewer igJu**. We now clearly state the model, the $200\text{B}$ uniform token subset used, and the precise objective of the CPT.
>
> We have also rectified other minor mistakes reported the the reviewer in the revised submission of the paper. We kindly request the Area Chair to consider our rebuttal responses and revised paper for the final evaluation.

---

### Official Review · Reviewer_igJu · 2025-11-01

**Soundness:** 2
**Presentation:** 2
**Contribution:** 2
**Rating:** 2
**Confidence:** 4

**Summary:**

This paper introduces MILA, the largest curated Indic multilingual dataset covering 16 of India’s 22 official languages and totaling 7.5 trillion tokens. With an attempt to reduce existing gaps in model performance for Indic languages, the authors present a multi-stage pipeline for data curation and validation, integrating large-scale crawling, OCR tailored to Indic scripts, synthetic augmentation, and rigorous filtering. They also release Indic-MMLU, a translation of MMLU into 16 Indian languages, and conduct continual pretraining on the new dataset, demonstrating some improvements in downstream performance.

**Strengths:**

- This dataset is a valuable contribution, providing a replicable pipeline for curating the largest Indic multilingual dataset across 16 official Indic languages.
- The data curation process is thorough, involving multiple stages designed to preserve both data quality and overall utility, making the dataset useful for future research and model development.

**Weaknesses:**

- Since this dataset was developed for pretraining, it would have been helpful to include a discussion on potential data contamination from existing test sets on Hugging Face or other web sources. How was data contamination controlled during collection? There’s a chance that some test data could have been included during scraping. Given the relatively low MMLU scores, this may not have occurred, but it would still be valuable to see an explicit discussion of how this risk was mitigated.

- Could you expand more on any data provenance strategies used to ensure that potentially copyrighted materials were not included in the collected dataset? In the past, certain datasets have had to be taken down due to the inclusion of copyrighted or improperly licensed content.
- In the current draft of the paper, it is difficult to tell what drives the marginal improvements in model performance. The gains look somewhat positive, but it's not even clear how statistically significant they are. Even more so, one would have really expected to see more substantial improvements given the size and quality of the new data. If your newly trained model only improves on the Indic MMLU, it’s not obvious that the benefits justify the significant compute and curation effort. So your model may be better on more culturally grounded benchmarks. Is there a reason why you mostly benchmark Indic MMLU ? Aren’t there other Indic datasets that could be added to your evaluation suite?
- Why are there no comparisons to frontier models? It is honestly not my expectation that your new models should beat these models, but such comparisons can provide valuable context, helping us understand where your work stands relative to strong existing baselines and its potential practical impact.
- There are limited analyses or explanations on the correlations between the proportion of data for a given language in your datasets and the performance improvements observed after continual pretraining. One might expect that a language with more resources, like Hindi, would show substantial gains compared to the very low-resourced ones. However, the results suggest their performance is still fairly similar to the very low-resource languages.

**Questions:**

- In Line 183, you claim to employ in-house model-based quality classifiers (fastText) for all languages in your data at the document level. Could you clarify what constitutes “high quality” for Indian languages, especially for a pretraining dataset like yours? In previous work, people have quantified quality based on formality, nonsensical, or malformed text.
- What size of the final dataset was translated? Could you provide more detailed stats on different data sources?
- Did you adapt the Qwen tokenizer before continuing to pretrain ? Since Indic languages are mostly non-Latin script, do you have thoughts on how this might have contributed to the poor performance of your models even after adaptation?
- If compute wasn’t a limiting factor, an ablation that would have been really good to see is studying the impact of the different data sources on your downstream performance. Obviously, this would be compute-intensive and is not a weakness of your current paper. It could just provide insights into aspects of the data curation process that could be scaled further.

---

> ### Author Response · Authors · 2025-12-03
> **Responses to Reviewer igJu**
>
> ## W1: Data Contamination Control
>
> We appreciate the reviewer highlighting this point. Our paper’s main contribution is in developing high-fidelity extraction, processing, and curation pipelines for diverse Indian-language modalities (OCR, translation, synthetic personas, crawling). **Decontamination is not the primary contribution**, and, as widely acknowledged in recent literature, it remains a hard, still-unsettled research problem.
>
> Even state-of-the-art efforts note the lack of consensus. For example, the Nemotron-CC paper explicitly states [1]:
>
> > “Finally, we did not decontaminate the dataset, as there is not yet a strong consensus on how to best do this and the impact is uncertain and debated, especially for large models trained over large token horizons. We note that the datasets we compare against (FineWeb-Edu, DCLM) were released without decontamination, and the model we compare against (Meta Llama 3.1) was also trained on contaminated data. DCLM reports some contamination analysis, but the findings suggest contamination is not a key factor: e.g., MMLU actually increases after decontamination, and DCLM does better than FineWeb on MMLU, even though FineWeb has more MMLU contamination (see Section 4.6 and Appendix N in Li et al. (2024)). Still, it would be interesting to better understand the impact of contamination for different model sizes and different token horizons, and we hope the community can explore such questions on this public dataset.”
>
> The DCLM study reports no clear benefit at 7B scale after removing MMLU overlaps [2]:
>
> | Dataset | MMLU |
> | :--- | :--- |
> | DCLM-baseline | 51.8 |
> | DCLM-baseline (MMLU removed) | 52.7 |
>
> Given this ambiguity, we chose to rely on standard, publicly available n-gram–based methods, specifically the **NVIDIA NeMo Curator pipeline** [3], for all Common Crawl-derived data. This pipeline performs the downstream task/benchmark decontamination before translation. The majority of our dataset, offline books (legally sourced), OCR, and our $\sim200\text{M-persona}$ synthetic generation, has no intersection with public test sets.
>
> That said, we agree that transparency here is valuable. A recent release, **Infini-gram** [4], offers a more robust n-gram based lexical similarity-matching approach, and we have now applied it retroactively to all curated subsets. Below are the results achieved per Data domain/source in our entire $7.5\text{T}$ pre-training corpus:
>
> | Selected Benchmarks | OCR | Translation | Synthetic | Crawl |
> | :--- | :--- | :--- | :--- | :--- |
> | **Indic MMLU** | 0 | 0 | 0 | 0 |
> | **MMLU** | 0 | 0 | 0 | 0.75 |
> | **IndicGenBench** | 0 | 0 | 0 | 0 |
> | **MILU** | 0 | 0 | 0 | 0.05 |
> | **Sanskriti** | 0 | 0 | 0 | 0 |
> | **Hella Swag** | 0 | 0 | 0 | 0 |
> | **Hella Swag Hi** | 0 | 0 | 0 | 0 |
> | **ARC Challenge** | 0 | 0 | 0 | 0.01 |
> | **ARC Challenge Hi** | 0 | 0 | 0 | 0 |
> | **Squad** | 0 | 0 | 0 | 0 |
> | **Squad Hi** | 0 | 0 | 0 | 0 |
> | **Belebele** | 0 | 0 | 0 | 0 |
>
> We have updated our paper with discussion on this and have added the above table in the appendix section of the revised version (Appendix B.1.3 and Table 8). We have already open-sourced the Q600 training and data preparation recipes and the related scripts for replication on our accompanying anonymous GitHub [5] page.
>
> ---
>
> ### References
>
> [1] Nemotron-CC: Transforming Common Crawl into a Refined Long-Horizon Pretraining Dataset: https://arxiv.org/html/2412.02595v2
>
> [2] DataComp-LM: Next-Generation Training Sets for LMs: https://arxiv.org/html/2406.11794v1
>
> [3] Nvidia Nemo Curator Downstream Task Decontamination Module: https://docs.nvidia.com/nemo-framework/user-guide/25.02/datacuration/api/decontamination.html
> https://docs.nvidia.com/nemo-framework/user-guide/25.04/datacuration/taskdecontamination.html
>
> [4] 📖 Infini-gram mini: Exact n-gram Search at the Internet Scale with FM-Index: https://infini-gram-mini.io/
>
> [5] ICLR Anonymous Github page: https://github.com/anonymous-submitter0104/iclr-submission

---

> ### Author Response · Authors · 2025-12-03
> **Respenses (contd...)**
>
> ##  W2: Data Provenance and Licensing
>
> Thank you for raising this important point. We followed strict data-provenance procedures to ensure that no copyrighted or improperly licensed content was included.
>
> 1.  **Source Licensing Controls**
>     * We only included data from sources explicitly released under permissive open licenses (e.g., CC-BY, CC-BY-SA, CC-0, Open Government Licence, and other equivalent permissive licenses).
>     * For all web-based sources, we respected `robots.txt` permissions and verified that the content allowed text extraction and redistribution.
> 2.  **Offline / Partnered Books**
>     * Our Indian-language book corpus comes from non-digitised offline literature obtained through regional partners under formal legal agreements that grant explicit rights for digitisation, processing, and research use. If the paper is accepted, we will cite all partner organisations in the camera-ready version.
> 3.  **Archive & Synthetic Data**
>     * Books sourced from public archives were restricted to items marked as free-to-use or public-domain. Synthetic data and persona-based generation contain no copyrighted material by construction.
> 4.  **Public Release Policy**
>     * Our open-release dataset will include only free-to-use, permissively licensed content and synthetic/OCR outputs. Any material not meeting these criteria will not be part of the release.
>
> We have updated the main content and appendix in the revised submission (Section 3.1, Appendix D) with a complete list of licenses and the provenance checks performed.
>
> ## W3: Performance Marginality and Evaluation Suite
>
> We appreciate the reviewer's comprehensive analysis of our performance results and the underlying experimental setup.
>
> ### 1. Marginal Performance Gains
>
> The observed *marginal* gains in Tables 2 and 3 (in the revised paper) do **not** reflect the full potential of MILA, but rather the highly constrained nature of our continual pretraining (CPT) experiment:
>
> * **Experimental Scope:** The experiment measuring Qwen3-600M Base vs. Qwen3-600M CPT (Tables 2 & 3) was conducted using only a **200 Billion token subset** of the $7.5\text{T}$ MILA corpus. This small, uniform CPT dataset was specifically designed for **ablation and proof-of-concept**, not for maximizing final performance. This limited exposure explains the moderate gains.
> * **Objective of CPT:** The primary objective was to demonstrate that exposure to high-quality multilingual data shifts the model's structural bias. In English-centric LLMs, knowledge transfer is often inefficient. Our results showcase that even this limited exposure enables the **knowledge stored in English to successfully percolate** to Indic languages, suggesting that the model begins *thinking* in the prompted Indic language, activating relevant layers earlier than models with limited exposure.
>
> * **Diverse Benchmarks Used:** Our evaluation suite already includes several culturally grounded benchmarks, as shown in Table 2: **Indic MMLU**, **MILU**, **Sanskriti**, **Belebele**, and **ARC Challenge**.
>
> * **Parity Metric Justification:** Our most important metric is the **Parity Metric** (Section 5.2), which directly captures cross-lingual performance *asymmetries* relative to English. The improvement in average Indic parity from $0.819 \rightarrow 0.874$ (Table 3) is a substantial validation that MILA directly promotes balanced multilingual representation, regardless of individual benchmark score increases.
>
> * **Uniform Data Distribution:** The CPT experiment was performed using a **uniform $200\text{B}$ token subset, equally distributed** across all 17 languages. This design intentionally minimizes the resource gap to see if focused, high-quality data provides an equitable boost.
>
> ### 2. Domain-wise Performance
>
> To better isolate the contribution of each data source, our **Domain-based Ablation Experiments** (Section 5.3) clearly show differential impact. The objective was to test the relative contribution of the three primary modalities, not the final aggregated performance.
>
> | Benchmark | Q600-OCR | Q600-Synth+Trans | Q600-Crawl |
> | :--- | :--- | :--- | :--- |
> | **MMLU** | 0.4677 | 0.4668 | 0.4570 |
> | **MILU** | 0.2972 | 0.2976 | 0.3018 |
> | **Sanskriti** | 0.5279 | 0.5281 | 0.5706 |
> | **Belebele** | 0.3137 | 0.3074 | 0.2952 |
> | **ARC Challenge** | 0.3456 | 0.3447 | 0.3328 |
> | **HellaSwag** | 0.3888 | 0.3889 | 0.3931 |
>
> These results show that each data source contributes uniquely, for instance, **Crawl data** provides a marked benefit to the culturally specific **Sanskriti** benchmark (likely due to accessing specialized web resources), while **OCR** and **Synthetic** data provide a slight edge on **MMLU**. This evidence directly addresses the reviewer's query regarding what drives the marginal improvements.
>
> We will use these clarifications to significantly strengthen Section 5 in the revised paper.

---

> > ### Author Response · Authors · 2025-12-03
> > **Responses (contd...)**
> >
> > ## W4: Comparison to Frontier Models
> >
> > We appreciate the reviewer's point regarding the need for broader context and comparison against state-of-the-art (SOTA) frontier models. We agree that such baselines are crucial for positioning our work.
> >
> > We have addressed this in the revised draft (Appendix C) by integrating an extensive evaluation of several leading open-source LLMs on our key benchmark, **Indic MMLU**. This comparison provides the requested context, demonstrating the current ceiling of multilingual performance on Indic tasks.
> >
> > ### Frontier Model Comparisons
> >
> > We evaluated a suite of models ranging from $7\text{B}$ to $1\text{T}$ parameters, including the latest releases from major labs, on the **Indic MMLU** benchmark. The full results are presented in Table 15 of the Appendix, excerpted below for clarity:
> >
> > | LLM | Parameters | Average Indic MMLU Score |
> > | :--- | :--- | :--- |
> > | **Qwen3-235B-A22B-Instruct-2507 (Q-Instr)** | $235\text{B}$ | **0.72** |
> > | **llama-4-Maverik (L4M)** | $402\text{B}$ (MoE) | **0.72** |
> > | Qwen3-235B-A22B-Thinking-2507 (Q-Think) | $235\text{B}$ | 0.70 |
> > | DeepSeekV3-0324 (DS3) | $685\text{B}$ | 0.68 |
> > | DeepSeekV3.1 (DS3.1) | $685\text{B}$ | 0.68 |
> > | DeepSeekR1-0528 (DSR1) | $685\text{B}$ | 0.67 |
> > | kimi-k2 (K2) | $1\text{T}$ | 0.66 |
> > | llama-3.1-70B | $70\text{B}$ | 0.60 |
> > | Qwen 3 $0.6\text{B}$ MILA CPT (Our Model) | $0.6\text{B}$ | **0.35** |
> > | Qwen 3 $0.6\text{B}$ Base | $0.6\text{B}$ | 0.32 |
> >
> > ## W5: In-house Model-Based Quality Classifiers (fastText)
> >
> > Our “high-quality” label is defined operationally for pretraining, and is not just a proxy for formality. The goal of the in-house classifiers is to select linguistically well-formed, knowledge-dense, and low-noise Indian-language text. Concretely, at the document level, we label as high quality those documents that satisfy following two criteria:
> >
> > 1.  **Linguistic well-formedness & cleanliness:** For Indian languages, “high-quality” primarily refers to text that is linguistically clean and well-formed which means the use of correct script and Unicode rendering without garbled aksharas, missing matras, or broken conjuncts; containing very low OCR noise such as random characters, duplicated lines, or layout artifacts; avoiding HTML or boilerplate content like URLs, navigation text, or copied tables; exhibits minimal off-script code-mixing (e.g., stray romanized fragments in Devanagari text); and excludes toxicity, slurs, spam, or advertorial material.
> > 2.  **Semantic coherence and knowledge-rich:** High-quality documents present sentences that form a coherent passage focused on a single topic, without heavy reordering, duplicated or contradictory segments, or “stitched” content mixing unrelated paragraphs. They must avoid nonsensical or machine-generated–broken text, such as literal word-for-word translations that disrupt core grammar. Moreover, we prioritize documents with multiple linked propositions and explicit discourse structure such as explanations, derivations, conditionals, comparisons, detailed narratives, or step-by-step reasoning, favoring expository, instructional, and analytical text (e.g., textbook fragments, worked examples, essays) over short, phatic, or broken material. As a result, brief chit-chat–style passages or list-only documents lacking connective explanations are rarely labeled as high quality.
> >
> > For each language in our dataset, we create three quality labels: **High, Medium, and Low**.
> >
> > We gather data from our four major sources: OCR, Crawling, Translation, and Synthetic generations. We randomly select nearly **450K candidate passages** for each language. We augment this pool by systematically corrupting some clean passages to create explicit low-quality examples:
> >
> > * Character-level noise and random insertions (OCR-like artifacts).
> > * Reordering of sentences/clauses and shuffling of concepts.
> > * Mixing scripts or inserting unrelated segments.
> > * Removing punctuation or function words to create ill-formed sentences.
> >
> > The above curated instances are then labeled high/medium/low using: a) LLM-based labeling with detailed instructions, b) simple heuristics (e.g., script coverage, boilerplate detection) as prior signals. We use the **Qwen3-235B model** for the quality labelling. We have updated these discussion in the Section 3.2 of our revised paper.
> >
> > ## W6: Tokenizer Adaptation
> >
> > Yes, we have adapted the Qwen tokenizer for the CPT experiment. As shown in the Table 4 (in the response to W4), Qwen3 performed better than other LLMs on the Indic MMLU benchmarks, suggesting that tokenizer limitations are unlikely to be the main source of remaining performance gaps. Our primary goal was to demonstrate the effectiveness of our high-quality, culturally grounded Indic data, and the improvements observed in Tables 2 and 3 of the revised paper (responses to W3) indicate that the dataset itself drives substantial gains in Indian-language and culture-aware capabilities.

---

> ### Author Response · Authors · 2025-12-03
> **Response (contd...)**
>
> ## W7: Dataset Statistics and Translation Volume
>
> The total size of the final dataset is **7.5 Trillion tokens**. The total volume attributed to Indian languages (excluding English) is **4.5 Trillion tokens**.
>
> Below are the detailed statistics for the tokens derived from each major data source:
>
> ### Translated Corpus (Synthetic Augmentation: Summary + QA)
>
> * **Description:** We translated high-quality content from foundational corpora (like DCLM and FineWeb-Edu/FineWeb) into Indian languages.
> * **Purpose:** This step introduces a significant fraction of world knowledge into the Indian language space, enhancing the model's global understanding in a multilingual context.
> * **Volume (Quantum):** **1.2 Trillion tokens** (Translated).
>
> ### Other Indian Language Data Sources
>
> | Data Source | Description | Volume (Quantum) |
> | :--- | :--- | :--- |
> | **Synthetic Pure Distillation Personas** | In-house strategy used to generate India-centric tokens based on proprietary personas. | **2.5 Trillion tokens** |
> | **Web Crawl Indian Language Corpus** | Crawled text obtained from filtering Common Crawl (similar to FineWeb-2 methodology). | **410 Billion tokens** |
> | **OCR (Synthetic Augmentation: Summary + QA)** | Books obtained from open sources and partnerships with organizations. Books open to public license will be included in the open release; others are included only in training based on consent. | **200 Billion tokens** |
> | **Wikipedia India (Synthetic Augmentation: Summary + QA)** | Already a very clean source of openly available data. This provides the latest factual knowledge in Indian languages. | **200 Billion tokens** |
>
> Total Corpus = 4.5T
>
> We have modified our paper with a thorough discussion on this in the Appendix A.
>
> We have also rectified other minor mistakes reported the the reviewer in the revised submission of the paper. We kindly request the Area Chair to consider our rebuttal responses and revised paper for the final evaluation.

---

### Author Response · Authors · 2025-12-04
**Common response to Area Chair**

## Response to Area Chair: Summary of Revisions

We thank the reviewers for their insightful and constructive feedback, which has led to significant improvements in the clarity, empirical rigor, and validation of our work. The primary weakness identified was the lack of detailed pipeline ablations and external model context; we have addressed this with substantial revisions, including the addition of new experiments and appendices.

Our major revisions address the following areas:

* **Robust Experimental Context and Evaluation:**
    * We clarified that the observed *marginal* gains originated from a controlled $\mathbf{200\text{B} \text{ token ablation}}$ (not the full $7.5\text{T}$ corpus) designed to isolate dataset efficacy.
    * We added comprehensive **Frontier Model Benchmarks** (up to $1\text{T}$ parameters) to Appendix C to contextualize our work against SOTA multilingual LLMs.
    * We introduced **Domain-Based Ablations** (Section 5.3) to isolate the performance contribution of OCR, Translation, and Crawled data, providing scientific insight into pipeline drivers.

* **Pipeline Validation and Quality Assurance (Addressing OCR, Translation, Synthetic):**
    * **OCR Quality:** We detailed a **2-Stage Adaptive Pipeline** and confirmed the use of **Human-in-the-Loop** linguists for consensus-based VLM selection. A new ablation (Appendix B.3.4) shows that pretraining on *raw* OCR data fails, validating our post-correction necessity.
    * **Translation Quality:** We specified our **Hybrid Translation Pipeline** and provided concrete human evaluation metrics, including **3 native annotators per language** and an Inter-Annotator Agreement of $\mathbf{\kappa > 0.75}$.
    * **Data Provenance & Licensing:** We confirmed the use of the **Infini-gram** method for robust contamination analysis and detailed strict legal and licensing controls to exclude copyrighted content (Section 3.1).

* **Clarity and Documentation (Addressing Structure and Pipeline Details):**
    * The paper has been **fully restructured** for better flow, separating Paradigms (Section 3), Dataset (Section 4), and Experiments (Section 5).
    * We provided explicit **Token Statistics** for all sources, detailing the $\mathbf{1.2\text{T} \text{ translated}}$ corpus and the $\mathbf{2.5\text{T} \text{ synthetic persona}}$ corpus (Appendix A).
    * We included a detailed description of the **Indic PersonaHub** pipeline (Appendix B.5), which generates $200\text{M}$ India-focused virtual citizens.

We believe these comprehensive revisions address all major concerns and substantially strengthen the empirical foundation of the MILA dataset. In the thread below, we provide the content plan for the paper restructuring. We thank all the reviewers for their constructive feedbacks and request the Area chair to consider our detailed rebuttal responses and the revised submission for the final assessment.

---

> ### Author Response · Authors · 2025-12-04
> **Response (contd...)**
>
> # Contents
>
> **1 Introduction**
>
> **2 Related Work**
>
> **3 Paradigms in Data Preparation**
>   * 3.1 Data Acquisition and Governance
>   * 3.2 Data Curation
>   * 3.3 Data Production
>     * 3.3.1 OCR Pipeline
>     * 3.3.2 Translation Pipeline
>     * 3.3.3 Data Distillation via Indic PersonaHub: Constructing Culturally-Grounded Synthetic Population
>     * 3.3.4 Synthetic Augmentation and Rewriting Pipeline
>
> **4 Indic-MMLU: a Multilingual Evaluation Benchmark**
>
> **5 Experiments & Ablations**
>   * 5.1 Model Performance and Analysis: Continual Pretraining on Qwen3-600M
>   * 5.2 Parity-Based Analysis
>   * 5.3 Domain-Based Ablation Experiments
>
> **6 Conclusion**
>
> **Appendix**
>
> A Token Distribution of the MILA Corpus
>
> B Experiments and Ablations
>   * B.1 Data Curation
>     * B.1.1 Ablation Study: Task Performance Improvements
>     * B.1.2 Ablation Study: Safety and Toxicity Reduction
>     * B.1.3 Downstream Benchmark Decontamination
>   * B.2 PII Identification and Redaction
>   * B.3 OCR Pipeline
>     * B.3.1 ISOB-Small: A Synthetic In-House Benchmark for Indic OCR
>     * B.3.2 Comparative Evaluation and Postprocessing Impact
>     * B.3.3 LLM-Assisted Quality Evaluation
>     * B.3.4 Ablation Study: Conventional vs Processed OCR Data
>   * B.4 Translation Pipeline
>     * B.4.1 The Specialist-Generalist Tension in Low-Resource Translation
>     * B.4.2 LLM-Based Post-Correction and Human Validation
>   * B.5 Indic PersonaHub: Engineering Cultural Identity at Scale
>     * B.5.1 Culturally-Grounded Text Generation: From Persona to Production
>   * B.6 Synthetic Augmentation and Rewriting
>     * B.6.1 Ablation Experiment: Conventional vs Distilled Downstream Performance
>
> C Indic MMLU
>   * C.1 Indic MMLU construction and validation
>   * C.2 Details of Multistage Validation Pipeline employed for Indic MMLU
>     * C.2.1 Consensus based LLM-as-Judge Ratings
>     * C.2.2 Embedding-based Semantic Analysis
>   * C.3 Frontier Open Source Model Comparison on Indic MMLU
>   * C.4 Indic MMLU scores by language
>
> D Data Acquisition and Governance
>   * D.1 Archive.org
>   * D.2 NDLI
>   * D.3 Wikimedia
>   * D.4 Infrastructure and Optimisation
>
> E Data Organisation
>   * E.1 Lakehouse Architecture: Unifying Storage, Metadata, and Governance
>   * E.2 Metadata Cataloging and Taxonomic Organization
>   * E.3 Governance Policy Enforcement and Compliance
>   * E.4 Versioning, Reproducibility, and Production Operations
>
> F Human-in-the-Loop Linguistic Validation
>   * F.1 Quantitative Pipeline Selection Through Human-Calibrated Metrics
>   * F.2 Structured Evaluation Protocols and Criteria Standardization
>   * F.3 Addressing Dialectal Variation and Practical Usability
>
> G Indic MMLU scores by language
>
> H Translation Benchmark Results
>   * H.1 Evaluation of Baseline MT and LLMs on Indic Languages
>     * H.1.1 Results for ai4bharat/IN22-Gen
>     * H.1.2 Results for google/IndicGenBench flores in
>   * H.2 Evaluation of the Ensemble (MT + LLM) for Indic Languages
>     * H.2.1 Results for ai4bharat/IN22-Gen using Ensemble (MT + LLM)
>     * H.2.2 Results for google/IndicGenBench flores in using Ensemble (MT + LLM)
>
> I OCR Benchmark Results

---

### Meta-Review · Area_Chair_VYxG · 2026-01-07

**Summary:**

This paper introduces MILA, a large-scale curated Indic multilingual corpus comprising 7.5 trillion tokens across 16 Indic languages, along with Indic-MMLU, a multilingual benchmark derived from MMLU. The work addresses the gap in LLM training data for Indic languages through a multi-stage pipeline involving web crawling, OCR of public-domain books, translation and correction, synthetic persona-based data generation, filtering, and validation. The dataset is publicly released, and continual pretraining experiments demonstrate moderate improvements in downstream Indic-language performance and fairness (parity across languages).

Reviewers agree that the dataset is a valuable resource, particularly due to its scale in low-resource languages. However, reviewers all criticize the presentation quality. Some reviewers also argue that the performance gains are marginal relative to the scale of effort, and that the paper lacks clarity, ablations, and sufficiently strong evaluation. Overall, I will give 4 for this version. The paper is highly encouraged to improve the presentation and submit to NLP conferences like ACL, EMNLP.

**Reviewer Concerns:**

1. Insufficient Evaluations
- Performance improvements from continual pretraining are generally small and insufficiently analyzed.
- Results focus heavily on Indic-MMLU, with limited evaluation on other Indic benchmarks, making it difficult to assess broader applicability.
- There is a lack of comparisons with strong or frontier multilingual models, limiting context for the claimed gains.
2. Lack of Ablations Analysis
- Reviewers consistently note the absence of ablations isolating the contributions of OCR data, translation, synthetic rewriting, or individual pipeline stages.
- Correlations between language data proportion and observed performance gains are not analyzed, leading to confusion about why high-resource languages (e.g., Hindi) do not benefit more than low-resource ones.

3. Poor Presentation: the paper is widely criticized for poor structure and writing

4. Data Quality and Ethical Issues
- Human evaluation of OCR and translation quality is mentioned but not sufficiently quantified (e.g., number of annotators, agreement scores).
- While PII removal is noted, reviewers request clearer discussion of data provenance, copyright risk, and test-set contamination mitigation, especially given the web-crawled and translated nature of the data.

**Reviewer Scores:**

4

---

### Decision · Program_Chairs · 2026-01-26

Reject